# GNNs Getting ComFy: Community and Feature Similarity Guided Rewiring

**Celia Rubio-Madrigal**[*1]  **Adarsh Jamadandi**[*1,2]  **Rebekka Burkholz**[1]
[1] CISPA Helmholtz Center for Information Security  [2] Universität des Saarlandes

## Abstract

Maximizing the spectral gap through graph rewiring has been proposed to enhance the performance of message-passing graph neural networks (GNNs) by addressing over-squashing. However, as we show, minimizing the spectral gap can also improve generalization. To explain this, we analyze how rewiring can benefit GNNs within the context of stochastic block models. Since spectral gap optimization primarily influences community strength, it improves performance when the community structure aligns with node labels. Building on this insight, we propose three distinct rewiring strategies that explicitly target community structure, node labels, and their alignment: (a) community structure-based rewiring (ComMa), a more computationally efficient alternative to spectral gap optimization that achieves similar goals; (b) feature similarity-based rewiring (FeaSt), which focuses on maximizing global homophily; and (c) a hybrid approach (ComFy), which enhances local feature similarity while preserving community structure to optimize label-community alignment. Extensive experiments confirm the effectiveness of these strategies and support our theoretical insights.

## 1 Introduction

Graph Neural Networks (GNNs) are a class of deep learning models that commonly adopt the paradigm of message passing (Gori et al., 2005; Scarselli et al., 2009; Bronstein et al., 2021), where information is repeatedly aggregated and diffused on the graph to generate a graph level representation that can be leveraged to perform either node-level (Kipf & Welling, 2017; Hamilton et al., 2017b; Veličković et al., 2018) or graph-level tasks (Errica et al., 2020). Although GNNs have found many applications in a wide array of fields, including Chemistry (Reiser et al., 2022), Biology (Bongini et al., 2023) and even Physics (Sanchez-Gonzalez et al., 2020; Shlomi et al., 2021), they are also known to have several limitations. For instance, GNNs can fail to distinguish simple graph structures (Leman, 1968; Morris et al., 2019; Papp et al., 2021). Some other problems include over-squashing (Alon & Yahav, 2021; Giovanni et al., 2023), where topological bottlenecks in the input graph desensitize the node features to information from distant nodes, and over-smoothing (Li et al., 2019; NT & Maehara, 2019; Oono & Suzuki, 2020; Zhou et al., 2021; Keriven, 2022), where node features tend to become indistinguishable due to repeated aggregations resulting from high model depth.

A popular approach to circumvent problems like over-squashing and over-smoothing is to make the input graph more amenable to message passing by rewiring the graph. This can be based on edge curvature (Topping et al., 2022; Giraldo et al., 2023; Nguyen et al., 2023) or maximizing the spectral gap (Karhadkar et al., 2023; Jamadandi et al., 2024). Spectral gap maximization, however, attenuates the graph's community structure. As our first contribution, we point out that also minimization, and thus an amplification of community structure, can improve the performance of GNNs, and provide a systematic analysis of the scenarios where one is preferred over the other. We argue that current rewiring techniques are limited in their effectiveness, as they do not account for the alignment between the nodes' ground truth labels and their cluster membership labels. If this 'graph-task' alignment is high, sometimes referred to as the *cluster hypothesis* (Chapelle et al., 2009), reducing the latent community structure can be detrimental to solving a task. Similarly, if the alignment is poor, spectral gap maximization can amplify the misalignment, which can still have degrading effects on GNN performance.

---

*Equal contribution. Corresponding email: `celia.rubio-madrigal@cispa.de`

We gain these insights by a theoretical analysis of random graphs drawn from the Stochastic Block Model (SBM), a paradigm model of graphs with community structure, on which we define a node classification task where we control two central quantities that determine the success of GNNs: the community strength and the graph-task alignment (§2.3). The main mechanism through which rewiring improves performance in this context is by adding edges between nodes that have similar features, and by removing edges between nodes with very different features, which would pollute each other's neighborhood aggregation and contribute to over-smoothing. Rewiring that improves feature similarity indirectly improves homophily, because nodes with the same label usually have more similar features. Our arguments thus align with the literature that suggests that homophily critically influences the performance of GNNs (Ma et al., 2022) and is also related to the alignment between the optimal kernel matrix and the adjacency matrix of the graph (Yang et al., 2024). To further corroborate our theoretical insight, we analyze in depth the effect of spectral rewiring on real-world graphs, and observe that the number of edges that improve the graph-task alignment (and thus homophily) correlate with the effectiveness of different spectral rewiring approaches.

To overcome the limitations of spectral rewiring, our theory and analysis provide insights into the mechanisms that can influence alignment and identify feature similarity as a promising additional criterion to take into account. This motivates our novel graph rewiring proposals, as shown in Figure 1. We introduce three different families of methods to study the importance of both topology and homophily, in isolation as well as in combination.

- The first one, ComMa, targets the strength of latent communities of the input graph: HigherComMa adds random intra-cluster edges and deletes inter-cluster edges, thus increasing the community structure. Its counterpart, LowerComMa, deletes intra-cluster edges and adds inter-cluster edges, lowering the community strength. These are randomized counterparts of the spectral methods, but they perform much faster, as they only require to run once a community detection algorithm at the beginning.

- The second method that we propose, FeaSt, aims to maximize the global feature similarity. It prioritizes edges according to this objective. It thus adds edges that increase the average similarity the most, and deletes existing edges that connect the least similar nodes. FeaSt performs especially well for highly homophilic graphs, as this procedure might help denoise the conflicting neighbourhoods. FeaSt therefore likes to focus on graph regions that already have a homophilic tendency, and further enhances this trend.

- ComFy rewires edges based on feature similarity, but is able to budget the number of edges according to the community they belong to, such that the effect is distributed across the graph. ComFy fares comparably well to FeaSt in homophilic settings, and outperforms other methods in heterophilic settings. Our extensive results on several GNN benchmarks prove that graph rewiring cannot be purely grounded on topological criteria, but that a combination of both topology and feature similarity is helpful.

## 1.1 RELATED WORK

**Graph rewiring.** A key component for GNNs is the input graph, since it not only acts as the data for model training but is also the computational structure on which *message passing* (Gilmer et al., 2017) is performed. Real-world graphs, however, can be noisy and sub-optimal for downstream tasks. For example, recent studies have pointed out issues like over-squashing (Alon & Yahav, 2021; Topping et al., 2022; Giovanni et al., 2023), caused by topological bottlenecks, which affect how information is diffused. This highlights the importance of the graph topology and begs the question: how can we obtain an optimal computational structure that aligns with the downstream task? Graph rewiring has emerged as a popular technique to effect changes to the edge structure. This can be done based on various criteria. For instance, Topping et al. (2022); Giraldo et al. (2023); Nguyen et al. (2023) propose to use different variants of Ricci curvature (Hamilton, 1988) to rewire the graph, while Black et al. (2023) propose the effective resistance (Chandra et al., 1996), and Banerjee et al. (2022); Deac et al. (2022) transform the input graph into an expander graph (Salez, 2021) for efficient message passing. Edges can be added or deleted and even though GNNs should be able to learn to drop task-irrelevant neighbors, trainability and expressiveness issues can limit this ability (Mustafa et al., 2023; Mustafa & Burkholz, 2024), which explains why edge deletions can also help fight over-smoothing in addition to over-squashing (Jamadandi et al., 2024).

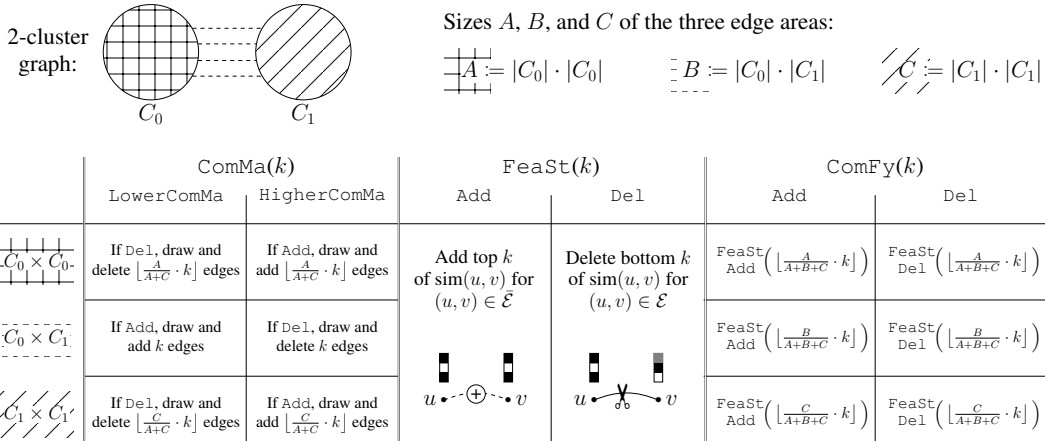

Figure 1: Behaviour of the proposed algorithms on a 2-cluster graph for $k$ edge modifications. Columns denote our methods and their variants (see §3, §B). Rows indicate 3 edge areas used for budgeting across the graph —except for FeaSt, which is global. The (latent) clusters are precomputed via Louvain. ComMa randomly draws edges from all intra or all inter-cluster areas, which is equivalent to drawing from each area with a proportional budget in expectation. This insight is translated to ComFy, but the edges are not drawn randomly but prioritized similarly to FeaSt.

**Spectral gap maximization.** Contemporaneously, spectral-based methods such as Karhadkar et al. (2023) aim to *maximize* the spectral gap by edge additions, as a larger spectral gap is inherently linked to faster mixing time (Levin et al., 2006) and thus better information flow. However, this can be detrimental in the case of heterophilic graphs (Ma et al., 2022; Platonov et al., 2023a) as we might add edges between nodes of different labels resulting in over-smoothing (Li et al., 2019; NT & Maehara, 2019; Oono & Suzuki, 2020; Zhou et al., 2021; Keriven, 2022). The spectral gap can also be maximized by deleting edges (Jamadandi et al., 2024) and this has shown to be beneficial in slowing down detrimental over-smoothing while simultaneously mitigating over-squashing, especially in heterophilic settings. Contrarily, Arnaiz-Rodríguez et al. (2022) advocate for spectral gap *minimization*, but do not explain when this could be advantageous.

**Graph and task alignment.** Our findings reveal that the underlying mechanism enhancing GNN performance by rewiring actually depends on whether we modify edges connecting nodes with similar or dissimilar features, that are usually associated with similar or dissimilar labels. In fact, Hussain et al. (2021) take a first step in this direction by analysing the interplay between community and node-labels. They propose an information-theoretic metric, and demonstrate its impact on performance by artificially creating and destroying communities in real-world graphs. This also highlights the importance of the positive influence of same-label neighbours and how different-label neighbours can impair node classification performance (Chen et al., 2020). We take this analysis several steps further and analyze why spectral rewiring cannot induce this alignment (Theorem 1).

The desirability of alignment between the graph structure and the task in GNNs has been explored in the context of their training dynamics by Yang et al. (2024). This study theoretically analyzes how GNN models tend to align their Neural Tangent Kernel (NTK) matrix $\Theta_t$ with the adjacency matrix $A$ of the input graph. They further derive a generalization bound for the NTK regime without considering node features, specifically in cases where the adjacency matrix $A$ is well-aligned with the optimal kernel matrix $\Theta^*$. This matrix $\Theta^*$ precisely indicates whether a pair of nodes share the same label, making this concept of alignment similar to ours —though not explicitly referring to the graph's communities— and to the concept of homophily. Our theory on SBMs supports this result on GNN performance, while additionally relating it to the denoising effect of node features by their neighborhoods (Theorem 2) and considering different levels of alignment (Theorem 3).

### 1.2 CONTRIBUTIONS

1. Complementing the graph rewiring literature on spectral gap maximization to fight over-squashing, we highlight real-world cases in which spectral gap minimization is more effective, contrary to

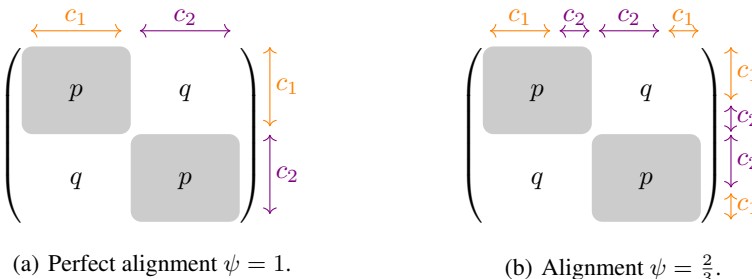

(a) Perfect alignment $\psi = 1$.

(b) Alignment $\psi = \frac{2}{3}$.

Figure 2: Adjacency matrices of $(p, q)$-SBM for different alignments. Shaded areas are intra-community edges drawn with probability $p$ (except self-loops), and unshaded areas are inter-community edges drawn with probability $q$. In Figure 2(a), the two communities match classes $c_1$ (orange) and $c_2$ (purple). In Figure 2(b), a third of nodes in each community are of the opposite class.

conventional approaches. These cases are characterized by high graph-task alignment (when community labels overlap with node labels).

2. Our theoretical insights on SBMs and experimental evidence identify the degree of task and graph structure alignment as the most critical underlying factor to explain when spectral gap rewiring improves a learning task. This highlights the major limitation of spectral-based methods, which is that they cannot improve the graph-task alignment directly.

3. To overcome this limitation, motivated by our theoretical insights, we propose to integrate feature similarity into graph rewiring approaches. We explore three novel strategies to study the effect of community structure and feature similarity in isolation (`ComMa` and `FeaSt`) and in combination (`ComFy`).

4. Extensive real-world experiments confirm our previous insights, highlighting the effectiveness of feature similarity. We find that homophilic graphs tend to benefit most from maximizing global feature similarity `FeaSt`, while heterophilic graphs gain most from a hybrid approach, `ComFy`, that maximizes feature similarity while respecting the community structure.

## 2 CONCEPTUAL ANALYSIS

### 2.1 SPECTRAL REWIRING AFFECTS COMMUNITY STRENGTH

Spectral rewiring approaches usually focus on reducing over-squashing by maximizing the spectral gap of the input graph. However, maximizing the gap has a distinct effect on its latent community structure. It is the case that, by maximizing the spectral gap, inter-community edges are added and intra-community edges are deleted, which attenuates the community strength (Theorem 1).

When there is a high graph-task alignment, which has also been termed as the cluster hypothesis (Chapelle et al., 2009), the addition of inter-community edges likely adds more inter-class edges, while the removal of intra-community edges likely deletes many intra-class edges. Consequently, message passing happens on a less informative computational structure, rendering the rewiring detrimental to the performance of any classifier (Theorem 2). On the other hand, by minimizing the spectral gap inter-community edges are deleted and intra-community edges are added, which strengthens the community structure. If this structure is highly aligned with the labels, the rewiring should be beneficial, as it increases feature similarity of nodes that have the same label, thus making different class nodes better separable.

To make these intuitive statements more rigorous and quantifiable, we relate community structure and node labels in a paradigmatic example of community structure: the Stochastic Block Model (SBM $(p, q, \mathcal{C})$), which is a random graph model with planted communities. The nodes are partitioned into $\mathcal{C}$ communities —we adopt a binary SBM ($\mathcal{C} = 2$) unless explicitly stated otherwise. We can observe the form of the adjacency matrix of a two-block $(p, q)$ SBM in Figure 2. The edges are randomly sampled with probabilities $p$ for intra-community edges and $q$ for inter-community edges. Both values critically influence the performance of GNNs on a sampled graph, as they determine the amount

of neighborhood aggregation. High values of $p$ and low values of $q$ lead to a strong, pronounced community structure. Thus, the node features after message passing tend to become more similar within communities in this setting. Similar values of $p \approx q$ would make the community structure difficult to detect and the feature distributions of different communities would not necessarily become more distinguishable after neighborhood aggregation.

To relate this reasoning to spectral gap optimization, we first establish a direct link to the community structure in SBMs.

**Theorem 1** (A less pronounced community structure corresponds to a higher spectral gap). *Let $G$ be a ($p$-$q$)-SBM with $N$ nodes in 2 equally-sized communities and intra/inter-edge probabilities $p > q$. Let $G^{del}$ be a ($p'$-$q$)-SBM where $p' < p$, and $G^{add}$ be a ($p$-$q'$)-SBM where $q' > q$. The (expected) spectral gap of $G$ is smaller than those of $G^{del}$ and $G^{add}$: $\lambda_1(G) < \lambda_1(G^{del})$, and $\lambda_1(G) < \lambda_1(G^{add})$. In fact, the spectral gap grows approximately like $-\frac{p-q}{q+p}$.*

In summary, increasing $q$ and decreasing $p$ increases the spectral gap but makes the community structure less pronounced, and vice versa. The next theorem establishes how this is connected to the performance of a model that performs sum aggregation, which we use as a tractable GNN proxy.

**Theorem 2** (A less pronounced community structure harms performance —if high graph-task alignment). *Let $G$ be the ($p$-$q$)-SBM from Theorem 1. Let $x_i$ be the single feature of node $i$ where $x_i \sim \mathcal{N}(-1, 1)$ if its class $\ell_i = c_1$ or $x_i \sim \mathcal{N}(1, 1)$ if its class $\ell_i = c_2$, and $\ell_i$ corresponds one-to-one to node $i$'s block membership. Let $f$ be an optimal classifier on the model's features, $X$, and $e(f, X)$ the (expected) proportion of misclassified nodes. After a step of sum aggregation, $e$ is monotonically decreasing with respect to $p$, and increasing with respect to $q$.*

## 2.2 VARYING THE AMOUNT OF GRAPH-TASK ALIGNMENT

Theorem 2 applies to an SBM with perfect alignment between its clusters and node labels. However, in real-world graphs, this assumption is rarely satisfied. The relationship between the task and the underlying community structure, which might not necessarily be pronounced, can take more complex forms. For instance, in heterophilic settings, similar nodes do not need to be connected, so the effect of spectral rewiring on them is not straightforward. While spectral rewiring can influence performance by modifying how pronounced the latent community structure is, aggregation on the input graph is much more effective if we improve the mentioned alignment directly, which spectral rewiring fails to do.

This intuition is corroborated and quantified by our theory. Theorem 3 describes the behaviour of the proportion of misclassified nodes after a step of neighborhood aggregation. Let $\psi$ capture the graph-task alignment. An illustration of an SBM with $\psi \neq 1$ can be found in Figure 2(b). If $\psi = 1$, we obtain the same behaviour (perfect alignment) as in Theorem 2. With $\psi = 0$, we obtain an SBM where the node labels are assigned oppositely to their communities, so by renaming the communities we also have perfect alignment. For $\psi = 0.5$, $P(M) = \Phi(0) = \frac{1}{2}$, so half the nodes are misclassified and this classifier is as good as a random choice. In this setup, most of the real distributions of neighbours follow binomials. For better interpretability, we have simplified the formula with normal approximations to look at the continuous trends. All nuances are derived in the proof (§A.3), which suggests that the central $\psi$ parameter controls GNN performance.

**Theorem 3** (The effect of different alignments on performance). *Let $G$ be the ($p$-$q$)-SBM from Theorem 1 ($p > q$). Let $x_i$ be the single feature of node $i$ where $x_i \sim \mathcal{N}(-1, 1)$ or $x_i \sim \mathcal{N}(1, 1)$ depending on its class, and $\ell_i$ its label, which may correspond to node $i$'s block membership with a fixed probability $\psi$. After a step of sum aggregation, the proportion of misclassified nodes of the best classifier $f$ is approximately*

$$P(M) \approx 1 - \psi + (2\psi - 1)\Phi\left(\frac{\frac{N}{2}(2\psi - 1)(p - q)}{\sqrt{\frac{N}{2}(p + q + p(1 - p) + q(1 - q) + 2(p - q)^2\psi(1 - \psi))}}\right)$$

## 2.3 EXPERIMENTS ON SBM FOR DIFFERENT $p$ AND $q$

The previously stated theorems are also supported by empirical results. Theorem 1 proves that maximizing the spectral gap results in a weaker latent community structure, while minimization

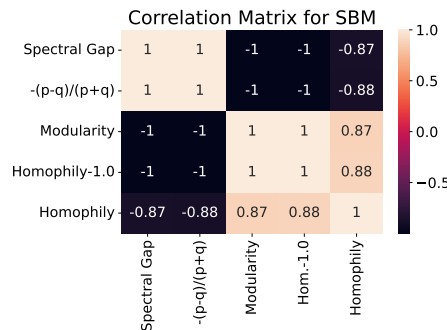

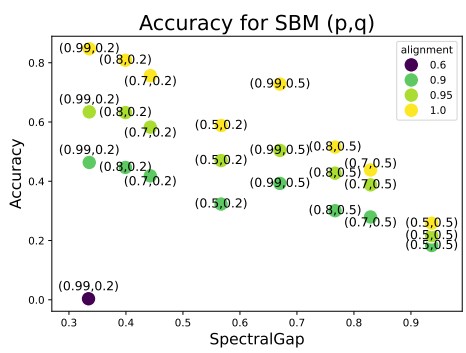

(a) Correlation of some scores for different values of $(p, q)$.

(b) Accuracy of a GCN trained on different $(p, q)$ (averaged for 8 different seeds).

Figure 3: The effects of Theorem 1 (for the spectral gap) and Theorems 2, 3 (for accuracy) on 1000-node SBM-$(p, q)$. Each SBM has different $p$ and $q$, where $p = \{0.5, 0.7, 0.8, 0.99\}$ and $q = \{0.2, 0.5\}$, and different alignment between the labels and the communities: $\{0.9, 0.95, 1\}$, as well as an example of $0.6$ alignment which gets practically null performance. The spectral gap correlates perfectly with $-\frac{p-q}{p+q}$, and negatively with the community structure and the homophily with perfect alignment. Thus, it is equivalent to plot Figure 3(b) with any of these as the x-axis.

enhances it. To quantify the impact of the spectral gap on the performance, we sample SBM graphs with normally distributed node features, whose means indicate their class membership. The class memberships are sampled from independent Bernoulli distributions whose probability (the alignment) depends on a node's community label. For different values of $p$ and $q$, we train a 2-layered GCN (Kipf & Welling, 2017) and measure the Normalized Mutual Information (NMI) (Chen et al., 2019) between the ground truth labels and the predictions made by the GCN, which we show in Figure 3(b).

Figure 3(a) furthermore validates that the spectral gap correlates with $-\frac{p-q}{q+p}$ and the community strength of the SBM (negatively), as well as with the graph's normalized homophily score *when the alignment is perfect*. When the alignment is weaker, the homophily also decreases homogeneously. In Figure 3(b), we compare the spectral gap of these different SBM graphs against the accuracy of a GCN trained on it, using a fixed train-test split.

We find that, in cases of high homophily and high alignment, it is beneficial to minimize the spectral gap, as the communities that get strengthened also correspond to the task labels. However, the spectral gap does not completely correlate with the GCN accuracy, as it can only affect the community strength. We can also see that a lack of graph-task alignment reduces the GNN performance, as shown by the different hues in the scatter plot. Changing the alignment only from $1.0$ to $0.95$ reduces dramatically the influence of different $(p, q)$ on the performance. But even given a fixed theoretical alignment, the topology of the graph can have nuanced effects on GNN accuracy. For instance, the SBM-$(0.5, 0.2)$ has a lower spectral gap (and higher homophily) than the SBM-$(0.99, 0.5)$, although a worse test performance. Yet, the latter has a higher density, which means it is potentially better at denoising and obtaining better separable node representations. This observation highlights potential benefits resulting from adding edges (and thus increasing the graph density) even without considering feature similarity or graph-task alignments.

## 2.4 ANALYSIS OF REAL-WORLD DATASETS

Real-world datasets usually have complex community structures and mixed alignment trends. Some parts of the graph might show good graph-task alignment while other parts do not invite for spectral-based rewiring. This makes it difficult to predict when minimization or maximization works best or how many edge modifications are required to see changes in GNN performance. On the one hand, very homophilic datasets might be similar to the SBM setup analyzed in the previous theorems, so spectral maximization is detrimental in the long run —as seen for Cora and Citeseer in Figure 4, where the alignment between labels and communities gets heavily reduced, and so does the accuracy. On

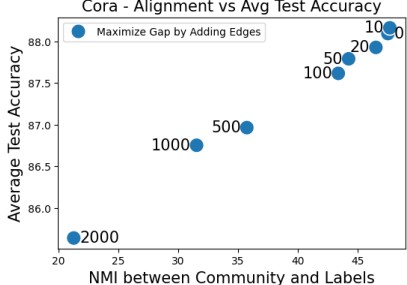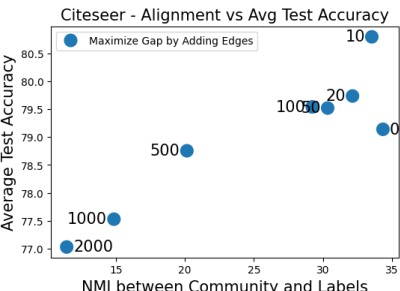

Figure 4: Maximizing the spectral gap (using (Jamadandi et al., 2024)) on Cora and Citeseer reduces both the graph-task alignment and the test accuracy. Labels denote the number of edge additions.

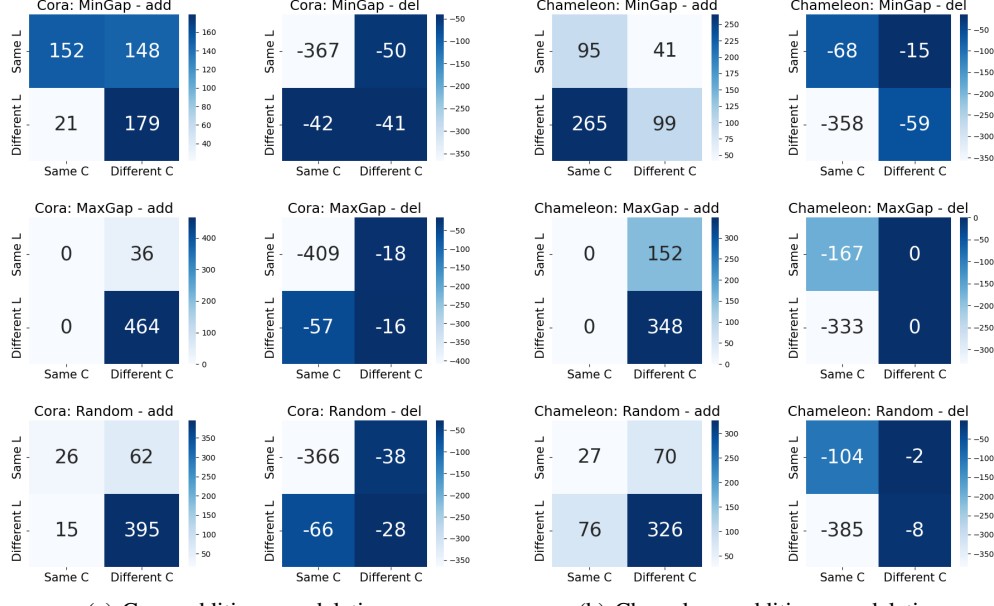

(a) Cora: additions vs. deletions    (b) Chameleon: additions vs. deletions

Figure 5: Alignment matrices for Cora (homophilic) and Chameleon (heterophilic) by a 500-edge rewiring method. In each row: spectral minimization and maximization from Jamadandi et al. (2024), and random rewiring. In each column: additions and deletions. Each alignment matrix compares the number of edges added/deleted in terms of the type of nodes it connects: with the Same or Different L(abel), and with the Same or Different C(ommunity).

the other hand, increasing connectivity might be key for some tasks, where, for example, information needs to travel across different clusters. All kinds of spectral rewiring methods can be effective for a small number of edge changes, as they might locally have a denoising effect for some (lucky) edges.

However, the trend variability for spectral rewiring might be explained by the type of edges it adds or deletes, considering both the node and community labels that they connect. Figure 5 visualizes the number of edges that connect nodes with the same or different node and community labels, for spectral minimization, maximization, and random rewiring of 500 edges, for both Cora and Chameleon. We use the spectral gap optimization algorithms presented in Jamadandi et al. (2024), as they are reliable in maximizing the spectral gap for additions and deletions, and we adapt them for minimization (as described in Algs. 1 and 2). The amount of edges for each type clearly changes from the homophilic to the heterophilic case for the different methods.

In the first row (spectral gap minimization), we see that minimization adds more same-community edges than the other two methods. When adding edges in homophilic settings (Cora), this is preferred,

because these same-community edges are mostly same-label edges (same C: 152/21). However, in heterophilic settings (Chameleon) the opposite is true: making the community structure more pronounced adds edges connecting different labels (same C: 95/265). Deletions are, however, more similar to random rewiring, with the exception of a subtle increase in the pruning of different-community edges for the heterophilic setting, compared to random (Different C: -15/-59).

In the second row (spectral gap maximization), the algorithm exclusively adds different-community edges. In homophilic settings, this is detrimental, as most of them will be from different classes (Different C: 36/464). However, in heterophilic settings, often nodes of the same class are connected, which helps align the community structure with the task (different C: 152/348). MaxGap also prunes almost exclusively same-community edges, which is again detrimental for the homophilic case (same C: -409/-57) but helps in the heterophilic case (same C: -167/333). The fact that spectral maximization by deletions helps especially in heterophilic settings is also supported by its strong benefits for GNN performance (Jamadandi et al., 2024).

The alignment matrices serve as a guiding principle to determine if spectral gap maximization or minimization should be preferred. However, spectral gap optimization fails to transform the input graph into a computational structure that is well aligned for the downstream task, which leaves the question, can we do better?

## 3 Graph Rewiring for Community-Node Label Alignment

**ComMa.** In our conceptual analysis, we have proven that spectral rewiring algorithms directly affect the community strength of the input graph, and that this can be detrimental to the task when there is an originally good alignment between community and node labels. Yet, pre-processing spectral rewiring methods are usually performed in a Greedy manner, and this causes the methods to affect newly obtained community structure but not the original one, which can get lost. To obtain clearer insights into the impact of community structure, we propose a non-Greedy and more efficient alternative to spectral rewiring: `ComMa`. This method modifies edges such that they increase or decrease the original community structure directly. The variant that increases community structure is called `HigherComMa` (Alg. 3), and corresponds to minimizing the spectral gap. The method that decreases it is `LowerComMa` (Alg. 4), and corresponds to maximizing the spectral gap. In general, this approach is flexible regarding the method that is applied to detect the community structure of the initial input graph. We use the Louvain algorithm (Blondel et al., 2008), as it scales to large graphs and is implemented by the library *nx_cugraph* for GPU acceleration. The non-accelerated algorithm runs in $O(|\mathcal{V}| \log |\mathcal{V}|)$. Rewiring only needs to consider the edges to add ($O(|\bar{\mathcal{E}}|)$) or delete ($O(|\mathcal{E}|)$), and to randomly pick a fixed number of them (provided by a hyperparameter $N$).

**FeaSt.** To make neighborhood aggregation more homogeneous to fight over-smoothing and likely increase homophily, we propose to maximize the pairwise *feature similarity* of all connected nodes in the graph. The feature (cosine) similarity between nodes $u$ and $v$ is defined as $\text{sim}(u, v) = \frac{\langle X_u, X_v \rangle}{\|X_u\| \|X_v\|}$, where $X_u, X_v$ are the respective features of nodes $u$ and $v$. Although this operation can also be accelerated by GPU, the non-accelerated computation runs in $O(|X_u||\mathcal{V}|^2)$. We consider all edges that can be added or deleted, and we rank them according to the similarity, which we would obtain if the edges were added or deleted, respectively. The $N$ modified edges are the top ones of this ranking, which can be obtained in $O(N|\bar{\mathcal{E}}|)$ for additions or $O(N|\mathcal{E}|)$ for deletions. The concrete formulas are specified in Alg. 5.

**ComFy.** While feature similarity maximization is a well performing pre-processing rewiring approach, it suffers from complementary pitfalls to the spectral rewiring methods. For the latter, it can be disadvantageous to ignore the task. For the former (`FeaSt`), it can be disadvantageous to not account for the original community structure of the graph. Therefore, we propose to restrict the similarity maximization to edges between particular pairs of communities or pairs within a community. We call this algorithm `ComFy` (Alg. 6). In this way, the effect of rewiring is spread across the whole graph, and the original structure is proportionally accounted for. This method still requires to compute all pairwise similarity values, and to detect the graph's original communities. Afterwards, it budgets the number of edges $B_{ij}$ to modify between each pair of communities $(i, j)$ (including intra-community with $i = j$) depending on their *sizes*, and such that the total sum of budgets is approximately $N$. For each $(i, j)$, we find the top $B_{ij}$ edges that maximize the similarity of edges bridging them. The complexity of this algorithm is thus comparable to the sum of the other two algorithms.

Table 1: Accuracy on node classification comparing different rewiring schemes.

| Method | Cora | Citeseer | Pubmed | Cornell | Texas | Wisconsin | Chameleon | Squirrel | Actor |
|---|---|---|---|---|---|---|---|---|---|
| GCN | 86.12±0.36 | 77.83±0.35 | 85.57±0.11 | 35.14±1.63 | 35.14±1.50 | 38.00±1.47 | 39.33±0.59 | 31.69±0.42 | 27.24±0.21 |
| GCN+BORF | 87.50±0.20 | 73.80±0.20 | NA | 50.80±1.10 | NA | 50.30±0.90 | **61.50±0.40** | NA | NA |
| GCN+FoSR | 83.50±0.39 | 75.47±0.31 | 86.08±0.10 | 40.54±1.47 | 51.35±1.75 | 54.00±1.46 | 32.36±0.37 | 27.57±0.21 | |
| GCN+ProxyAddMin | 84.10±0.39 | 78.77±0.40 | 86.15±0.10 | 45.95±1.50 | 48.65±1.45 | 42.00±1.23 | 39.33±0.55 | 33.71±0.40 | 28.03±0.22 |
| GCN+ProxyAddMax | 85.92±0.43 | 79.25±0.35 | 86.41±0.11 | 48.65±1.41 | 40.54±1.64 | 50.00±1.25 | 38.20±0.70 | 35.06±0.44 | 25.99±0.20 |
| GCN+ProxyDelMin | 85.92±0.37 | 79.01±0.34 | 86.28±0.11 | 45.95±1.50 | 48.65±1.63 | 44.00±1.13 | 39.89±0.59 | 34.83±0.45 | 26.58±0.25 |
| GCN+ProxyDelMax | 86.32±0.38 | **81.84±0.38** | 85.95±0.11 | 54.05±1.67 | 48.65±1.35 | 52.00±1.33 | 39.33±0.70 | 34.61±0.39 | 27.30±0.22 |
| GCN+HigherComMaAdd | 83.64±0.38 | 77.13±0.38 | 85.86±0.10 | 49.93±1.34 | 52.66±1.47 | 50.55±1.24 | 41.23±0.72 | 34.51±0.40 | 30.92±0.21 |
| GCN+HigherComMaDel | 83.82±0.31 | 77.31±0.41 | 85.90±0.11 | 49.03±1.26 | 48.57±1.53 | 50.32±1.38 | 40.44±0.69 | 34.66±0.39 | 30.71±0.24 |
| GCN+LowerComMaAdd | 83.41±0.37 | 77.15±0.36 | 85.85±0.09 | 51.08±1.67 | 50.29±1.71 | 50.95±1.29 | 40.61±0.64 | 34.48±0.39 | 30.79±0.23 |
| GCN+LowerComMaDel | 83.61±0.35 | 77.39±0.37 | 85.90±0.10 | 49.69±1.43 | 50.59±1.52 | 50.61±1.35 | 40.43±0.71 | 34.76±0.40 | 30.79±0.22 |
| GCN+FeaStAdd | 87.73±0.39 | 78.54±0.34 | 86.43±0.09 | 59.46±1.49 | 54.05±1.51 | 60.00±1.09 | 43.26±0.62 | **39.33±0.73** | 31.25±0.22 |
| GCN+FeaStDel | **90.74±0.39** | 81.60±0.39 | **86.76±0.10** | 51.35±1.63 | **64.86±1.43** | 60.00±1.27 | 42.70±0.69 | 36.40±0.36 | 31.97±0.21 |
| GCN+ComFyAdd | 87.73±0.26 | 77.36±0.38 | 86.74±0.10 | 67.57±1.68 | 62.16±1.52 | 62.00±1.12 | 41.57±0.83 | 36.85±0.38 | 32.30±0.25 |
| GCN+ComFyDel | 88.13±0.27 | 78.07±0.35 | 86.23±0.11 | **70.27±1.50** | 64.86±1.51 | 66.00±1.34 | 45.51±0.76 | 39.10±0.43 | 31.12±0.19 |

Table 2: Node classification on Large Heterophilic Datasets comparing different rewiring schemes.

| Method | Roman-Empire | Amazon-Ratings | Minesweeper |
|---|---|---|---|
| Baseline | 70.30±0.73 | 47.20±0.33 | 89.49±0.07 |
| GCN+FoSR | 73.60±1.11 | 49.68±0.73 | 89.66±0.04 |
| GCN+ProxyAddMin | 79.18±0.06 | 49.30±0.05 | 89.56±0.05 |
| GCN+ProxyAddMax | 77.54±0.74 | 49.72±0.41 | 89.63±0.05 |
| GCN+ProxyDelMin | 79.09±0.05 | 49.57±0.06 | 89.60±0.05 |
| GCN+ProxyDelMax | 77.45±0.68 | **49.75±0.46** | 89.58±0.04 |
| GCN+FeaStAdd | **79.67±0.07** | 49.46±0.07 | 89.75±0.05 |
| GCN+FeaStDel | 78.99±0.05 | 49.19±0.06 | 89.02±0.04 |
| GCN+FeaStAddDel | 79.03±0.07 | 49.39±0.07 | 89.62±0.05 |
| GCN+ComFyAdd | 79.53±0.07 | 49.29±0.04 | **89.76±0.05** |
| GCN+ComFyDel | 79.17±0.07 | 49.21±0.06 | 89.66±0.05 |
| GCN+ComFyAddDel | 79.27±0.06 | 49.45±0.07 | 89.40±0.08 |

## 4 EXPERIMENTS

We conduct a comprehensive set of experiments for all proposed algorithms on various benchmark datasets. Our backbone model is GCN (Kipf & Welling, 2017). Our rewiring techniques could be combined with any GNN model. We focus on a simple, common base architecture, as we compare many rewiring techniques in a comparable environment. Our proposed rewiring algorithms include: `HigherComMa` which randomly adds/deletes intra-community edges and inter-community edges respectively based on communities detected (Newman, 2006; Blondel et al., 2008); `LowerComMa` which does the opposite by randomly deleting intra-class edges and adding inter-class edges based on the communities detected; `FeaSt`, which rewires the graph to maximize the pair-wise cosine similarity between node features; and `ComFy`, a hybrid version of other two algorithms that uses both the community structure and the feature similarity to rewire the graph. We use the suffixes Add, Delete and AddDel to represent only additions, deletions, or both. Our baselines with which we compare our algorithms are spectral gap maximization methods such as FoSR (Karhadkar et al., 2023), ProxyAddMax, and ProxyDelMax proposed in Jamadandi et al. (2024). We further modify the latter algorithms to also *minimize* the spectral gap, resulting in methods ProxyAddMin (Alg. 1) and ProxyDelMin (Alg. 2). The results for the Ricci curvature-based method BORF (Nguyen et al., 2023) is directly taken from their paper —hence Not Available (NA) for a few datasets. For all tables, the best-performing methods are highlighted in **bold**, and the second best-performing methods are highlighted with underlines. More details on the hyperparameters used are described in §C.

In Table 1, we test our algorithms on a variety of homophilic and heterophilic graphs: Cora (McCallum et al., 2000), Citeseer (Sen et al., 2008), Pubmed (Namata et al., 2012), Cornell, Texas, Wisconsin, Chameleon, Squirrel, and Actor (Platonov et al., 2023c). We find that `FeaSt-Del` performs especially well for homophilic graphs. However, `ComFy-Del` seems to be in the lead for the heterophilic ones, and performs comparably for some of the homophilic ones. In Table 2 we present the results on accuracy for the large heterophilic graph benchmarks (Platonov et al., 2023c) for the spectral rewiring methods, for `FeaSt` and `ComFy`. While `FeaSt-Add` has some good results, all

Table 3: Accuracy on node classification with both additions and deletions.

| Method | Cora | Citeseer | Pubmed | Cornell | Texas | Wisconsin | Chameleon | Squirrel | Actor |
|---|---|---|---|---|---|---|---|---|---|
| GCN | 86.12±0.36 | 77.83±0.35 | 85.57±0.11 | 35.14±1.63 | 35.14±1.50 | 38.00±1.47 | 39.33±0.59 | 31.69±0.42 | 27.24±0.21 |
| GCN+BORF | 87.50±0.20 | 73.80±0.20 | NA | 50.80±1.10 | NA | 50.30±0.90 | **61.50±0.40** | NA | NA |
| GCN+FoSR | 83.50±0.39 | 75.47±0.31 | 86.08±0.10 | 40.54±1.47 | 51.35±1.75 | 54.00±1.46 | 41.01±0.63 | 32.36±0.37 | 27.57±0.21 |
| GCN+HigherComMa | 83.82±0.34 | 77.32±0.38 | 85.83±0.11 | 48.92±1.48 | 52.44±1.64 | 51.35±1.40 | 41.22±0.75 | 34.70±0.40 | 30.81±0.19 |
| GCN+LowerComMa | 83.76±0.35 | 77.05±0.37 | 85.82±0.10 | 51.46±1.49 | 50.29±1.59 | 50.42±1.27 | 40.49±0.62 | 34.11±0.38 | 30.60±0.22 |
| GCN+FeaSt | 85.71±0.36 | 80.19±0.34 | 87.01±0.12 | 54.05±1.62 | 56.76±1.65 | 58.00±1.26 | 44.94±0.70 | 35.73±0.48 | 32.63±0.21 |
| GCN+ComFy | **88.93±0.31** | **80.42±0.46** | **87.22±0.10** | **62.16±1.49** | **59.46±1.68** | **64.00±1.08** | 46.63±0.69 | **37.75±0.41** | **33.09±0.21** |

Table 4: Runtime for different rewiring schemes, in seconds, for 50 edges.

| Method | Cora | Citeseer | Chameleon | Squirrel |
|---|---|---|---|---|
| FoSR | 4.69 | 5.33 | 5.04 | 19.48 |
| ProxyAddMax | 4.30 | 3.13 | 1.15 | 9.12 |
| ProxyAddMin | 5.03 | 3.63 | 1.08 | 10.01 |
| ProxyDelMax | 1.18 | 0.86 | 1.46 | 7.26 |
| ProxyDelMin | 3.59 | 2.85 | 3.12 | 8.43 |
| ComMaAdd | 0.05 | 0.03 | 0.04 | 0.63 |
| ComMaDel | 0.05 | 0.03 | 0.04 | 0.68 |
| FeaStAdd | 1.78 | 0.92 | 0.56 | 4.43 |
| FeaStDel | 1.73 | 0.91 | 0.56 | 4.52 |
| ComFyAdd | 6.29 | 3.85 | 2.84 | 8.72 |
| ComFyDel | 6.68 | 3.73 | 2.99 | 8.97 |

`ComFy` variants seem to also perform comparably. Finally, in Table 3 we present results for both simultaneous additions and deletions for our methods.

Table 4 reports the computational efficiency compared to baselines, in seconds, when adding or deleting 50 edges. Concretely, `ComMa` is orders of magnitude faster than the spectral methods, `FeaSt` beats most of the baselines, and `ComFy` is comparable to them. The runtime of methods `HigherComMa` and `LowerComMa` are exactly the same, which we denote by `ComMa`.

## 5 CONCLUSIONS

We have introduced three novel graph rewiring techniques —`ComMa`, `FeaSt`, and `ComFy`— designed to improve the performance of Graph Neural Networks (GNNs) by focusing on the alignment between the graph structure and the target task.

Through our theoretical analysis, we have identified this alignment as a critical factor in explaining performance gains and highlighted it as a major limitation of purely topological-based rewiring strategies that they cannot improve this alignment directly. We have discussed this specifically in the context of spectral gap maximization, a widely adopted strategy to address over-squashing, which attenuates the community structure of a graph. However, when the community labels overlap with the node labels, minimizing the spectral gap (thus amplifying the community structure) would yield significant performance improvements instead.

The basic mechanism behind this improvement is the increase of feature similarity by neighborhood aggregation. In line with this finding, we have shown that rewiring techniques that explicitly take feature similarity into account, such as `FeaSt` and `ComFy`, can lead to significant performance gains, particularly in highly homophilic settings. Our proposed `ComFy` method, which balances community structure and feature similarity, was shown to outperform spectral rewiring methods in heterophilic settings, where feature alignment across different communities plays a critical role.

Our comprehensive experiments on real-world datasets confirm the effectiveness of these rewiring strategies, demonstrating that a combination of topological and feature-based approaches is key to overcoming the limitations of spectral methods. We believe that this work lays the foundation for future research on task-aware rewiring strategies, and opens the door to more sophisticated methods that leverage both graph topology and node features to optimize GNN performance across a wide range of graph-based applications.

ACKNOWLEDGMENTS AND DISCLOSURE OF FUNDING

The authors gratefully acknowledge the Gauss Centre for Supercomputing e.V. for funding this project by providing computing time on the GCS Supercomputer JUWELS at Jülich Supercomputing Centre (JSC). We also gratefully acknowledge funding from the European Research Council (ERC) under the Horizon Europe Framework Programme (HORIZON) for proposal number 101116395 SPARSE-ML.

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

APPENDIX

# A    PROOFS

## A.1    PROOF OF THEOREM 1

*Proof.* We consider an SBM with 2 classes and $\frac{N}{2}$ nodes in each class, with intra-class edge probability $p$ and interclass edge probability $q$. Its adjacency matrix $A$ is a random matrix where $A_{ij} = \text{Bernoulli}(p)$ if nodes $i, j$ are in the same cluster, and $A_{ij} = \text{Bernoulli}(q)$ otherwise. For a large $N$, the adjacency matrix $A$ can be approximated by its expected value, which is a block matrix: $\tilde{A} = \begin{pmatrix} P & Q \\ Q & P \end{pmatrix}$, where $P = p \cdot \mathbb{1}_{\frac{N}{2}} + (1-p)I_{\frac{N}{2}}$, where all values are $p$ except the diagonal which consists of ones, and $Q = q \cdot \mathbb{1}_{\frac{N}{2}}$, where all values are $q$. By summing up each row we find that the expected degree matrix $\tilde{D}$ is the diagonal matrix with entries $\tilde{D}_{ii} = 1 - p + \frac{N}{2}(p+q)$.

To find the second largest eigenvalue $\lambda_2$, we need to spectrally analyze the (expected) normalized Laplacian of $\tilde{A}$; that is, $\mathcal{L} = I - \tilde{D}^{-1/2}\tilde{A}\tilde{D}^{-1/2}$. We have that $\tilde{D}^{-1/2} = \left(\frac{1}{1-p+\frac{N}{2}(p+q)}\right)^{1/2} I_N$, so

$$\tilde{D}^{-1/2}\tilde{A}\tilde{D}^{-1/2} = \left(\frac{1}{1-p+\frac{N}{2}(p+q)}\right)\tilde{A} := \tilde{d}\tilde{A}, \text{ where we define } \tilde{d} \text{ for convenience.}$$

Then $\mathcal{L} = I - \tilde{d}\tilde{A}$. We need to find $\lambda$ such that $\det(\mathcal{L} - \lambda I) = 0$.

$$\mathcal{L} - \lambda I = I - \tilde{d}\tilde{A} - \lambda I = -\tilde{d}\tilde{A} - (\lambda-1)I = \begin{pmatrix} -\tilde{d}P - (\lambda-1)I & -\tilde{d}Q \\ -\tilde{d}Q & -\tilde{d}P - (\lambda-1)I \end{pmatrix}$$

$(-\tilde{d}P - (\lambda-1)I)$ and $-\tilde{d}Q$ commute, so by (Silvester, 2000), the determinant of that matrix is

$$\det\left(\left(-\tilde{d}P - (\lambda-1)I\right)^2 - \left(-\tilde{d}Q\right)^2\right) = \det\left(\tilde{d}(Q-P) - (\lambda-1)I\right)\det\left(-\tilde{d}(Q+P) - (\lambda-1)I\right)$$

We have that $Q - P = (q-p) \cdot \mathbb{1}_{\frac{N}{2}} + (1-p)I_{\frac{N}{2}}$, which has eigenvalues $(1-p)$ and $((q-p)\frac{N}{2} + (1-p))$, so we finally get the required eigenvalue

$$\lambda_2 = \tilde{d}((q-p)\frac{N}{2} + (1-p)) + 1 = \frac{(q-p)\frac{N}{2} + (1-p)}{(q+p)\frac{N}{2} + (1-p)} + 1$$

For $N > 2$ and $p, q \in (0,1)$: $\frac{\partial}{\partial p}\left(\frac{(q-p)\frac{N}{2}+(1-p)}{(q+p)\frac{N}{2}+(1-p)} + 1\right) = -\frac{2N(Nq+2)}{((N-2)p+Nq+2)^2} < 0$, while $\frac{\partial}{\partial q}\left(\frac{((p-q)\frac{N}{2}+(1-p))}{((p+q)\frac{N}{2}+(1-p))} + 1\right) = \frac{2N^2p}{((N-2)p+Nq+2)^2} > 0$. This proves that $\lambda_2$ increases when $p$ decreases, and when $q$ increases. So a higher spectral gap is related to a lower community structure. $\qquad\square$

### A.1.1    EXTENSIONS OF THE THEOREM

The argument still follows for a higher amount of blocks. Let $\tilde{A} = \begin{pmatrix} P & Q & Q \\ Q & \ddots & Q \\ Q & Q & P \end{pmatrix}$ with $k$ diagonal $P$ blocks of sizes $\frac{N}{k}$ each. The degree of every node is now $\tilde{D}_{ii} = 1 - p + \frac{N}{k}p + \frac{N(k-1)}{k}q$. Because of the block structure of our matrix, we still get the second eigenvalue from the difference between on and off diagonal blocks $Q - P$, which now has eigenvalues $(1-p)$ and $((q-p)\frac{N}{k} + (1-p))$. Therefore

$$\lambda_2 = \tilde{d}((q-p)\frac{N}{k} + (1-p)) + 1 = \frac{(q-p)\frac{N}{k} + (1-p)}{\frac{N}{k}p + \frac{N(k-1)}{k}q + (1-p)} + 1 = \frac{-k(p-1) - N(p-q)}{k(Nq-p+1) + N(p-q)} + 1$$

If $k$ is constant with respect to $N$, this quantity grows like $1 - \frac{p-q}{(k-1)q+p}$. If $k = aN$, then it goes to 1 as $N$ increases.

The argument also still holds for different-sized communities. Let $\tilde{A} = \begin{pmatrix} P1 & Q1 \\ Q2 & P2 \end{pmatrix}$, where $P1 = p \cdot \mathbb{1}_M + (1-p)I_M$ and $P2 = p \cdot \mathbb{1}_{N-M} + (1-p)I_{N-M}$, where all values are $p$ except the diagonal which consists of ones, and $Q1 = q \cdot \mathbb{1}_M$, $Q2 = q \cdot \mathbb{1}_{N-M}$, where all values are $q$. We assume that $M > N - M$. Then $\tilde{D}$ is the diagonal matrix with entries $\tilde{D}_{ii} = (1 + (M-1)p + (N-M)q)$ if $i < M$, and $\tilde{D}_{ii} = (1 + (N-M-1)p + Mq)$ otherwise. We define $\tilde{d}_1 = \frac{1}{1+(M-1)p+(N-M)q}$ and $\tilde{d}_2 = \frac{1}{1+(N-M-1)p+Mq}$ for convenience. Because $\tilde{d}_1 < \tilde{d}_2$, the second largest eigenvalue will come from the interactions of the first block. So the eigenvalues are $(1-p)$ and $((q-p)M + (1-p))$. Therefore

$$\lambda_2 = \tilde{d}_1((q-p)M + (1-p)) + 1 = \frac{(q-p)M + (1-p)}{(1 + (M-1)p + (N-M)q)} + 1$$

If $M = aN$, this quantity grows like $1 - \frac{a(p-q)}{ap+(1-a)q}$. If $M$ is constant, then the second block gets bigger than the first and we get the second eigenvalue from it instead.

## A.2   PROOF OF THEOREM 2

*Proof.* We consider an SBM with 2 classes and $\frac{N}{2}$ nodes in each class, with intra-class edge probability $p$ and inter-class edge probability $q$. Each node $i \in \{0, \dots, N-1\}$ has one feature, $x_i$, and a label $\ell_i$ which corresponds to the block it belongs to: $\ell_i = 1 \Leftrightarrow i \geq \frac{N}{2}$. The task is, therefore, to predict each node's community association. In this case, the alignment of communities and labels is perfect.

Each feature $x_i$ is aligned with its label following a normal distribution: class-0 node features follow $\mathcal{N}(-\mu_0, \sigma_0^2)$, while class-1 node features follow $\mathcal{N}(\mu_0, \sigma_0^2)$, as shown in Figure 6(a). A perfect classifier $f$ without any knowledge of the graph structure builds a decision boundary at $x = 0$. The expected number of misclassified nodes is $\frac{N}{2}$ times the intersection area of both distributions —because they are normalized from a population of $\frac{N}{2}$ each. Such area is $2 \cdot \Phi(\frac{-\mu_0}{\sigma_0})$, where $\Phi$ is the cumulative distribution function of the standard normal distribution (see Figure 6(b)). Therefore, the proportion of misclassifications is $e(f) = \frac{N}{2} \cdot 2 \cdot \Phi(\frac{-\mu_0}{\sigma_0}) \cdot \frac{1}{N} = \Phi(\frac{-\mu_0}{\sigma_0})$. As $\Phi$ is a cumulative function, it is monotonically increasing with respect to its argument.

The classification error of $f$ can be reduced by performing a step of message passing on the graph, which utilizes the community information to further separate the two classes. We shall consider a single round of sum aggregation as an example.

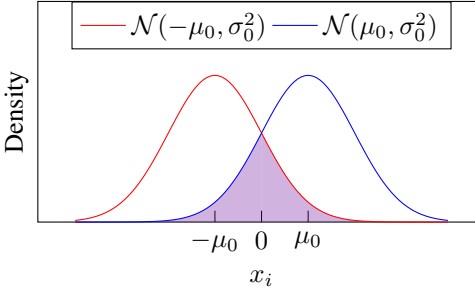 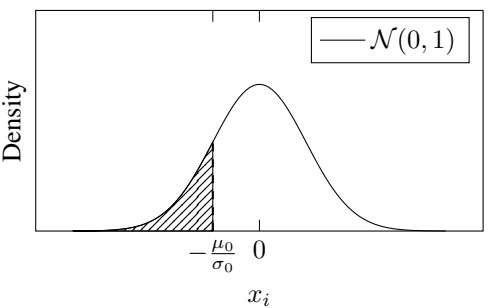

(a) The distribution of features from both clusters before training. The area in purple corresponds to nodes wrongly classified by the decision boundary $x = 0$.

(b) The cumulative distribution function at $x = -1$ of $\mathcal{N}(\mu_0, \sigma_0^2)$ is equal to the cumulative distribution function at $x = -\frac{\mu_0}{\sigma_0}$ of the standard normal distribution $\mathcal{N}(0, 1)$, which is $\Phi(\frac{-\mu_0}{\sigma_0})$. The purple area of Figure 6(a) is two times this quantity.

Figure 6: Illustration of the setup for the feature distributions for Theorem 2.

Any node has an expected $\mathcal{E}_p = p \cdot \left(\frac{N}{2} - 1\right)$ intra-class neighbors, plus itself, and an expected $\mathcal{E}_q = q \cdot \frac{N}{2}$ inter-class neighbors. In the next proof A.3 we compute the same quantity with neighbour distributions instead, for a more fine-grained approximation. The hidden state of a class-1 node $i$ after a step of sum aggregation is therefore the sum of $\mathcal{E}_p + 1$ random variables $\sim \mathcal{N}(\mu_0, \sigma_0^2)$ and $\mathcal{E}_q$ random variables $\sim \mathcal{N}(-\mu_0, \sigma_0^2)$. This follows another normal distribution with mean $\mu_1 := \mu_0 \cdot (1 + \mathcal{E}_p - \mathcal{E}_q)$ and variance $\sigma_1^2 := \sigma_0^2 \cdot (1 + \mathcal{E}_p + \mathcal{E}_q)$. Conversely, the hidden state of a class-0 node $i$ follows a normal distribution of mean $-\mu_1$ and the same variance $\sigma_1^2$. The decision boundary of a perfect classifier is still at $x = 0$, but the average proportion of misclassified nodes is now $\Phi(\frac{-\mu_1}{\sigma_1})$, which depends on $p$ and $q$. Specifically, it tends to be monotonically decreasing with respect to $p$; this means that the higher the community structure, the more accurate the classifier can be, because there is more information to utilize.

Let us take $\mu_0 = 1$ and $\sigma_0 = 1$ to simplify the calculations. We need to check that $\frac{\partial}{\partial p}\left(\frac{-\mu_1}{\sigma_1}\right) < 0$. For $N > 2$ and $p, q \in (0, 1)$:

$$\mu_1 = 1 \cdot \left(1 + p\left(\frac{N}{2} - 1\right) - q \cdot \frac{N}{2}\right) = 1 - p + \frac{N}{2} \cdot (p - q)$$

$$\sigma_1^2 = 1 \cdot \left(1 + p\left(\frac{N}{2} - 1\right) + q \cdot \frac{N}{2}\right) = 1 - p + \frac{N}{2} \cdot (p + q)$$

$$-\frac{\mu_1}{\sigma_1} = -\frac{1 - p + \frac{N}{2} \cdot (p - q)}{\sqrt{1 - p + \frac{N}{2} \cdot (p + q)}}$$

$$\frac{\partial}{\partial p}\left(\frac{-\mu_1}{\sigma_1}\right) = -\frac{(N-2)((N-2)p + 3Nq + 2)}{2\sqrt{2}\left((N-2)p + Nq + 2\right)^{\frac{3}{2}}} < 0$$

$$\iff (N-2)((N-2)p + 3Nq + 2) > 0$$

On the other side, $\frac{\partial}{\partial q}\left(\frac{-\mu_1}{\sigma_1}\right) > 0$.

$$\frac{\partial}{\partial q}\left(\frac{-\mu_1}{\sigma_1}\right) = \frac{N(6 + 3(N-2)p + Nq)}{2\sqrt{2}(2 + (N-2)p + Nq)^{\frac{3}{2}}} > 0 \iff N(6 + 3(N-2)p + Nq) > 0$$

This proves that, by reducing the community structure (either by decreasing $p$ or increasing $q$), then the quantity $\frac{-\mu_1}{\sigma_1}$ increases, so the expected proportion of misclassified nodes $e(f) = \Phi\left(\frac{-\mu_1}{\sigma_1}\right)$ also increases. In consequence, it harms the performance of classifier $f$.

The graph's information provides a better separation between the two classes if the intra-class edge probability is high enough. From this we can conclude that reducing the intra-class edge probability is not a good strategy to improve the classification performance for any model on the graph. $\qquad\square$

### A.3  PROOF OF THEOREM 3

*Proof.* We consider another SBM with 2 classes and $\frac{N}{2}$ nodes in each class, with intra-class edge probability $p$ and inter-class edge probability $q$. Each node $i \in \{0, \ldots, N-1\}$ has again one feature, $x_i$, aligned with its class label $\ell_i$ following a normal distribution: $\mathcal{N}(-\mu_0, \sigma_0^2)$ for class 0 and $\mathcal{N}(\mu_0, \sigma_0^2)$ for class 1. However, now $\ell_i$ corresponds to its community with a fixed probability $\psi$ —recovering Theorem 2 when $\psi = 1$.

What is the probability of any node $i$ such that, after a round of sum aggregation, its modified representation $x_i'$ is now misclassified ($M$)? As the two classes are symmetric:

$$P(M) = P(M, L_0) + P(M, L_1) = P(L_0)P(M|L_0) + P(L_1)P(M|L_1)$$

$$= \frac{1}{2}P(M|L_0) + \frac{1}{2}P(M|L_1) = P(M|L_0)$$

Then the question becomes the following: what is the probability of a node with label $L_0$ being misclassified? It depends whether it belongs to community $C_0$ or $C_1$.

$$P(M|L_0) = P(M, C_0|L_0) + P(M, C_1|L_0) = P(C_0|L_0)P(M|L_0, C_0) + P(C_1|L_0)P(M|L_0, C_1)$$
$$= \psi P(M|L_0, C_0) + (1 - \psi)P(M|L_0, C_1)$$

$P(M|L_0, C_0) = P(X'_{(L_0, C_0)} > 0)$. We now need to calculate what is the predicted label of a $(L_0, C_0)$ node after a sum aggregation round. For this we need the distribution of its neighbours. We consider the node to have a self loop, as it uses its own feature too.

- The number of nodes $(L_0, C_0)$ (that are not node $i$) follows a binomial distribution $N_0 \sim B(\frac{N}{2} - 1, \psi)$. However, for easiness of proof we will approximate it by a normal distribution, which is appropriate for $N$ large enough: $N_0 \sim \mathcal{N}((\frac{N}{2} - 1)\psi, (\frac{N}{2} - 1)\psi(1 - \psi))$. The amount of them connected to node $i$ follows a conditional binomial distribution $H_{00} \sim B(n_0, p) \mid N_0 = n_0$, which we again approximate by $H_{00} \sim \mathcal{N}(n_0 p, n_0 p(1 - p)) \mid N_0 = n_0$.

- The number of nodes $(L_0, C_1)$ that are connected to node $i$ follows $H_{01} \sim B(\frac{N}{2} - 1 - n_0, q) \mid N_0 = n_0$, approximated by $H_{01} \sim \mathcal{N}((\frac{N}{2} - 1 - n_0)q, (\frac{N}{2} - 1 - n_0)q(1 - q)) \mid N_0 = n_0$.

- Since $H_{00}$ and $H_{01}$ are conditionally independent given $N_0 = n_0$, their sum $H_0 = H_{00} + H_{01}$ also follows a normal distribution with parameters given by the sum of their means and variances. Thus, the number of total $L_0$ nodes connected to node $i$ (except itself) follows $H_0 \sim \mathcal{N}(n_0 p + (\frac{N}{2} - 1 - n_0)q, n_0 p(1 - p) + (\frac{N}{2} - 1 - n_0)q(1 - q)) \mid N_0 = n_0$. We are going to get rid of the dependency of $N_0$ by estimating it by a normal distribution with the mean and variance of the marginal distribution of $H_0$:

$$\mathbb{E}[H_0] = \mathbb{E}[\mathbb{E}[H_0|N_0]] = \mathbb{E}[N_0]p + \left(\frac{N}{2} - 1 - \mathbb{E}[N_0]\right)q$$
$$= \left(\frac{N}{2} - 1\right)\psi p + \left(\frac{N}{2} - 1 - \left(\frac{N}{2} - 1\right)\psi\right)q$$
$$= \left(\frac{N}{2} - 1\right)(p\psi + q(1 - \psi))$$

$$\text{Var}[H_0] = \mathbb{E}[\text{Var}(H_0|N_0)] + \text{Var}(\mathbb{E}[H_0|N_0])$$
$$= \mathbb{E}[N_0]p(1 - p) + \left(\frac{N}{2} - 1 - \mathbb{E}[N_0]\right)q(1 - q)$$
$$+ \text{Var}\left(N_0(p - q) + \left(\frac{N}{2} - 1\right)q\right)$$
$$= \left(\frac{N}{2} - 1\right)\psi p(1 - p) + \left(\frac{N}{2} - 1 - \left(\frac{N}{2} - 1\right)\psi\right)q(1 - q)$$
$$+ (p - q)^2 \left(\frac{N}{2} - 1\right)\psi(1 - \psi)$$
$$= \left(\frac{N}{2} - 1\right)(\psi p(1 - p) + (1 - \psi)q(1 - q) + (p - q)^2\psi(1 - \psi))$$

- The number of nodes $(L_1, C_1)$ follows $N_1 \sim B(\frac{N}{2}, \psi)$, approximated by $N_1 \sim \mathcal{N}(\frac{N}{2}\psi, \frac{N}{2}\psi(1 - \psi))$. The amount of them connected to node $i$ follows $H_{11} \sim B(n_1, q) \mid N_1 = n_1$, approximated by $H_{11} \sim \mathcal{N}(n_1 q, n_1 q(1 - q)) \mid N_1 = n_1$.

- The number of nodes $(L_1, C_0)$ that are connected to node $i$ follows $H_{10} \sim B(\frac{N}{2} - n_1, p) \mid N_1 = n_1$, approximated by $H_{10} \sim \mathcal{N}((\frac{N}{2} - n_1)p, (\frac{N}{2} - n_1)p(1 - p)) \mid N_1 = n_1$.

- Similarly to $L_0$, the number of total $L_1$ nodes connected to node $i$ follows $H_1 \sim \mathcal{N}(n_1 q + (\frac{N}{2} - n_1)p, n_1 q(1-q) + (\frac{N}{2} - n_1)p(1-p)) \mid N_1 = n_1$. We will estimate it by a normal distribution with its mean and variance:

$$\mathbb{E}[H_1] = \mathbb{E}[\mathbb{E}[H_1|N_1]] = \mathbb{E}[N_1]q + (\frac{N}{2} - \mathbb{E}[N_1])p$$

$$= \frac{N}{2}\psi q + \left(\frac{N}{2} - \frac{N}{2}\psi\right)p$$

$$= \frac{N}{2}(p(1-\psi) + q\psi)$$

$$\text{Var}[H_1] = \mathbb{E}[\text{Var}(H_1|N_1)] + \text{Var}(\mathbb{E}[H_1|N_1])$$

$$= \mathbb{E}[N_1]q(1-q) + \left(\frac{N}{2} - \mathbb{E}[N_1]\right)p(1-p) + \text{Var}\left(N_1(q-p) + \frac{N}{2}p\right)$$

$$= \frac{N}{2}\psi q(1-q) + \left(\frac{N}{2} - \frac{N}{2}\psi\right)p(1-p) + (p-q)^2\frac{N}{2}\psi(1-\psi)$$

$$= \frac{N}{2}(\psi q(1-q) + (1-\psi)p(1-p) + (p-q)^2\psi(1-\psi))$$

The representation of node $i$ after one step of sum aggregation is the summation of $H_0 + 1$ (independent) normal distributions $\sim \mathcal{N}(-\mu_0, \sigma_0^2)$ and $H_1$ (independent) normal distributions $\sim \mathcal{N}(\mu_0, \sigma_0^2)$. Therefore:

$$X'_{(L_0, C_0)} \sim \mathcal{N}(-\mu_0(1 + h_0 - h_1), \sigma_0^2(1 + h_0 + h_1)) \mid H_0 = h_0, \ H_1 = h_1$$

Again calculating its mean and variance:

$$\mathbb{E}[X'_{(L_0, C_0)}] = \mathbb{E}[\mathbb{E}[X'_{(L_0, C_0)}|H_0, H_1]] = -\mu_0(1 + \mathbb{E}[H_0] - \mathbb{E}[H_1])$$

$$= -\mu_0\left(1 + \left(\frac{N}{2} - 1\right)(p\psi + q(1-\psi)) - \frac{N}{2}(p(1-\psi) + q\psi)\right)$$

$$\text{Var}[X'_{(L_0, C_0)}] = \mathbb{E}[\text{Var}(X'_{(L_0, C_0)}|H_0, H_1)] + \text{Var}(\mathbb{E}[X'_{(L_0, C_0)}|H_0, H_1])$$

$$= \sigma_0^2(1 + \mathbb{E}[H_0] + \mathbb{E}[H_1]) + \mu_0^2(\text{Var}(H_0) + \text{Var}(H_1))$$

$$= \sigma_0^2\left(1 + \left(\frac{N}{2} - 1\right)(p\psi + q(1-\psi)) + \frac{N}{2}(p(1-\psi) + q\psi)\right)$$

$$+ \mu_0^2\left(\left(\frac{N}{2} - 1\right)(\psi p(1-p) + (1-\psi)q(1-q) + (p-q)^2\psi(1-\psi))\right.$$

$$\left. + \frac{N}{2}(\psi q(1-q) + (1-\psi)p(1-p) + (p-q)^2\psi(1-\psi))\right)$$

For a more clear analysis of this formula, we take $\mu_0 = 1, \sigma_0 = 1$ and $N$ large enough:

$$\mathbb{E}[X'_{(L_0, C_0)}] \approx -\frac{N}{2}(p\psi + q(1-\psi) - p(1-\psi) - q\psi) = -\frac{N}{2}(2\psi - 1)(p - q)$$

$$\text{Var}[X'_{(L_0, C_0)}] \approx \frac{N}{2}\left(p\psi + q(1-\psi) + p(1-\psi) + q\psi + 2(p-q)^2\psi(1-\psi)\right.$$

$$\left. + \psi p(1-p) + (1-\psi)q(1-q) + \psi q(1-q) + (1-\psi)p(1-p)\right)$$

$$= \frac{N}{2}(p + q + p(1-p) + q(1-q) + 2(p-q)^2\psi(1-\psi))$$

Finally, we have $P(X'_{(L_0, C_0)} > 0) \approx 1 - \Phi\left(\frac{-\mathbb{E}[X'_{(L_0, C_0)}]}{\sqrt{\text{Var}[X'_{(L_0, C_0)}]}}\right) =$

$$\Phi\left(\frac{\frac{N}{2}(2\psi - 1)(p - q)}{\sqrt{\frac{N}{2}(p + q + p(1-p) + q(1-q) + 2(p-q)^2\psi(1-\psi))}}\right)$$

---

**Algorithm 1** Proxy Spectral Gap based Greedy Graph Addition (PROXYADDMIN)

---
**Require:** Graph $\mathcal{G} = (\mathcal{V}, \mathcal{E})$, num. edges to add $N$, spectral gap $\lambda_1(\mathcal{L}_{\mathcal{G}})$, second eigenvector $f$ of $\mathcal{G}$.
  **repeat**
    **for** $(u, v) \in \bar{\mathcal{E}}$ **do**
      Consider $\hat{\mathcal{G}} = \mathcal{G} \cup (u, v)$.
      Calculate proxy value for the spectral gap of $\hat{\mathcal{G}}$ used in Jamadandi et al. (2024):
      $\lambda_1(\mathcal{L}_{\hat{\mathcal{G}}}) \approx \lambda_1(\mathcal{L}_{\mathcal{G}}) + ((f_u - f_v)^2 - \lambda_1(\mathcal{L}_{\mathcal{G}}) \cdot (f_u^2 + f_v^2))$
    **end for**
    Find the edge that maximizes the proxy: $(u^+, v^+) = \underset{(u,v) \in \bar{\mathcal{E}}}{\arg\min} \lambda_1(\mathcal{L}_{\hat{\mathcal{G}}})$.
    Update graph edges: $\mathcal{E} = \mathcal{E} \cup \{(u^+, v^+)\}$.
    Update degrees: $d_{u^+} = d_{u^+} + 1, d_{v^+} = d_{v^+} + 1$
    Update eigenvectors and eigenvalues of $\mathcal{G}$ accordingly.
  **until** $N$ edges added.
  **Output :** Denser graph $\hat{\mathcal{G}} = (\mathcal{V}, \hat{\mathcal{E}})$.

---

**Algorithm 2** Proxy Spectral Gap based Greedy Graph Sparsification (PROXYDELMIN)

---
**Require:** Graph $\mathcal{G} = (\mathcal{V}, \mathcal{E})$, num. edges to prune $N$, spectral gap $\lambda_1(\mathcal{L}_{\mathcal{G}})$, second eigenvector $f$.
  **repeat**
    **for** $(u, v) \in \mathcal{E}$ **do**
      Consider $\hat{\mathcal{G}} = \mathcal{G} \setminus (u, v)$.
      Calculate proxy value for the spectral gap of $\hat{\mathcal{G}}$ used in Jamadandi et al. (2024):
      $\lambda_1(\mathcal{L}_{\hat{\mathcal{G}}}) \approx \lambda_1(\mathcal{L}_{\mathcal{G}}) - ((f_u - f_v)^2 - \lambda_1(\mathcal{L}_{\mathcal{G}}) \cdot (f_u^2 + f_v^2))$
    **end for**
    Find the edge that minimizes the proxy: $(u^-, v^-) = \underset{(u,v) \in \mathcal{E}}{\arg\min} \lambda_1(\mathcal{L}_{\hat{\mathcal{G}}})$.
    Update graph edges: $\mathcal{E} = \mathcal{E} \setminus \{(u^-, v^-)\}$.
    Update degrees: $d_{u^-} = d_{u^-} - 1, d_{v^-} = d_{v^-} - 1$
    Update eigenvectors and eigenvalues of $\mathcal{G}$ accordingly.
  **until** $N$ edges deleted.
  **Output :** Sparse graph $\hat{\mathcal{G}} = (\mathcal{V}, \hat{\mathcal{E}})$.

---

For $P(M|L_0, C_1) = P(X'_{(L_0, C_1)} > 0)$, the calculation of the predicted label of a $(L_0, C_1)$ node follows exactly the same steps, but exchanging $p$ and $q$, as the probabilities for nodes to be connected to node $i$ are now exactly of the opposite community. So we have $P(X'_{(L_0, C_1)} > 0) \approx$

$$\Phi\left(\frac{\frac{N}{2}(2\psi - 1)(q - p)}{\sqrt{\frac{N}{2}(p + q + p(1 - p) + q(1 - q) + 2(p - q)^2\psi(1 - \psi))}}\right) = 1 - P(X'_{(L_0, C_0)} > 0)$$

And $P(M) \approx \psi P(X'_{(L_0, C_0)} > 0) + (1 - \psi)(1 - P(X'_{(L_0, C_0)} > 0)) = (1 - \psi) + (2\psi - 1)P(X'_{(L_0, C_0)} > 0)$. $\qquad\square$

## B   ALGORITHMS

### B.1   SPECTRAL GAP MINIMIZATION

We use the algorithms presented in Jamadandi et al. (2024) for adding (`ProxyAddMax`) and deleting edges (`ProxyDelMax`) based on a proxy of the spectral gap. We modify these algorithms to minimize the gap and call them `ProxyAddMin` (Alg. 1) and `ProxyDelMin` (Alg. 2).

### B.2   COMMUNITY AND FEATURE SIMILARITY ALGORITHMS

We present the three algorithm families proposed in this paper for graph rewiring: `ComMa` (Algs. 3, 4), `FeaSt` (Alg. 5), and `ComFy` (Alg. 6). Each algorithm has an addition and a deletion variant.

Furthermore, `ComMa` has two extra variants for increasing (`HigherComMa`, Alg. 3) or decreasing (`LowerComMa`, Alg. 4) the community structure.

---

**Algorithm 3** `HigherComMa`: Increasing community structure. Variants: ADD, DEL

---

**Require:** Graph $\mathcal{G} = (\mathcal{V}, \mathcal{E})$, num. edges to add (ADD) / delete (DEL) $N$.
    Calculate communities (here used Louvain): $(C_0, \ldots, C_m) = \text{CommunityDetection}(\mathcal{G})$
    **if** ADD **then**
        Consider set of edges to add: $E = \{(u, v) \in \bar{\mathcal{E}} \mid \text{Comm}[u] = \text{Comm}[v]\}$
    **else if** DEL **then**
        Consider set of edges to delete: $E = \{(u, v) \in \mathcal{E} \mid \text{Comm}[u] \neq \text{Comm}[v]\}$
    **end if**
    **repeat**
      **if** ADD **then**
        Randomly and uniformly pick an edge $e$ from $E$
        Update graph edges: $\mathcal{E} = \mathcal{E} \cup \{e\}$ (ADD) or $\mathcal{E} = \mathcal{E} \setminus \{e\}$ (DEL)
      **end if**
    **until** $N$ edges modified.
    **Output :** Modified graph $\hat{\mathcal{G}} = (\mathcal{V}, \hat{\mathcal{E}})$.

---

**Algorithm 4** `LowerComMa`: Decreasing community structure. Variants: ADD, DEL

---

**Require:** Graph $\mathcal{G} = (\mathcal{V}, \mathcal{E})$, num. edges to add (ADD) / delete (DEL) $N$.
    Calculate communities (here used Louvain): $(C_0, \ldots, C_m) = \text{CommunityDetection}(\mathcal{G})$
    **if** ADD **then**
        Consider set of edges to add: $E = \{(u, v) \in \bar{\mathcal{E}} \mid \text{Comm}[u] \neq \text{Comm}[v]\}$
    **else if** DEL **then**
        Consider set of edges to delete: $E = \{(u, v) \in \mathcal{E} \mid \text{Comm}[u] = \text{Comm}[v]\}$
    **end if**
    **repeat**
      **if** ADD **then**
        Randomly and uniformly pick an edge $e$ from $E$
        Update graph edges: $\mathcal{E} = \mathcal{E} \cup \{e\}$ (ADD) or $\mathcal{E} = \mathcal{E} \setminus \{e\}$ (DEL)
      **end if**
    **until** $N$ edges modified.
    **Output :** Modified graph $\hat{\mathcal{G}} = (\mathcal{V}, \hat{\mathcal{E}})$.

---

**Algorithm 5** `FeaSt`: Maximizing feature similarity. Variants: ADD, DEL

---

**Require:** Graph $\mathcal{G} = (\mathcal{V}, \mathcal{E})$, node features $X$, num. edges to add (ADD) / delete (DEL) $N$.
    Calculate the pairwise cosine similarity of $X$: $\text{sim}(u, v)$.
    Calculate the mean of the current graph's similarity values: $\overline{\text{sim}} = \frac{1}{|\mathcal{E}|} \sum_{(u,v) \in \mathcal{E}} \text{sim}(u, v)$
    **if** ADD **then**
      **for** $(u, v) \in \bar{\mathcal{E}}$ **do**
        Calculate the graph's mean similarity in the presence of $(u, v)$: $\text{rank}(u, v) = \frac{\overline{\text{sim}}|\mathcal{E}| + \text{sim}(u,v)}{|\mathcal{E}| + 1}$
      **end for**
    **else if** DEL **then**
      **for** $(u, v) \in \mathcal{E}$ **do**
        Calculate the graph's mean similarity in the absence of $(u, v)$: $\text{rank}(u, v) = \frac{\overline{\text{sim}}|\mathcal{E}| - \text{sim}(u,v)}{|\mathcal{E}| - 1}$
      **end for**
    **end if**
    Find top $N$ edges in the ranking: $E_N = \text{top}_N(\text{rank}(u, v))$
    Update graph edges: $\mathcal{E} = \mathcal{E} \cup E_N$ (ADD) or $\mathcal{E} = \mathcal{E} \setminus E_N$ (DEL)
    **Output :** Modified graph $\hat{\mathcal{G}} = (\mathcal{V}, \hat{\mathcal{E}})$.

---

---

**Algorithm 6** `ComFy`: Maximizing feature similarity across communities. Variants: `ADD`, `DEL`

---

**Require:** Graph $\mathcal{G} = (\mathcal{V}, \mathcal{E})$, node features $X$, (approx.) num. edges to add (`ADD`) / delete (`DEL`) $N$.

   Calculate the pairwise cosine similarity of $X$: $\text{sim}(u, v)$.
   Calculate communities (here used Louvain): $(C_0, \ldots, C_m) = \text{CommunityDetection}(\mathcal{G})$
   Consider pairs of communities: $\mathcal{C} = \{(C_i, C_j) \in C \times C \mid i \leq j\}$
   **for** $(C_i, C_j) \in \mathcal{C}$ **do**
       Existing edges between $C_i$ and $C_j$: $E_{ij} = \{(u, v) \in \mathcal{E} \mid \text{Comm}[u] = C_i \wedge \text{Comm}[v] = C_j\}$
       Calculate the mean of the existing edges' similarities: $\overline{\text{sim}}_{ij} = \frac{1}{|E_{ij}|} \sum_{(u,v) \in E_{ij}} \text{sim}(u, v)$
       **if** `ADD` **then**
           Non-existing edges between $C_i, C_j$: $\overline{E_{ij}} = \{(u, v) \in \bar{\mathcal{E}} \mid \text{Comm}[u] = C_i \wedge \text{Comm}[v] = C_j\}$

           **for** $(u, v) \in \overline{E_{ij}}$ **do**
               Calculate the mean similarity in the presence of $(u, v)$: $\text{rank}_{ij}(u, v) = \frac{\overline{\text{sim}}_{ij}|E_{ij}| + \text{sim}(u,v)}{|E_{ij}| + 1}$
           **end for**
       **else if** `DEL` **then**
           **for** $(u, v) \in E_{ij}$ **do**
               Calculate the mean similarity in the absence of $(u, v)$: $\text{rank}_{ij}(u, v) = \frac{\overline{\text{sim}}_{ij}|E_{ij}| - \text{sim}(u,v)}{|E_{ij}| - 1}$
           **end for**
       **end if**
       Calculate edge budget: $B(i, j) = \text{round}(N \cdot \frac{|C_i| \cdot |C_j|}{\sum_{(C_x, C_y) \in \mathcal{C}} |C_x| \cdot |C_y|})$
       Find top $B(i, j)$ edges in the ranking: $E_{ij}^{B(i,j)} = \text{top}_{B(i,j)}(\text{rank}_{ij}(u, v))$
       Update graph edges: $\mathcal{E} = \mathcal{E} \cup E_{ij}^{B(i,j)}$ (`ADD`) or $\mathcal{E} = \mathcal{E} \setminus E_{ij}^{B(i,j)}$ (`DEL`)
   **end for**
   **Output :** Modified graph $\hat{\mathcal{G}} = (\mathcal{V}, \hat{\mathcal{E}})$. Number of modifications is approximately $N$.

---

## C  DATASETS AND HYPERPARAMETERS

### C.1  DATASET STATISTICS

In Table 5 we provide a summary of the datasets used for the experiments (§4). We also provide various metrics such as Edge Label Informativeness (ELI) and Adjusted Homophily score proposed in (Platonov et al., 2023b). The Normalized Mutual Information (NMI), Adjusted Mutual Information (AMI) and Modularity score after performing community detection using the Louvain method to understand how informative the graph structure is. The adjustment in AMI is only necessary when comparing between sets of different size, but within a dataset the number of classes does not change. Therefore we can compare the effect of our algorithm by means of the NMI's value. However, the AMI is useful to compare the alignment across different datasets.

### C.2  DETAILS OF THE EXPERIMENTS

We use PyTorch Geometric (Fey & Lenssen, 2019) and Deep Graph Library (DGL) (Wang et al., 2019) for all our experiments. For datasets Cora, Citeseer, Pubmed, Cornell, Texas, Wisconsin, Chameleon, Squirrel and Actor we use a 60/20/20 split for train/test/validation respectively. The hyperparameters are tuned on the validation set. Our backbone model is a 2-layered GCN (Kipf & Welling, 2017). We report the final test accuracy averaged over 100 splits of the data. For datasets Roman-empire, Amazon-ratings and Minesweeper we use the code base of the authors Platonov et al. (2023c), where the datasets are split 50/25/25 for train/test/validation respectively. Our backbone model here is a 5-layered GCN and the final test accuracy is reported averaged over 10 splits. We report the hyperparameters such as the Normalized Mutual Information (NMI) between the cluster labels and the ground truth labels after community detection (Blondel et al., 2008) before and after rewiring the graph to understand how it affects the community structure-node label alignment. We also report the number of edges added and deleted to effect the required change in test accuracy. The empirical runtimes, in seconds, are presented in seconds in Tables 6,7,9,11,10,12. Our code is available here: `https://github.com/RelationalML/ComFy`.

Table 5: Dataset statistics.

| Dataset | #Nodes | #Edges | NMI | AMI | ELI | Homophily | Modularity |
|---|---|---|---|---|---|---|---|
| Cora | 2708 | 10138 | 0.4556 | 0.4489 | 0.5802 | 0.7637 | 0.8023 |
| Citeseer | 3327 | 7358 | 0.3270 | 0.3151 | 0.4437 | 0.6620 | 0.8519 |
| Pubmed | 19717 | 88648 | 0.1973 | 0.1966 | 0.4092 | 0.6860 | 0.7671 |
| Cornell | 183 | 277 | 0.1250 | 0.0202 | 0.1556 | -0.2201 | 0.6227 |
| Texas | 183 | 279 | 0.0673 | 0.0016 | 0.19234 | -0.2936 | 0.5548 |
| Wisconsin | 251 | 450 | 0.0867 | 0.0351 | 0.1310 | -0.1732 | 0.6293 |
| Chameleon | 890 | 8854 | 0.1035 | 0.0823 | 0.0138 | 0.0295 | 0.6680 |
| Squirrel | 2223 | 57850 | 0.0176 | 0.0153 | 0.0013 | 0.0086 | 0.4451 |
| Actor | 7600 | 26659 | 0.0044 | -0.0002 | 0.00017 | 0.00277 | 0.5113 |
| CS | 18333 | 163788 | 0.5528 | 0.5501 | 0.6467 | 0.7845 | 0.7321 |
| Photo | 7650 | 238162 | 0.6845 | 0.6835 | 0.6662 | 0.7850 | 0.7363 |
| Physics | 34493 | 495924 | 0.4376 | 0.4372 | 0.7222 | 0.8724 | 0.6627 |
| Roman-Empire | 22662 | 32927 | 0.0214 | 0.0030 | 0.1101 | -0.0468 | 0.9887 |
| Amazon-Ratings | 24492 | 93050 | 0.0426 | 0.0381 | 0.0398 | 0.1402 | 0.9645 |
| Minesweeper | 10000 | 39402 | 0.0011 | 0.0004 | 0.0001 | 0.0094 | 0.8860 |

Table 6: Empirical runtimes for FeaSt based rewiring.

| Dataset | AvgTestAcc | NMIBefore | NMIAfter | EdgesAdded | EdgesDeleted | Rewire Time (s) |
|---|---|---|---|---|---|---|
| Cora | 87.730±0.390 | 0.456 | 0.432 | 1000 | 0 | 9.333 |
| Cora | 90.740±0.390 | 0.456 | 0.432 | 0 | 500 | 9.637 |
| Citeseer | 78.540±0.340 | 0.327 | 0.338 | 1000 | 0 | 8.549 |
| Citeseer | 81.600±0.390 | 0.327 | 0.330 | 0 | 10 | 8.422 |
| Pubmed | 86.430±0.090 | 0.197 | 0.206 | 1000 | 0 | 89.224 |
| Pubmed | 86.760±0.100 | 0.197 | 0.196 | 0 | 50 | 89.745 |
| Cornell | 59.460±1.490 | 0.125 | 0.099 | 20 | 0 | 7.402 |
| Cornell | 51.350±1.400 | 0.125 | 0.114 | 0 | 5 | 7.525 |
| Texas | 54.050±1.510 | 0.067 | 0.063 | 5 | 0 | 7.630 |
| Texas | 64.860±1.430 | 0.067 | 0.190 | 0 | 100 | 7.568 |
| Wisconsin | 60.000±1.090 | 0.087 | 0.077 | 10 | 0 | 7.533 |
| Wisconsin | 60.000±1.270 | 0.087 | 0.134 | 0 | 50 | 7.538 |
| Chameleon | 43.260±0.620 | 0.103 | 0.103 | 20 | 0 | 9.513 |
| Chameleon | 42.700±0.690 | 0.103 | 0.103 | 0 | 20 | 9.093 |
| Squirrel | 35.510±0.440 | 0.018 | 0.018 | 50 | 0 | 14.566 |
| Squirrel | 36.400±0.360 | 0.018 | 0.018 | 0 | 100 | 13.219 |
| Actor | 31.250±0.220 | 0.004 | 0.005 | 100 | 0 | 79.587 |
| Actor | 31.970±0.210 | 0.004 | 0.006 | 0 | 100 | 78.594 |

Table 7: Empirical runtimes for FeaSt+Add+Delete.

| Dataset | AvgTestAcc | NMIBefore | NMIAfter | EdgesAdded | EdgesDeleted | Rewire Time (s) |
|---|---|---|---|---|---|---|
| Cora | 85.710±0.360 | 0.456 | 0.464 | 10 | 10 | 11.450 |
| Cora | 85.710±0.360 | 0.456 | 0.464 | 10 | 10 | 11.450 |
| Citeseer | 80.190±0.340 | 0.327 | 0.322 | 50 | 50 | 10.777 |
| Citeseer | 80.190±0.340 | 0.327 | 0.322 | 50 | 50 | 10.777 |
| Pubmed | 87.010±0.120 | 0.197 | 0.198 | 1000 | 1000 | 97.618 |
| Pubmed | 87.010±0.120 | 0.197 | 0.198 | 1000 | 1000 | 97.618 |
| Cornell | 54.050±1.620 | 0.125 | 0.115 | 5 | 5 | 8.774 |
| Cornell | 54.050±1.620 | 0.125 | 0.115 | 5 | 5 | 8.774 |
| Texas | 56.760±1.650 | 0.067 | 0.165 | 100 | 100 | 8.747 |
| Texas | 56.760±1.650 | 0.067 | 0.165 | 100 | 100 | 8.747 |
| Wisconsin | 58.000±1.260 | 0.087 | 0.120 | 50 | 50 | 8.079 |
| Wisconsin | 58.000±1.260 | 0.087 | 0.120 | 50 | 50 | 8.079 |
| Chameleon | 44.940±0.700 | 0.103 | 0.158 | 100 | 100 | 8.780 |
| Chameleon | 44.940±0.700 | 0.103 | 0.158 | 100 | 100 | 8.780 |
| Squirrel | 35.730±0.480 | 0.018 | 0.019 | 500 | 500 | 13.645 |
| Squirrel | 35.730±0.480 | 0.018 | 0.019 | 500 | 500 | 13.645 |
| Actor | 32.630±0.210 | 0.004 | 0.008 | 50 | 50 | 82.668 |
| Actor | 32.630±0.210 | 0.004 | 0.008 | 50 | 50 | 82.668 |

Table 8: Empirical runtimes for ComFy

| Dataset | AvgTestAcc | NMIBefore | NMIAfter | EdgesAdded | EdgesDeleted | Rewire Time (s) |
|---|---|---|---|---|---|---|
| Cora | 87.73±0.26 | 0.4556 | 0.4580 | 100 | 0 | 14.99 |
| Cora | 88.13±0.27 | 0.4556 | 0.44876 | 0 | 2000 | 18.39 |
| Citeseer | 77.36±0.38 | 0.32701 | 0.32492 | 100 | 0 | 11.75 |
| Citeseer | 78.07±0.35 | 0.32701 | 0.35509 | 0 | 1000 | 11.86 |
| Pubmed | 86.74±0.10 | 0.19726 | 0.19572 | 50 | 0 | 415.23 |
| Pubmed | 86.23±0.11 | 0.19726 | 0.20603 | 0 | 2000 | 415.59 |
| Cornell | 67.57±1.68 | 0.1249 | 0.0955 | 10 | 0 | 9.91 |
| Cornell | 70.27±1.50 | 0.1249 | 0.1269 | 0 | 10 | 9.50 |
| Texas | 62.16±1.52 | 0.0672 | 0.0678 | 10 | 0 | 9.74 |
| Texas | 64.86±1.51 | 0.0672 | 0.0915 | 0 | 0 | 9.74 |
| Wisconsin | 62.00±1.12 | 0.0866 | 0.1526 | 50 | 0 | 9.79 |
| Wisconsin | 66.00±1.34 | 0.0866 | 0.1180 | 0 | 50 | 10.36 |
| Chameleon | 41.57±0.83 | 0.10349 | 0.14758 | 100 | 0 | 13.98 |
| Chameleon | 45.51±0.76 | 0.10349 | 0.10340 | 0 | 1500 | 10.94 |
| Squirrel | 36.85±0.38 | 0.01762 | 0.01762 | 500 | 0 | 17.51 |
| Squirrel | 39.10±0.43 | 0.01762 | 0.01762 | 0 | 1500 | 20.98 |
| Actor | 32.30±0.25 | 0.00436 | 0.00491 | 500 | 0 | 143.81 |
| Actor | 31.12±0.19 | 0.00436 | 0.01364 | 0 | 2000 | 141.79 |

Table 9: Empirical runtimes for HigherComMa based rewiring.

| Dataset | AvgTestAcc | NMI | EdgesAdded | EdgesDeleted | FinalGap | Rewire Time (s) |
|---|---|---|---|---|---|---|
| Cora | 83.64±0.38 | 0.4531 | 100 | 0 | 0.004825 | 0.06 |
| Cora | 83.82±0.31 | 0.4565 | 0 | 127 | 0.003925 | 0.05 |
| Citeseer | 77.31±0.40 | 0.3252 | 10 | 0 | 0.001555 | 0.03 |
| Citeseer | 77.31±0.41 | 0.3273 | 0 | 10 | 0.001551 | 0.03 |
| Pubmed | 85.83±0.11 | 0.1933 | 50 | 0 | 0.014013 | 7.47 |
| Pubmed | 85.90±0.11 | 0.1975 | 0 | 50 | 0.013990 | 8.67 |
| Cornell | 49.03±1.26 | 0.1283 | 5 | 0 | 0.079053 | 0.01 |
| Cornell | 49.93±1.34 | 0.1038 | 0 | 10 | 0.000001 | 0..01 |
| Texas | 52.66±1.47 | 0.0633 | 100 | 0 | 0.072213 | 0.01 |
| Texas | 48.57±1.53 | 0.0695 | 0 | 10 | 0.062387 | 0.01 |
| Wisconsin | 50.55±1.24 | 0.0886 | 5 | 0 | 0.074916 | 0.01 |
| Wisconsin | 50.32±1.38 | 0.0866 | 0 | 10 | 0.068169 | 0.01 |
| Chameleon | 41.23±0.72 | 0.1536 | 100 | 0 | 0.006417 | 0.04 |
| Chameleon | 40.44±0.69 | 0.0875 | 0 | 20 | 0.005920 | 0.04 |
| Squirrel | 34.51±0.40 | 0.0176 | 20 | 0 | 0.051575 | 0.63 |
| Squirrel | 34.66±0.39 | 0.0150 | 0 | 1000 | 0.050114 | 0.68 |
| Actor | 30.92±0.21 | 0.0080 | 20 | 0 | 0.032282 | 6.57 |
| Actor | 30.71±0.24 | 0.0222 | 0 | 50 | 0.032679 | 6.42 |

Table 10: Empirical runtimes for HigherComMa+add+delete based rewiring.

| Dataset | AvgTestAcc | NMIAfter | EdgesAdded | EdgesDeleted | FinalGap | Rewire Time (s) |
|---|---|---|---|---|---|---|
| Cora | 83.820±0.340 | 0.463 | 50 | 100 | 0.004 | 0.06 |
| Citeseer | 77.320±0.380 | 0.332 | 10 | 50 | 0.000 | 0.03 |
| Pubmed | 85.830±0.110 | 0.195 | 100 | 50 | 0.014 | 7.14 |
| Cornell | 48.920±1.480 | 0.113 | 10 | 17 | 0.000 | 0.01 |
| Texas | 52.440±1.640 | 0.067 | 100 | 10 | 0.068 | 0.01 |
| Wisconsin | 51.350±1.400 | 0.091 | 20 | 10 | 0.069 | 0.01 |
| Chameleon | 41.220±0.750 | 0.094 | 500 | 100 | 0.004 | 0.04 |
| Squirrel | 34.700±0.400 | 0.016 | 20 | 500 | 0.051 | 0.82 |
| Actor | 30.810±0.190 | 0.025 | 100 | 50 | 0.032 | 6.88 |

Table 11: Empirical runtimes for LowerComMa based rewiring.

| Dataset | AvgTestAcc | NMIAfter | EdgesAdded | EdgesDeleted | FinalGap | Rewire Time (s) |
|---|---|---|---|---|---|---|
| Cora | 83.41±0.37 | 0.4604 | 10 | 0 | 0.008448 | 0.693458 |
| Cora | 83.61±0.35 | 0.4583 | 0 | 2 | 0.004784 | 0.663406 |
| Citeseer | 77.15±0.36 | 0.3195 | 10 | 0 | 0.001556 | 0.686848 |
| Citeseer | 77.39±0.37 | 0.3240 | 0 | 4 | 0.001555 | 0.697997 |
| Pubmed | 85.85±0.090 | 0.1899 | 50 | 0 | 0.014267 | 13.045066 |
| Pubmed | 85.90±0.10 | 0.1844 | 0 | 7 | 0.014069 | 11.933355 |
| Cornell | 51.08±1.67 | 0.0695 | 100 | 0 | 0.131261 | 7.787908 |
| Cornell | 49.69±1.43 | 0.1249 | 0 | 1 | 0.080970 | 7.696815 |
| Texas | 50.29±1.71 | 0.0516 | 100 | 0 | 0.201174 | 8.309750 |
| Wisconsin | 50.95±1.29 | 0.1094 | 20 | 0 | 0.089029 | 8.400457 |
| Wisconsin | 50.61±1.35 | 0.0886 | 0 | 4 | 0.076910 | 8.699127 |
| Chameleon | 40.61±0.64 | 0.0791 | 50 | 0 | 0.007699 | 8.595781 |
| Chameleon | 40.43±0.71 | 0.1034 | 0 | 17 | 0.006315 | 8.385635 |
| Squirrel | 34.48±0.39 | 0.0207 | 100 | 0 | 0.056416 | 10.908470 |
| Squirrel | 34.76±0.40 | 0.0166 | 0 | 12 | 0.051370 | 9.688496 |
| Actor | 30.79±0.23 | 0.0055 | 500 | 0 | 0.070495 | 17.127213 |
| Actor | 30.79±0.22 | 0.0077 | 0 | 2 | 0.032679 | 15.543629 |

Table 12: Empirical runtimes for LowerComMa+add+delete based rewiring.

| Dataset | AvgTestAcc | NMIAfter | EdgesAdded | EdgesDeleted | FinalGap | Rewire Time (s) |
|---|---|---|---|---|---|---|
| Cora | 83.73±0.32 | 0.4466 | 10 | 2 | 0.007464 | 0.660313 |
| Citeseer | 77.42±0.38 | 0.3197 | 10 | 4 | 0.001556 | 0.427487 |
| Pubmed | 85.87±0.10 | 0.2036 | 50 | 7 | 0.014268 | 13.815228 |
| Cornell | 52.36±1.52 | 0.0690 | 100 | 1 | 0.132826 | 8.077217 |
| Texas | 51.6±1.53 | 0.0516 | 100 | 0 | 0.201174 | 7.848707 |
| Wisconsin | 51.45±1.33 | 0.1096 | 5 | 4 | 0.078300 | 7.622503 |
| Chameleon | 40.9±0.66 | 0.0995 | 20 | 17 | 0.007556 | 7.997294 |
| Squirrel | 34.75±0.41 | 0.0187 | 100 | 12 | 0.056383 | 10.509906 |
| Actor | 30.85±0.23 | 0.0034 | 20 | 2 | 0.036560 | 15.892848 |

Table 13: Different community detection metrics for various datasets after applying FeaSt.

| Dataset | NMIBefore | NMIAfter | AMIBefore | AMIAfter | ModularityBefore | ModularityAfter | ELIBefore | ELIAfter | HomBefore | HomAfter | Test Acc |
|---|---|---|---|---|---|---|---|---|---|---|---|
| FeaSt-Add | | | | | | | | | | | |
| Cora | 0.4556 | 0.4726 | 0.4489 | 0.4656 | 0.8023 | 0.8021 | 0.5802 | 0.5822 | 0.7637 | 0.7659 | 89.74±0.26 |
| Citeseer | 0.3270 | 0.3384 | 0.3151 | 0.3249 | 0.8519 | 0.8401 | 0.4437 | 0.4633 | 0.6620 | 0.67998 | 79.48±0.40 |
| Chameleon | 0.1035 | 0.1035 | 0.0823 | 0.0823 | 0.6680 | 0.6680 | 0.0138 | 0.0138 | 0.0295 | 0.0295 | 44.94±0.78 |
| Squirrel | 0.0176 | 0.0167 | 0.0153 | 0.0143 | 0.4451 | 0.4451 | 0.001325 | 0.00133 | 0.00861 | 0.00869 | 35.73±0.43 |
| FeaSt-Del | | | | | | | | | | | |
| Cora | 0.4556 | 0.4497 | 0.4489 | 0.4379 | 0.8023 | 0.8039 | 0.5802 | 0.5816 | 0.7637 | 0.7645 | 87.32±0.30 |
| Citeseer | 0.3270 | 0.3400 | 0.3151 | 0.3212 | 0.8519 | 0.8558 | 0.4437 | 0.4523 | 0.6620 | 0.6694 | 78.38±1.46 |
| Chameleon | 0.1035 | 0.1047 | 0.0823 | 0.0809 | 0.6680 | 0.6655 | 0.0138 | 0.0137 | 0.0295 | 0.0297 | 47.19±0.62 |
| Squirrel | 0.0176 | 0.0184 | 0.0153 | 0.0151 | 0.4451 | 0.4453 | 0.001325 | 0.00133 | 0.00861 | 0.00865 | 37.75±0.39 |

Table 14: Different community detection metrics for various datasets after applying ComFy.

| Dataset | NMIBefore | NMIAfter | AMIBefore | AMIAfter | ModularityBefore | ModularityAfter | ELIBefore | ELIAfter | HomBefore | HomAfter | Test Acc |
|---|---|---|---|---|---|---|---|---|---|---|---|
| ComFy-Add | | | | | | | | | | | |
| Cora | 0.4556 | 0.4556 | 0.4489 | 0.4489 | 0.8023 | 0.8032 | 0.5802 | 0.5776 | 0.7637 | 0.7620 | 89.13±0.26 |
| Citeseer | 0.3270 | 0.3297 | 0.3151 | 0.3175 | 0.8519 | 0.8501 | 0.4437 | 0.44033 | 0.6620 | 0.6602 | 80.42±0.39 |
| Chameleon | 0.1035 | 0.0842 | 0.0823 | 0.0706 | 0.6680 | 0.6687 | 0.0138 | 0.0140 | 0.0295 | 0.0307 | 47.19±0.62 |
| Squirrel | 0.0176 | 0.0176 | 0.0153 | 0.0153 | 0.4451 | 0.4451 | 0.0013 | 0.0013 | 0.0086 | 0.0086 | 37.75±0.39 |
| ComFy-Del | | | | | | | | | | | |
| Cora | 0.4556 | 0.4499 | 0.4489 | 0.4382 | 0.8023 | 0.8021 | 0.5802 | 0.5806 | 0.7637 | 0.7634 | 88.33±0.31 |
| Citeseer | 0.3270 | 0.3263 | 0.3151 | 0.3133 | 0.8519 | 0.8678 | 0.4437 | 0.4461 | 0.6620 | 0.6650 | 81.37±0.36 |
| Chameleon | 0.1035 | 0.1044 | 0.0823 | 0.0705 | 0.6680 | 0.6649 | 0.0138 | 0.0139 | 0.0295 | 0.0298 | 45.51±0.64 |
| Squirrel | 0.0176 | 0.0176 | 0.0153 | 0.0153 | 0.4451 | 0.4451 | 0.0013 | 0.0013 | 0.0086 | 0.0086 | 37.75±0.42 |

## C.3 SENSITIVITY TO HYPERPARAMETERS

The hyperparameters used in the experiments are given in Table 15. For the large heterophilic datasets Roman-empire, Amazon-ratings and Minesweeper we use the model hyperparameters recommended by the authors (Platonov et al., 2023b). However, we found setting the learning rate to $3e^{-3}$ instead of $3e^{-5}$ yields better results. The GCN hyperparameters on the other datasets are tuned based on the validation set through a grid search. As is common, we found that the performance of not only our

method but GNNs in general is sensitive to the learning rate but otherwise robust across datasets and architectures. Our rewiring techniques do not require any change of the basic GNN hyperparameters. In fact, we use the same ones as the baselines. Our rewiring methods come with an additional hyperparameter, i.e., the rewiring budget (and thus how many edges are added or deleted). This is true for all the rewiring methods that have been proposed (Topping et al., 2022; Karhadkar et al., 2023; Giraldo et al., 2023; Nguyen et al., 2023; Black et al., 2023; Jamadandi et al., 2024) in the literature. In Table 16 and Table 17 we present results for 4 datasets Cora, Citeseer, Chameleon and Squirrel and how their performance varies with respect to edge modification budgets. While the performance is clearly sensitive to this choice, the rewiring budget seems to be task specific but not very architecture specific, as we could use the same budgets for different GNN variants, such as GIN and GraphSAGE.

Table 15: GCN hyperparameters used in the experiments.

| Dataset | LR | Dropout | HiddenDimension |
|---|---|---|---|
| Cora | 0.01 | 0.41 | 128 |
| Citeseer | 0.01 | 0.31 | 32 |
| Pubmed | 0.01 | 0.41 | 32 |
| Cornell | 0.001 | 0.51 | 128 |
| Texas | 0.001 | 0.51 | 128 |
| Wisconsin | 0.001 | 0.51 | 128 |
| Chameleon | 0.001 | 0.21 | 128 |
| Squirrel | 0.001 | 0.51 | 128 |
| Actor | 0.001 | 0.51 | 128 |
| CS | 0.001 | 0.51 | 512 |
| Photo | 0.01 | 0.51 | 512 |
| Physics | 0.01 | 0.51 | 512 |
| Roman-empire | 0.003 | 0.31 | 512 |
| Amazon-ratings | 0.003 | 0.31 | 512 |
| Minesweeper | 0.003 | 0.31 | 512 |

Table 16: GCN test accuracy variability for different edge budgets for FeaSt.

| | GCN+FeaSt | | | |
|---|---|---|---|---|
| Dataset | EdgesAdded | Accuracy | EdgesDeleted | Accuracy |
| Cora | 10 | 85.11±0.37 | 10 | 82.49±0.39 |
| | 50 | 79.88±0.41 | 50 | 83.70±0.34 |
| | 100 | 86.72±0.36 | 100 | 86.92±0.34 |
| | 500 | 83.90±0.35 | 500 | **90.74±0.39** |
| | 1000 | **87.73±0.39** | 1000 | 85.51±0.31 |
| Citeseer | 10 | 77.36±0.35 | 10 | **81.60±0.39** |
| | 50 | 77.59±0.37 | 50 | 74.06±0.36 |
| | 100 | 75.24±0.41 | 100 | 78.30±0.33 |
| | 500 | 75.94±0.35 | 500 | 75.71±0.39 |
| | 1000 | **78.54±0.34** | 1000 | 75.00±0.33 |
| Chameleon | 20 | **43.26±0.62** | 20 | **42.70±0.69** |
| | 50 | 38.20±0.71 | 50 | 41.01±0.68 |
| | 100 | 41.01±0.64 | 100 | 35.96±0.68 |
| | 500 | 37.08±0.64 | 500 | 40.45±0.63 |
| | 1000 | 40.45±0.62 | 1000 | 39.33±0.73 |
| Squirrel | 20 | 33.26±0.38 | 20 | 34.38±0.40 |
| | 50 | **35.51±0.44** | 50 | 35.28±0.38 |
| | 100 | 33.48±0.44 | 100 | **36.40±0.36** |
| | 500 | 33.26±0.37 | 500 | 33.71±0.39 |
| | 1000 | 33.26±0.38 | 1000 | 32.36±0.38 |

# D  ADDITIONAL EXPERIMENTS

## D.1  COMPARISON WITH VARIOUS BASELINES

We compare our proposed algorithms with other diverse methods (Bi et al., 2023; Arnaiz-Rodríguez et al., 2022; Dong & Kluger, 2023) in Table 18. In (Bi et al., 2023) the authors suggest to use multi-order moments to model a neighbor's feature distribution and propose MM-GNN to use a multi-order moment embedding and an attention mechanism to weight importance of certain nodes going beyond single statistic aggregation mechanisms such as mean, max and sum. In (Arnaiz-Rodríguez et al.,

Table 17: GCN test accuracy variability for different edge budgets for ComFy

| | GCN+ComFy | | | |
|---|---|---|---|---|
| Dataset | EdgesAdded | Accuracy | EdgesDeleted | Accuracy |
| Cora | 50 | 86.72±0.27 | 500 | 86.52±0.27 |
| | 100 | 87.73±0.26 | 1000 | 84.31±0.27 |
| | 500 | 86.12±0.32 | 1500 | 85.71±0.31 |
| | 1000 | **85.11±0.27** | 2000 | **88.13±0.27** |
| Citeseer | 50 | 77.36±0.38 | 500 | 75.24±0.38 |
| | 100 | **77.36±0.38** | 1000 | **78.07±0.35** |
| | 500 | 75.47±0.33 | 1500 | 76.42±0.36 |
| | 1000 | 75.71±0.39 | 2000 | 74.76±0.39 |
| Chameleon | 5 | 35.39±0.72 | 100 | 41.57±0.73 |
| | 10 | 38.20±0.73 | 500 | 37.08±0.69 |
| | 50 | 41.01±0.64 | 1000 | 44.38±0.69 |
| | 100 | **41.57±0.83** | 1500 | **45.51±0.76** |
| | 500 | 39.33±0.60 | 2000 | 42.13±0.74 |
| Squirrel | 5 | **36.85±0.38** | 100 | 35.51±0.41 |
| | 10 | 30.34±0.44 | 500 | 33.71±0.40 |
| | 50 | 34.16±0.41 | 1000 | 37.08±0.41 |
| | 100 | 32.81±0.37 | 1500 | **39.10±0.43** |
| | 500 | 34.61±0.42 | 2000 | 36.85±0.39 |

2022), the authors propose DiffWire, an inductive way to rewire the graph based on the Lovász bound by formulating two new layers that are interspersed between regular GNN layers. In (Dong & Kluger, 2023) the authors propose a way to de-noise the graph by proposing graph propensity score (GPS) and GPS-PE (with positional encoding) methods to rewire the graph. Although the authors call their method "graph rewiring", the proposal involves separating edges in the graph as training edges and message-passing edges and use a self-supervised link prediction task to impute edges between nodes. Note that these methods go beyond 'rewiring-as-a-pre-processing' paradigm, which is the case for all our proposed algorithms. We report the results reported in their respective papers, and hence NA for some datasets. Not all code is made available for reproducing the results. We also report our proposed algorithms with different variant of GNNs such as GIN (Xu et al., 2019) and GraphSAGE (Hamilton et al., 2017b) to emphasize on the fact that our rewiring algorithms can be combined with any GNN model. The top performance is highlighted in bold. From the table we can clearly see that our proposed algorithms outperform the chosen diverse baselines on 6 out of 9 datasets.

Table 18: Additional baselines with diverse methods.

| Method | Cora | Citeseer | Pubmed | Cornell | Texas | Wisconsin | Chameleon | Squirrel | Actor |
|---|---|---|---|---|---|---|---|---|---|
| MM-GNN | 84.21±0.56 | 73.03±0.58 | 80.26±0.69 | NA | NA | NA | 63.32 ± 1.31 | 51.38 ± 1.73 | NA |
| GCN+DiffWire | 83.66±0.60 | 72.26±0.50 | 86.07±0.10 | 69.04±2.2 | NA | 79.05±2.1 | NA | NA | 31.98±0.30 |
| GPS | 79.5±0.80 | 71.5±0.60 | 77.7±0.30 | 74.6±3.00 | 80.0±1.80 | 77.3±4.40 | 41.5±3.60 | 43.0±0.90 | **38.3±0.70** |
| GPS-PE | 80.5±0.80 | 71.5±0.40 | 77.7±0.50 | 68.6±4.70 | 75.1±4.30 | 78.8±1.50 | 37.6±1.60 | 34.9±1.30 | 36.3±0.80 |
| GCN+FeaStAdd | 87.73±0.39 | 78.54±0.34 | 86.43±0.09 | 59.46±1.49 | 54.05±1.51 | 60.00±1.09 | 43.26±0.62 | 39.33±0.73 | 31.25±0.22 |
| GCN+FeaStDel | **90.74±0.39** | **81.60±0.39** | 86.76±0.10 | 51.35±1.63 | 64.86±1.43 | 60.00±1.27 | 42.70±0.69 | 36.40±0.36 | 31.97±0.21 |
| GCN+ComFyAdd | 87.73±0.26 | 77.36±0.38 | 86.74±0.10 | 67.57±1.68 | 62.16±1.52 | 62.00±1.12 | 41.57±0.83 | 36.85±0.38 | 32.30±0.25 |
| GCN+ComFyDel | 88.13±0.27 | 78.07±0.35 | 86.23±0.11 | 70.27±1.50 | 64.86±1.51 | 66.00±1.34 | 45.51±0.76 | 39.10±0.43 | 31.12±0.19 |
| GIN+FeaStAdd | 87.12±0.34 | 75.71±0.41 | 88.36±0.11 | 51.35±1.62 | 70.27±1.48 | 62.00±1.40 | 42.70±0.64 | 38.20±0.48 | 28.62±0.23 |
| GIN+FeaStDel | 85.31±0.34 | 73.35±0.48 | **89.83±0.12** | 59.46±1.73 | 72.97±1.34 | 70.00±1.31 | 45.51±0.60 | 40.67±0.43 | 29.21±0.23 |
| GIN+ComFyAdd | 84.10±0.28 | 75.00±0.46 | 89.75±0.14 | 62.16±1.99 | 67.57±1.48 | 68.00±1.32 | 46.07±0.72 | 38.43±0.47 | 29.74±0.21 |
| GIN+ComFyDel | 85.71±0.37 | 74.29±0.39 | 88.46±0.11 | 56.76±1.60 | 67.57±1.50 | 66.00±1.42 | 51.12±0.73 | 40.67±0.54 | 30.33±0.22 |
| GraphSAGE+FeaStAdd | 89.74±0.26 | 79.48±0.40 | 86.84±0.11 | 81.08±1.46 | 75.68±1.52 | 80.00±1.04 | 44.94±0.78 | 35.73±0.43 | 37.37±0.22 |
| GraphSAGE+FeaStDel | 87.32±0.30 | 80.42±0.39 | 87.62±0.10 | 78.38±1.46 | 81.08±1.43 | 86.00±1.07 | 47.19±0.62 | 37.75±0.39 | 37.76±0.21 |
| GraphSAGE+ComFyAdd | 89.13±0.26 | 81.37±0.36 | 88.33±0.09 | **89.19±1.37** | 81.08±1.52 | **86.00±1.06** | 43.82±0.72 | 37.30±0.41 | 35.86±0.22 |
| GraphSAGE+ComFyDel | 88.33±0.31 | 81.60±0.37 | 88.03±0.11 | 78.38±1.41 | **83.78±1.47** | 78.00±1.13 | 45.51±0.64 | 37.75±0.42 | 36.45±0.22 |

## D.2 SCALABILITY

We present additional results for large homophilic graphs to understand how our proposed algorithms scale with increasing graph size. The statistics for the datasets used is presented in Table 5. We present results on CS, Physics, and Photo (Shchur et al., 2019) available as PyTorch geometric datasets. We train a two-layered GCN with the following hyperparameters, the learning rate = $\{0.001, 0.01\}$ and hidden dimension size = 512. The results are presented in Table 19. Further, we pick the largest dataset among these which is Physics with $34,493$ nodes and $495,924$ edges and run a version of our algorithms which samples the nodes randomly and calculates the feature similarity and rewires only on the subset of those nodes. We use sampling ratio 0.2 to represent 20% of the nodes. The results are presented in Table 20. Clearly from the table, we can see that our proposed algorithms are robust,

in that, we can bring down the runtime significantly and still obtain comparable accuracy to the full graph.

Table 19: Node classification results on large homophilic graphs.

| Dataset | Method | Edges Modified | Rewire Time (s) | Accuracy |
|---------|--------|----------------|-----------------|----------|
| CS | GCNBaseline | NA | NA | 91.76±0.08 |
| | FeaStAdd | 500 | 52.20 | 92.10±0.08 |
| | FeaStDel | 10000 | 53.52 | **92.71±0.06** |
| | ComFyAdd | 100 | 318.58 | 91.98±0.06 |
| | ComFyDel | 500 | 331.71 | 92.30±0.08 |
| Physics | GCNBaseline | NA | NA | 94.55±0.04 |
| | FeaStAdd | 100 | 190.24 | 94.85±0.05 |
| | FeaStDel | 500 | 192.07 | 95.01±0.05 |
| | ComFyAdd | 100 | 1282.06 | **95.04±0.05** |
| | ComFyDel | 500 | 1300.39 | 94.69±0.05 |
| Photo | GCNBaseline | NA | NA | 78.70±0.41 |
| | FeaStAdd | 100 | 42.63 | 79.10±0.47 |
| | FeaStDel | 10000 | 40.34 | 81.10±0.51 |
| | ComFyAdd | 100 | 82.49 | 77.30±0.60 |
| | ComFyDel | 1000 | 80.94 | **81.60±0.49** |

Table 20: Node classification results on Physics dataset with node sampling.

| Sampling ratio | Method | Rewire Time (s) | Accuracy |
|----------------|--------|-----------------|----------|
| 100 | FeaStAdd | 190.24 | 94.85±0.05 |
| | FeaStDel | 192.07 | 95.01±0.05 |
| | ComFyAdd | 1282.06 | **95.04±0.05** |
| | ComFyDel | 1300.39 | 94.69±0.05 |
| 20 | FeaStAdd | 32.60 | 94.60±0.05 |
| | FeaStDel | 33.03 | 94.86±0.04 |
| | ComFyAdd | 718.40 | 94.62±0.05 |
| | ComFyDel | 713.65 | 94.53±0.05 |

## D.3 RESULTS WITH DIFFERENT GNN VARIANTS

In Table 21 we present results for GIN (Xu et al., 2019) and GraphSAGE (Hamilton et al., 2017a) variants, demonstrating that our rewiring schemes are architecture agnostic and can be used as a pre-processing step to make the input graphs amenable to message-passing. We also add MLP as a baseline.

Table 21: Accuracy on node classification with GIN and GraphSAGE.

| Method | Cora | Citeseer | Pubmed | Cornell | Texas | Wisconsin | Chameleon | Squirrel | Actor |
|--------|------|----------|--------|---------|-------|-----------|-----------|----------|-------|
| MLP | 73.02±0.39 | 70.84±0.51 | 87.68±0.10 | 73.54±1.45 | 76.22±1.45 | 81.68±1.06 | 35.70±0.69 | 31.84±0.40 | 36.05±0.23 |
| GIN | 85.51±0.29 | 74.53±0.41 | 88.33±0.12 | 37.84±1.62 | 54.05±1.61 | 56.00±1.21 | 41.57±0.64 | 37.08±0.39 | 24.21±0.22 |
| GIN+FeaStAdd | 87.12±0.34 | 75.71±0.41 | 88.36±0.11 | 51.35±1.62 | 70.27±1.48 | 62.00±1.40 | 42.70±0.64 | 38.20±0.48 | 28.62±0.23 |
| GIN+FeaStDel | 85.31±0.34 | 73.35±0.48 | **89.83±0.12** | 59.46±1.73 | 72.97±1.34 | 70.00±1.31 | 45.51±0.60 | 40.67±0.43 | 29.21±0.23 |
| GIN+ComFyAdd | 84.10±0.28 | 75.00±0.46 | 89.75±0.14 | 62.16±1.99 | 67.57±1.48 | 68.00±1.32 | 46.07±0.72 | 38.43±0.47 | 29.74±0.21 |
| GIN+ComFyDel | 85.71±0.37 | 74.29±0.39 | 88.46±0.11 | 56.76±1.60 | 67.57±1.50 | 66.00±1.42 | **51.12±0.73** | **40.67±0.54** | 30.33±0.22 |
| GraphSAGE | 87.73±0.26 | 77.12±0.31 | 86.56±0.10 | 67.57±1.36 | 78.38±1.37 | 76.00±1.18 | 38.76±0.61 | 35.96±0.38 | 35.99±0.21 |
| GraphSAGE+FeaStAdd | **89.74±0.26** | 79.48±0.40 | 86.84±0.11 | 81.08±1.46 | 75.68±1.52 | 80.00±1.04 | 44.94±0.78 | 35.73±0.43 | 37.37±0.22 |
| GraphSAGE+FeaStDel | 87.32±0.30 | 80.42±0.39 | 87.62±0.10 | 78.38±1.46 | **81.08±1.43** | **86.00±1.07** | 47.19±0.62 | 37.75±0.39 | **37.76±0.21** |
| GraphSAGE+ComFyAdd | 89.13±0.26 | 81.37±0.36 | 88.33±0.09 | **89.19±1.37** | 81.08±1.52 | **86.00±1.06** | 43.82±0.72 | 37.30±0.41 | 35.86±0.22 |
| GraphSAGE+ComFyDel | 88.33±0.31 | **81.60±0.37** | 88.03±0.11 | 78.38±1.41 | 83.78±1.47 | 78.00±1.13 | 45.51±0.64 | 37.75±0.42 | 36.45±0.22 |

## D.4 OUR ALGORITHMS AGAINST FEATURE NOISE

To understand how our proposed algorithm perform in presence of feature noise, we artificially add Gaussian noise with 0 mean and standard deviation $\{0.01, 0.03, 0.05, 0.08, 0.1, 0.2\}$ controlling the level of noise. We compare our proposed algorithms FeaSt and ComFy against the baseline GCN for

increasing feature noise. We add/delete 10 edges. This is shown in Figure 7 for datasets Chameleon and Squirrel. Evidently, our proposed algorithms are robust to noise perturbations and consistently outperform the baseline by a large margin.

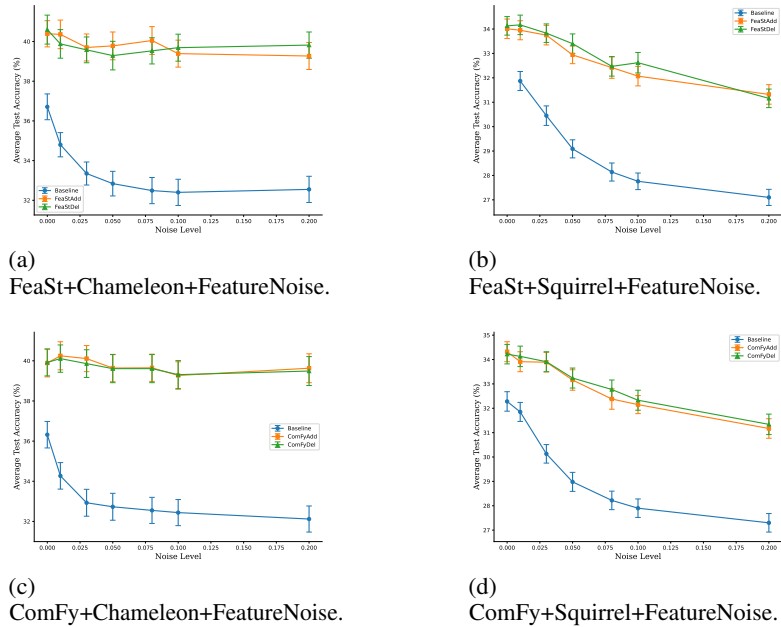

(a)
FeaSt+Chameleon+FeatureNoise.

(b)
FeaSt+Squirrel+FeatureNoise.

(c)
ComFy+Chameleon+FeatureNoise.

(d)
ComFy+Squirrel+FeatureNoise.

Figure 7: We analyse the behaviour of GCNs and our rewiring methods FeaSt and ComFy in presence of feature noise.

### D.5 OUR ALGORITHMS AGAINST LABEL NOISE

To understand how our algorithms perform in presence of label noise, we randomly flip a certain percentage of labels in the training node before rewiring the graph. We flip $\{0, 3, 5, 10, 20, 50\}$ percent of the labels and compare the baseline GCN and our methods FeaSt and ComFy with 10 edge additions/deletions. We plot the results in Figure 8, for increasing label noise, we can see that our methods are as robust as the baseline, because they lose performance at the same rate.

### D.6 OUR ALGORITHMS ON SBMS WITH LOWER COMMUNITY STRUCTURE

Using the same setups as in the proofs for Theorems 2 and 3, we evaluate the performance of ComMa (Figure 9) and ComFy (Figure 10) on SBM graphs with varying levels of community strength, under edge additions or deletions of $0, 50, 200, 500$. Classification accuracy is measured via a simple mean aggregation step across four tasks, defined by different levels of alignment between labels and communities: $\psi \in 0.7, 0.8, 0.9, 1.0$. Results are averaged over 8 seeds.

**Levels of community strength.** For a 2-class, $n$-node SBM with $(p, q) = \left( \frac{a \ln(n)}{n}, \frac{b \ln(n)}{n} \right)$, it is known (Abbe, 2018, Thm. 13) that the community structure is recoverable when $|\sqrt{a} - \sqrt{b}| > \sqrt{2}$. For $n = 100$ and $q = 0.2$, this implies a community detection threshold of $p > \left( \sqrt{\frac{0.2n}{\ln(n)}} + \sqrt{2} \right)^2 \cdot \frac{\ln(n)}{n} \approx 0.56$. Values below this threshold can be seen as having low community structure. We analyze four settings: three below and one above the threshold, with $p \in 0.3, 0.4, 0.5, 0.6$.

**Behaviour on SBMs for ComMa.** This particular SBM setup obtains performance gains as community strength increases —although this is not guaranteed for all types of graphs and tasks. In this case, HigherComMa (Figure 9) can show advantages, but will suffer when the graph structure cannot be recovered. This is the case for $p = 0.3$, especially with edge additions. However, for $p = 0.4$ and $p = 0.5$ (both still below the threshold), edge additions provide consistent benefits.

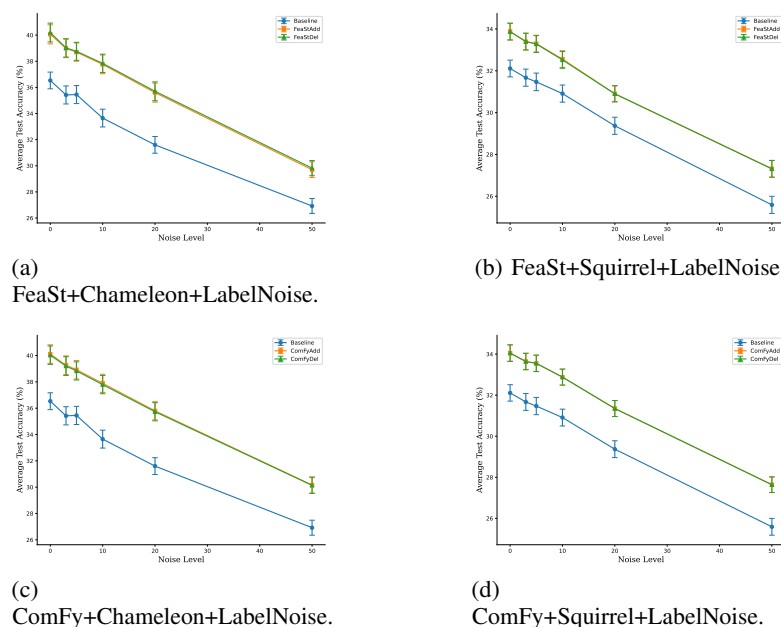

(a)
FeaSt+Chameleon+LabelNoise.

(b) FeaSt+Squirrel+LabelNoise.

(c)
ComFy+Chameleon+LabelNoise.

(d)
ComFy+Squirrel+LabelNoise.

Figure 8: We analyse the behaviour of GCNs and our rewiring methods FeaSt and ComFy in presence of label noise.

For $p = 0.6$, where the community structure becomes clearer, deletions cease to be useful, and performance plateaus as the number of deletions grows.

**Behaviour on SBMs for ComFy.** ComFy (Figure 10) is effective when the community structure is not clear, as it enhances the communities' signal via feature denoising (e.g., for $p = 0.3$). As $p$ increases, tasks with high alignment ($\psi = 1.0$) gain little from ComFy, while those with noisier label alignments ($\psi = 0.8$, $\psi = 0.7$) continue to benefit.

**Behaviour on SBMs for FeaSt.** In this simple setup with only two communities, ComFy's distribution of communities is not required for good performance. In fact, its trends match those of FeaSt, as is shown in Figure 11. Yet, Feast shows higher improvements in absolute terms (especially in high Alignment $\psi = 1$) due to the homophilic setup considered.

These trends align with our theoretical predictions (§2.3) and are also consistent with the results observed on real-world GNN benchmarks (§4).

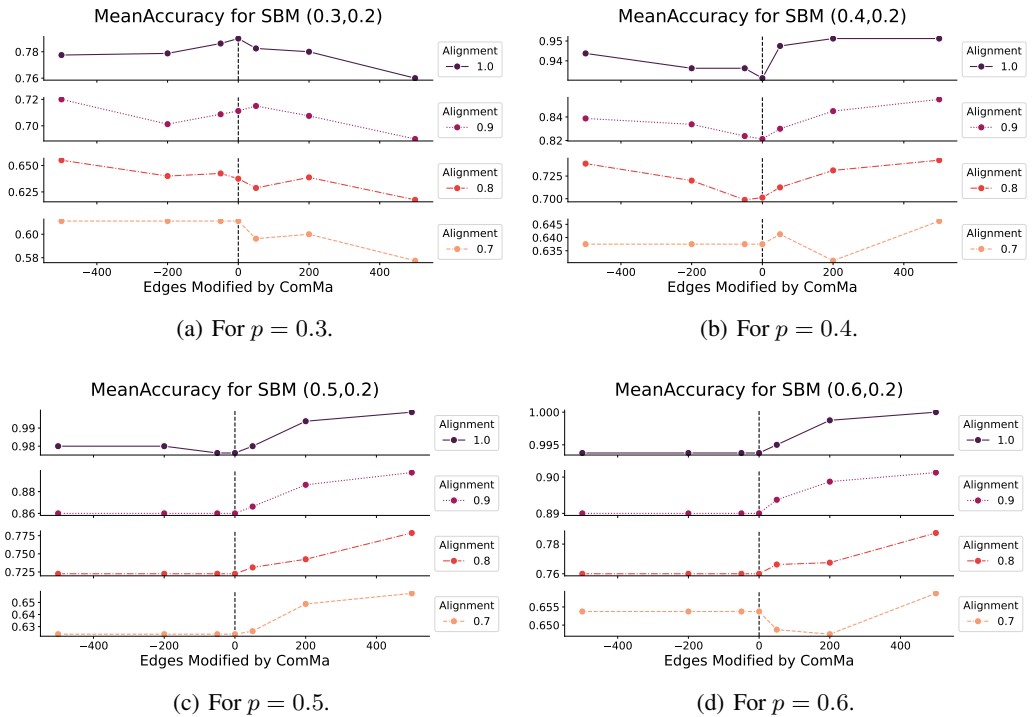

Figure 9: The effect of ComMa on mean aggregation in SBMs for low levels of community strength. Each figure is a different SBM-$(p, 0.2)$. Their rows are different levels of alignment.

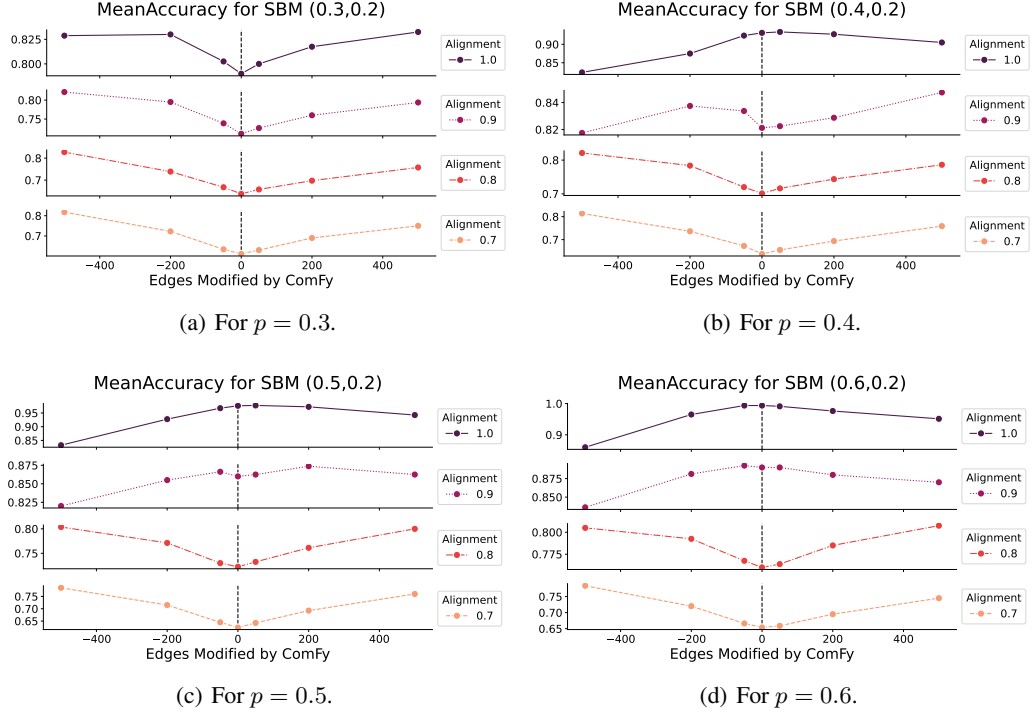

Figure 10: The effect of ComFy on mean aggregation in SBMs for low levels of community strength. Each figure is a different SBM-$(p, 0.2)$. Their rows are different levels of alignment.

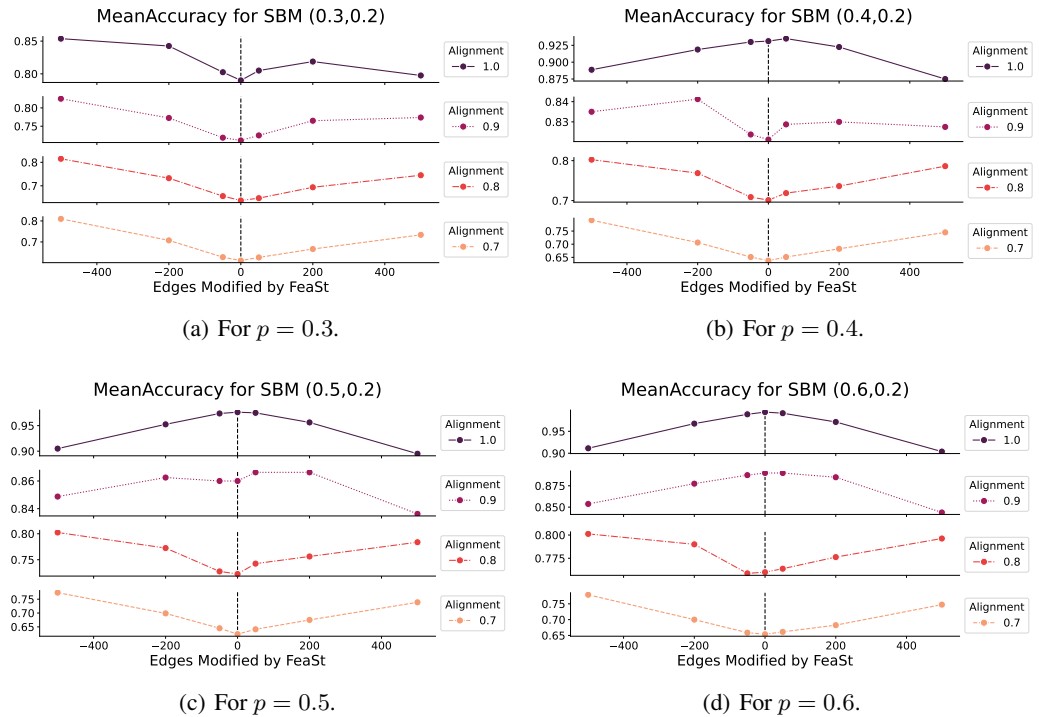

(a) For $p = 0.3$.

(b) For $p = 0.4$.

(c) For $p = 0.5$.

(d) For $p = 0.6$.

Figure 11: The effect of FeaSt on mean aggregation in SBMs for low levels of community strength. Each figure is a different SBM-$(p, 0.2)$. Their rows are different levels of alignment.

### D.7 ILLUSTRATION OF THEOREM 3

In Figure 12 we show 3D plots illustrating the expected proportion of misclassified nodes $P(M)$ in a Stochastic Block Model (SBM) as described in the setting in Theorem 3. In the plots, when $P(M)$ is low (yellow), performance is better, and when it is high (purple), performance is worse. The plot shows $P(M)$ as a function of alignment $\psi$, and different configurations of $(p, q)$:

- 12(a) for $\psi$ and the theoretical spectral gap formula in the limit $-\frac{p-q}{p+q}$ from Theorem 1.

- 12(b) and 12(c) for $p$ and $q$ given fixed $\psi \in \{0.7, 1.0\}$. As $\psi$ increases, the minimum (purple) and maximum (yellow) possible performance extend their range.

- 12(d), 12(e), 12(f) for $\psi$ and $p$ given fixed values of $q \in \{0.2, 0.5, 0.7\}$. As the value of $q$ increases, the community structure is less pronounced. Therefore, the range of values of $p$ for which performance can achieve 100% (in yellow) becomes more and more narrow.

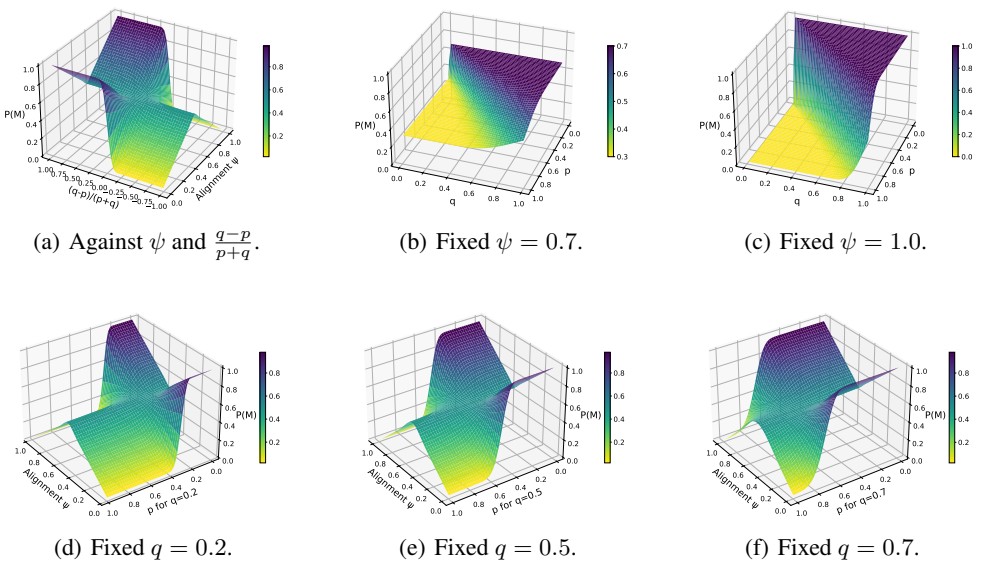

(a) Against $\psi$ and $\frac{q-p}{p+q}$.   (b) Fixed $\psi = 0.7$.   (c) Fixed $\psi = 1.0$.

(d) Fixed $q = 0.2$.   (e) Fixed $q = 0.5$.   (f) Fixed $q = 0.7$.

Figure 12: Plots illustrating the expected proportion of misclassified nodes $P(M)$ from Theorem 3.

