# OpenReview forum: "GNNs Getting ComFy: Community and Feature Similarity Guided Rewiring"
_ICLR.cc/2025/Conference — ICLR 2025 Poster_

### Official Review · Reviewer_tdhE · 2024-11-01

**Soundness:** 2
**Presentation:** 3
**Contribution:** 2
**Rating:** 6
**Confidence:** 4

**Summary:**

The authors propose different network rewiring approaches as a preprocessing step to align network structure with downstream tasks. More specifically, they consider adding or removing edges guided by the network's community structure as determined through modularity maximisation with Louvain. They also consider adding and removing links guided by the nodes' attributes. This approach is guided by the idea of presenting graph neural networks, which use the network itself as their computation graph, with more meaningful edges. By doing so, the authors hope to combat issues such as oversmooting and squashing. They evaluate their approach empirically on a set of real-world networks with homophilic and heterophil link patterns. They achieve better node classification performance than some baselines on some of those networks.

I find the paper well written and easy to follow, however, there are a few points that remained unclear to me.

**Strengths:**

- The motivation behind the work is clear and the authors demonstrate that their contribution improves the performance on one downstream task, namely node classification.
- The authors' claims are supported with theorems, proofs and extensive empirical experiments.

**Weaknesses:**

- Many results are presented in the paper, however, I find that it is not always clear what the specific message of each figure and table is. For example, I find that Fig. 1 does not have a strong message, and Fig. 5 could have been summarised more succinctly in a short paragraph.
- Only one downstream task is considered, namely node classification. Therefore, it remains unclear how the proposed approaches affect the performance in other tasks.
- While the proposed approach seems to improve node classification performance in some settings, I am concerned that this is only due to the fact that the authors essentially consider community detection, supported by modifying the network such that the communities become clearer.
- The authors' approach appears to rely on knowing the network's community structure, which is unknown in real-world networks.

**Questions:**

- What task is actually considered in the paper? While the authors discuss node classification, it seems very close to community detection to me. In that case, I don't find it surprising that node classification performance improves as the network is modified such that community structure becomes stronger. Could the authors clarify how they are doing node classification? For example, is it done in a semi-supervised fashion where 80% of the nodes have labels and the remaining 20% should be predicted by the GCN, or similar? If that is so, are ground-truth labels required for the approach to work? If labels are not used and node classification is done in an unsupervised fashion, how is it different from community detection?
- What about using other GNN architectures besides GCN or other aggregation functions besides sum aggregation? Can we expect that the proposed rewiring approaches help in those settings or are they limited to GCN and sum aggregation? I would recommend trying other architectures such as GIN or SAGE, perhaps even a MLP to check whether the rewiring might make graph neural network superfluous.
- Is there a reason why NMI was used to measure the agreement between community structures and not AMI, which has become the preferred measure in network science because it adjusts for chance?
- If I understood correctly, rewiring with FeaSt is based on the assumption that there is a correlation between nodes' community memberships and their features. But what if this is not the case? In real-world networks, it could be that the "wrong" features have been recorded. Or more generally, how do you determine whether the nodes' features are informative? Is there perhaps a "simple test" that can tell the practitioner whether FeaST will work well for their data?
- Related to the previous question, it seems that, given that there is a strong correlation between nodes' community memberships and their features, ComMa and FeaSt should not be that different. Is that correct?
- Theorem 1 assumes an SBM with exactly two equally sized blocks. How does the theorem generalise to networks with more blocks that have different sizes? And how well does the proposed insight translate to real networks?
- In Fig. 4, do the point labels denote the number of added edges?
- I find that using the Louvain algorithm to detect communities might need some more motivation or at least a discussion of its limitations. For example, modularity maximisation suffers from the well-known resolution limit, which is often "addressed" by using a resolution parameter. How does this limitation affect ComMa and ComFy?
- I think a table summarising the used datasets' statistics should be included in the paper. As of now, it is unclear how large the large networks are.

Minor points
- I think there is a typo in line 182: "grain"
- The labels in Fig. 3b are somewhat messy and I would suggest cleaning them up a bit so that they don't overlap.
- I have a hard time imagining what the approximate proportion of misclassified nodes (Theorem 3) looks like. It would have been nice to see a plot.

---

> ### Author Response · Authors · 2024-11-21
> **Response to Reviewer tdhE (1)**
>
> We thank the reviewer for the time and effort that they spent on their review. We are very grateful for their valuable feedback and revised our manuscript accordingly. Below, we provide a point by point response.
>
> **W1. Message of figures and tables:**
>
> We thank the reviewer for the feedback on the figures and tables. We have revised our manuscript accordingly. For instance, we have substituted Figure 1 by a figure that explains our proposed methods pictorially, which complements the textual descriptions in Sections 1 and 3. We believe that this improves clarity of our methods and of our algorithmic contributions. Figure 5 contains a lot of information that is critical for our main message. For that reason, we have to explain its findings in a longer paragraph.
>
> **W2. Other downstream tasks:**
>
> Our work focuses on node classification because our analysis accounts for label distributions across the nodes in the graph, and their alignment with the latent community structure. While our methods could in principle also be applied to graph classification, this is beyond our scope, as our theoretical insights provide intuition specifically for node classification. As this is a highly relevant learning setup comprising many different application domains, we believe, this focus is still of high interest to the ICLR community.
>
> **W3. Are we only making the community structure more clear?**
>
> One of our proposals, ComMa, focuses on rewiring based on community detection only. Yet, our additional proposals FeaSt and ComFy, which take feature similarity into account, outperform this proposal significantly. This shows that we are not only making the community structure more pronounced. Instead, incorporating feature similarity gives us valuable information about the task.
>
> Moreover, the purpose of our theory is to explain why it is not always beneficial to make the community structure clearer. If the alignment between community labels and node labels is not high, it can be better for the performance to make the community structure less pronounced (see Theorem 3). A real world example is given by Cornell, for instance.
>
> More generally, we consider a high variety of GNN benchmarks on which we run our algorithms. The effect of our algorithms on NMI, AMI, Modularity score, Adjusted Homophily and Edge Label Informativeness proposed in [1] before and after rewiring along with the test accuracy of GraphSAGE on Cora, Citeseer, Chameleon, and Squirrel datasets is presented in Tables 13 and 14, which shows that the effect of our feature similarity based rewiring decisions is nuanced. In general, our rewiring decisions that take feature similarity into account lead to performance gains more consistently. On purpose, rewiring usually changes too few edges to have a great impact on the graph topology and its overall community structure. This is why community structure improvement does not seem to be a dominant factor. The obtained performance gains seem to be better explained by improved feature aggregations.
>
> [1] Platonov et. al NeurIPS 2023. Characterizing Graph Datasets for Node Classification: Homophily–Heterophily Dichotomy and Beyond
>
> **W4. Unknown real community structure:**
>
> For the experimental section (S4) we use a community detection algorithm (in this case Louvain) to estimate the latent community structure. The performance of our rewiring depends less on the quality of this algorithm but more on the alignment between the detected community structure with the node labels (see Theorem 3). For graphs without a clear community structure, this is not a problem. ComFy would still provide benefits, as it still follows the feature similarity maximization principle, yet, more locally in comparison with FeaSt.
>
> **Q1. Considered task: node classification:**
>
> We focus in this paper on node classification tasks in a semi-supervised setting. The concrete splits are stated in Appendix C. To boost the performance of GNNs on these tasks, we propose three different graph rewiring methods (as pre-processing strategies). Thus, a rewired graph is provided as input to a GNN learning algorithm, which learns to perform a node classification task.
>
> It is not our objective to improve community detection. We only employ community detection (concretely the Lovain algorithm) to inform our graph rewiring (specifically ComMA and ComFy).

---

> ### Author Response · Authors · 2024-11-21
> **Response to Reviewer tdhE (2)**
>
> **Q2. More diverse GNN architectures:**
>
> To clarify, we use a 2- layered GCN (which uses mean aggregation) as our backbone architecture for all our experiments. Our proposed rewiring methods are designed to help pre-process the input graph before feeding it into a GNN, thus our algorithms can be used in conjunction with any kind of GNN model and is not restricted to any specific GNN variant. To explore the impact of our methods on other GNN architectures, we have added Table 21 below and in Section D.3 of our manuscript. It reports results for FeaSt and ComFy on GIN (sum aggregation, 2-layered) and GraphSAGE (which uses max aggregation, 2-layered) for different datasets. Our proposals lead to significant performance gains in all cases, demonstrating the versatility of our methods.
>
> Table 21
> | Method             | Cora        | Citeseer    | Pubmed       | Cornell       | Texas         | Wisconsin    | Chameleon     | Squirrel      | Actor       |
> |--------------------|-------------|-------------|--------------|---------------|---------------|--------------|---------------|---------------|-------------|
> | MLP                | 73.02±0.39  | 70.84±0.51  | 87.68±0.10   | 73.54±1.45    | 76.22±1.45    | 81.68±1.06   | 35.70±0.69    | 31.84±0.40    | 36.05±0.23  |
> | **GIN**            | 85.51±0.29  | 74.53±0.41  | 88.33±0.12   | 37.84±1.62    | 54.05±1.61    | 56.00±1.21   | 41.57±0.64    | 37.08±0.39    | 24.21±0.22  |
> | GIN+FeaStAdd       | 87.12±0.34  | 75.71±0.41  | 88.36±0.11   | 51.35±1.62    | 70.27±1.48    | 62.00±1.40   | 42.70±0.64    | 38.20±0.48    | 28.62±0.23  |
> | GIN+FeaStDel       | 85.31±0.34  | 73.35±0.48  | **89.83±0.12** | 59.46±1.73 | 72.97±1.34    | 70.00±1.31   | 45.51±0.60    | 40.67±0.43    | 29.21±0.23  |
> | GIN+ComFyAdd       | 84.10±0.28  | 75.00±0.46  | 89.75±0.14   | 62.16±1.99    | 67.57±1.48    | 68.00±1.32   | 46.07±0.72    | 38.43±0.47    | 29.74±0.21  |
> | GIN+ComFyDel       | 85.71±0.37  | 74.29±0.39  | 88.46±0.11   | 56.76±1.60    | 67.57±1.50    | 66.00±1.42   | **51.12±0.73** | **40.67±0.54** | 30.33±0.22 |
> | **GraphSAGE**      | 87.73±0.26  | 77.12±0.31  | 86.56±0.10   | 67.57±1.36    | 78.38±1.37    | 76.00±1.18   | 38.76±0.61    | 35.96±0.38    | 35.99±0.21  |
> | GraphSAGE+FeaStAdd | **89.74±0.26** | 79.48±0.40 | 86.84±0.11 | 81.08±1.46    | 75.68±1.52    | 80.00±1.04   | 44.94±0.78    | 35.73±0.43    | 37.37±0.22  |
> | GraphSAGE+FeaStDel | 87.32±0.30  | 80.42±0.39  | 87.62±0.10   | 78.38±1.46    | **81.08±1.43** | **86.00±1.07** | 47.19±0.62  | 37.75±0.39    | **37.76±0.21** |
> | GraphSAGE+ComFyAdd | 89.13±0.26  | 81.37±0.36  | 88.33±0.09   | **89.19±1.37** | **81.08±1.52** | **86.00±1.06** | 43.82±0.72  | 37.30±0.41    | 35.86±0.22  |
> | GraphSAGE+ComFyDel | 88.33±0.31  | **81.60±0.37** | 88.03±0.11 | 78.38±1.41    | 83.78±1.47    | 78.00±1.13   | 45.51±0.64    | 37.75±0.42    | 36.45±0.22  |
>
> **Q3. NMI vs. AMI:**
>
> We thank the reviewer for the suggestion to use the adjusted version of mutual information. We note that we solely use the NMI in order to compare how our algorithm changes the alignment (between class labels and community labels) within a specific dataset. The adjustment in AMI is only necessary when comparing between sets of different size, but within a dataset the number of classes does not change. Therefore we can compare the effect of our algorithm by means of the NMI’s value. However, the AMI is useful to compare the alignment across different datasets, so it is a valuable addition to our paper and we have added a discussion in the Appendix C.1. In Tables 13 and 14 we report the NMI, AMI, Modularity score, Adjusted Homophily and Edge Label Informativeness proposed in [1] before and after rewiring using FeaSt and ComFy and the test accuracy of GraphSAGE on Cora, Citeseer, Chameleon, and Squirrel datasets. In general, AMI shows similar trends as the NMI.
>
> [1] Platonov et. al NeurIPS 2023. Characterizing Graph Datasets for Node Classification: Homophily–Heterophily Dichotomy and Beyond

---

> ### Author Response · Authors · 2024-11-21
> **Response to Reviewer tdhE (3)**
>
> **Q4. Uninformative/noisy features:**
>
> We agree that real-world graphs will exhibit a variety of behaviors such as noisy graph structure and features that might be noisy or less informative which will all have different effects on the generalization performance of GNNs. In practice, we would have to evaluate different models and rewiring strategies on a validation set to identify which works best in our setting.
>
> In general, if node features are noisy or less informative, the performance of most GNNs should suffer. To analyze how our rewiring algorithms fare in the presence of feature noise, we present ablations for datasets Chameleon, and Squirrel, where we vary the amount of Gaussian noise added to the node features, and then train a GCN. As expected, the performance of the baseline GCN drops significantly. While our proposed algorithms FeaSt and ComFy are more robust to noise compared to the baseline. This is shown in plots added to section D.4 (Figure 7). In D.5 (Figure 8), we also plot the results for label noise. We randomly flip {0,3,5,10,20,50} percent of training labels and train GCN with both ComFy and FeaSt. From the figures, we can see that our proposed algorithms are robust to both feature and label noise, they consistently start at a higher test accuracy and decrease but still maintain a significant margin compared to the baseline.
>
> **Q5. Difference between ComMa and FeaSt:**
>
> ComMa and FeaSt are different even if community labels and features are highly correlated, because they rewire according to different principles. The difference also depends on the feature distribution, not only the node labels.
>
> ComMa draws edges to rewire randomly but respects the community structure in doing so. In contrast, FeaSt ranks edges according to their contribution to feature similarity without respecting the community structure. Thus, its edge modifications could be very clustered. Thus, ComMa rewires in a more distributed way. ComFy shares the distributed rewiring with ComMa but still ranks edges locally according to feature similarity.
>
> As a consequence, ComFy would be more similar to ComMa if features were distributed so that a feature similarity based ranking would result in similar rewiring decisions as a random ranking.
>
> **Q6. Extension of theory to more blocks or different sizes:**
>
> The concept of alignment between the task and the blocks still holds for more blocks and different sizes, and it is equally relevant for model performance. The fact that the spectral gap maximization destroys community strength still holds. We have extended Theorem 1’s proof to more blocks and blocks of different size in Section A.1.1. We can furthermore see how that translates to real-world datasets in Figure 4 and Section 2.4, where we run spectral based rewiring methods and calculate the accuracy versus the alignment (Fig. 4) and the number of same-class and same-community edges modified (Section 2.4).
>
> **Q7. Meaning of point labels in Figure 4:**
>
> The reviewer’s assumption is correct. We have added the information in the figure caption for better clarity. We thank the reviewer for pointing this out.
>
> **Q8. Choice of Louvain algorithm and relation to resolution limit:**
>
> Our rewiring proposals can be combined with any community detection algorithm of choice. We picked Louvain, as it scales to large graphs, is readily implemented in the Python package networkX, and can utilize GPUs. While the Louvain method sometimes produces badly connected communities, this caveat is not problematic in our setting, as we do not perform any rewiring in this case. We do not necessarily require community structure of high resolution, as we are more interested in larger communities, in which feature similarity comparisons of the edges lead to a meaningful ranking or edges for rewiring.
>
> To still analyze the impact of the community detection resolution limit, we have added Figures 9-11 to the Appendix D.6, which analyzes the impact of our proposed rewiring algorithms in our theoretical setting. We discuss scenarios below and above the community detection threshold. As our theory suggests, we find that ComMa, which relies solely on community structure, fares well in situations of pronounced community structure and good alignment between community and node labels. However, its performance deteriorates when the community structure is weak or not well detectable. In these scenarios, our proposals that take feature similarity into account induce higher performance gains, as feature similarity also carries signals about node labels and thus which nodes should be connected. This nicely motivates our methods, as it explains why it can be beneficial to combine topological information and features in rewiring decisions. If either information is less reliable or noisy, the other datasource still carries relevant information that can guide the rewiring.

---

> ### Author Response · Authors · 2024-11-21
> **Response to Reviewer tdhE (4)**
>
> **Q9. Dataset statistics:**
>
> We thank the reviewer for the suggestion to include dataset statistics, which we have added in Section C.1, Table 5.
>
> **M1. Typo:**
>
> We thank the reviewer for finding the typo grain. We have changed it to gain.
>
> **M2. Messy labels in Figure 3b:**
>
> We have adjusted the positions of labels so that none are overlapping.
>
> **M3. Visualization plots for Theorem 3 formula:**
>
> We have added a visualization to Section D.7, Figure 12.
>
> We have included 6 plots which show in six 3D snapshots the 4D surface resulting from the proportion of misclassified nodes in Theorem 3. The plot shows $P(M)$ as a function of alignment, and different configurations of $(p,q)$:
>
> * $P(M)$ against alignment and the theoretical spectral gap formula in the limit $-(p-q)/(p+q)$ from Theorem 1.
>
> * $P(M)$ against $p$ and $q$ given fixed alignment $\psi\in$  {0.7, 1.0}. As the alignment increases, the minimum (purple) and maximum (yellow) possible performance extend their range.
>
> * $P(M)$ against alignment and $p$ given fixed values of $q \in$ {0.2, 0.5, 0.7}. As the value of $q$ increases, the community structure is less pronounced. Therefore, the range of values of $p$ for which performance can achieve $100\%$ (in yellow) becomes narrower.
>
>
> References:
>
> [1] Chen et al. ICLR 2010, Supervised Community Detection with Line Graph Neural Networks.
>
> [2] Shchur et al. Deep Learning on Graphs Workshop, KDD, 2019. Overlapping Community Detection with Graph Neural Networks.
>
> [3] Hansen et al. ICML 2023. Total Variation Graph Neural Networks
>
> [4] Yang et al. ICML 2024. How Graph Neural Networks Learn: Lessons from Training Dynamics
>
> [5] Emmanuel Abbe. Journal of Machine Learning Research 2018. Community detection and stochastic block models: Recent developments

---

> ### Comment · Area_Chair_6Hrh · 2024-11-25
>
> Could please acknowledge and respond to the rebuttal.

---

> > ### Comment · Reviewer_tdhE · 2024-11-27
> >
> > I would like to thank the reviewers for their detailed answers, which addressed most of my concerns. I am still not fully convinced that the choice of community detection algorithms is not important. I don't follow the argument for why using NMI is sufficient in this scenario since we may obtain different numbers of communities, and hence, different numbers of clusters between different methods. However, as the authors have included both and also provided extensive further experiments with different GNN architectures to analyse the behaviour of their approach under different settings in addition to their detailed answers, I raise my score.

---

> > > ### Author Response · Authors · 2024-12-03
> > >
> > > We sincerely thank you for your thoughtful feedback and for raising your score. Regarding the concern about community detection algorithms, we acknowledge that the results depend on the specific method, but our framework is designed to be flexible in this regard. As for the AMI, we fully agree that it enables comparing alignment across datasets with differing numbers of classes, which is why we included it in Tables 13 and 14, as you also acknowledged. We only used the NMI in situations where the number of communities is identical in our comparison. Thank you again for your valuable input.

---

### Official Review · Reviewer_NZqY · 2024-11-03

**Soundness:** 3
**Presentation:** 3
**Contribution:** 3
**Rating:** 6
**Confidence:** 3

**Summary:**

In this paper, the authors introduced three novel graph rewiring techniques designed to improve the performance of Graph Neural Networks by focusing on the alignment between the graph structure and the target task. Experiments confirm the effectiveness of these strategies and support theoretical insights.

**Strengths:**

(1) The author established a systematic theoretical framework through the Random Block Model and analyzed the impact of community strength and graph task alignment on GNN performance. To some extent this paper enriched relevant theoretical research.
(2) The proposed FeaSt and ComFy  integrate feature similarity into graph rewiring methods, exploring how to optimize the performance of GNN by considering node features.
(3) In addition to theoretical analysis, the paper also demonstrated the performance changes of GNN under different alignment levels of graph tasks by generating Random Block Model (SBM) graphs with different alignment levels. The proposed methods have a certain degree of scalability.

**Weaknesses:**

(1) The article is mainly based on the Random Block Model (SBM) for theoretical analysis, but real-world graph data often has more complex structures and dynamic changes, and the assumptions of SBM may not fully capture these complexities.
(2) The method proposed in the article does not clearly explain and analyze the impact of adding or removing edges on model performance.
(3) The article did not discuss in detail how the proposed rewiring strategy performs in situations where the community structure is unclear or there is some noise.

**Questions:**

(1) Could the authors further explain the setting of hyper-parameters, such as increasing or deleting the number of edges, and how it affects the performance of the model?

(2) Will the performance of ComFy strategy decrease due to the unclear structure of the original community? The authors should better discuss this situation.

---

> ### Author Response · Authors · 2024-11-21
> **Response to Reviewer NZqY (1)**
>
> We thank the reviewer for the time and effort that they spent on their review. We are very grateful for their valuable feedback and revised our manuscript accordingly. Below, we provide a point by point response.
>
> **W1. Theory for SBM:**
>
> We agree that our theory focuses on the SBM, which does not capture all complexities of real world data (which also varies across datasets). However, our theory fulfills its purpose. It introduces a theoretically tractable minimal setting that provides basic insights into the mechanisms of how graph rewiring can boost GNN performance. Our theoretical results thus motivate the algorithms, which we propose, that achieve superior performance on real world datasets, as our extensive experiments demonstrate.
>
> **W2. The impact of adding and deleting edges on model performance:**
>
> Our theory captures the impact of adding and deleting edges by increasing or decreasing $p$ and $q$, respectively. Theorem 3 states how model performance is influenced by these factors.
>
> We have furthermore added Figures 9-11 to Appendix D.6, which analyzes the impact of our proposed rewiring algorithms in our theoretical setting. As our theory suggests, we find that ComMa, which relies solely on community structure, fares well in situations of pronounced community structure and good alignment between community and node labels. However, its performance deteriorates when the community structure is weak. In these scenarios, our proposals that take feature similarity into account, induce higher performance gains, as feature similarity also carries signals about node labels and thus which nodes should be connected.
>
> Furthermore, our experiments distinguish the cases of adding and deleting edges. Thus, their impact on the performance in comparison with baselines can be evaluated in detail. Our methods generally lead to significant performance gains. As a general trend, we observe that ComFy edge deletions tend to perform particularly well on heterophilic graphs, while FeaSt deletions often win in homophilic settings. Yet, also edge additions are competitive. In Table 16 and 17 we present how the test accuracy varies with respect to different edge modifications for Cora, Citeseer, Chameleon and Squirrel datasets.
>
> **W3. Sensitivity to low community structure and noise:**
>
> Even in case of less pronounced community structure, our algorithms FeaSt and ComFy still improve the feature similarity, as shown in the experimental section (S4). The Squirrel, Actor, and Texas datasets have the overall lowest modularity among the considered real world benchmarks (see Table 5). Accordingly, FeaSt achieves the highest performance on Squirrel and Texas overall, ignoring the community structure. ComFy, which also takes community structure into consideration, performs slightly better on Actor. In particular situations of less informative community structure motivate our proposal to guide rewiring by feature similarity (and not only community structure).
>
> **Robustness to noise:**
>
> We have added ablations to Sections D.4 and D.5 that study the robustness of our method ComFy and FeaSt with respect to different degrees of feature and label noise. We plot the GCN test accuracy vs feature and label noise and compare with GCN+FeaSt and GCN+ComFy both additions and deletions on chameleon and squirrel datasets. Gaussian noise with different noise level {0,0.01,0.03,0.05,0.08,0.1,0.2} is added to the node features and for label noise we randomly flip {0,3,5,10,20,50} percent of training labels. We generally find that our method responds to noise similarly as the baseline. Yet, as it starts from a higher accuracy (at 0 noise) it also maintains a higher level of performance when we increase the noise.
>
> **Q1. Sensitivity to hyperparameters:**
>
> The GCN hyperparameters are tuned based on the validation set through a grid search. We have added corresponding tables in Section C.3 (Tables 15,16,17). As is common, we found that the performance of not only our method but GNNs in general is sensitive to the learning rate but otherwise robust across datasets and architectures. Our rewiring techniques do not require any change of the basic GNN hyperparameters. In fact, we use the same ones as the baselines. Our rewiring methods come with an additional hyperparameter, i.e., the rewiring budget (and thus how many edges are added or deleted). Tables 15 and 16 in the manuscript (and the table below) present results for 4 datasets (Cora, Citeseer, Chameleon and Squirrel) and how their performance varies with respect to edge modification budgets. While the performance is clearly sensitive to this choice, the rewiring budget seems to be task specific but not very architecture specific, as we could use the same budgets for different GNN variants, such as GIN and GraphSAGE.

---

> ### Author Response · Authors · 2024-11-21
> **Response to Reviewer NZqY (2)**
>
> **Q1. (cont.)**
>
> GCN Performance wrt edge budgets for FeaSt (Table 16)
>
> | Dataset     | Edges Added | Accuracy      | Edges Deleted | Accuracy      |
> |-------------|-------------|---------------|---------------|---------------|
> | GCN+FeaSt |             |               |               |               |
> | Cora       | 10          | 85.11±0.37    | 10            | 82.49±0.39    |
> |                | 50          | 79.88±0.41    | 50            | 83.70±0.34    |
> |                | 100         | 86.72±0.36    | 100           | 86.92±0.34    |
> |                | 500         | 83.90±0.35    | 500       | 90.74±0.39|
> |                | 1000    | 87.73±0.39 | 1000          | 85.51±0.31    |
> | Citeseer   | 10          | 77.36±0.35    | 10        | 81.60±0.39 |
> |                | 50          | 77.59±0.37    | 50            | 74.06±0.36    |
> |                | 100         | 75.24±0.41    | 100           | 78.30±0.33    |
> |                | 500         | 75.94±0.35    | 500           | 75.71±0.39    |
> |                | 1000    | 78.54±0.34 | 1000      | 75.00±0.33    |
> | Chameleon  | 20          | 43.26±0.62    | 20            | 42.70±0.69    |
> |                | 50          | 38.20±0.71    | 50            | 41.01±0.68    |
> |                | 100         | 41.01±0.64    | 100           | 35.96±0.68    |
> |                | 500         | 37.08±0.64    | 500           | 40.45±0.63    |
> |                | 1000        | 40.45±0.62    | 1000          | 39.33±0.73    |
> | Squirrel   | 20          | 33.26±0.38    | 20            | 34.38±0.40    |
> |                | 50      | 35.51±0.44 | 50            | 35.28±0.38    |
> |                | 100         | 33.48±0.44    | 100       | 36.40±0.36 |
> |                | 500         | 33.26±0.37    | 500           | 33.71±0.39    |
> |                | 1000        | 33.26±0.38    | 1000          | 32.36±0.38    |
>
> GCN Performance wrt edge budgets for ComFy (Table 17)
> | Dataset    | Edges Added | Accuracy      | Edges Deleted | Accuracy      |
> |------------|-------------|---------------|---------------|---------------|
> | GCN+ComFy |             |               |               |               |
> | Cora      | 50          | 86.72±0.27    | 500           | 86.52±0.27    |
> |               | 100    | 87.73±0.26 | 1000      | 84.31±0.27    |
> |               | 500         | 86.12±0.32    | 1500      | 85.71±0.31    |
> |               | 1000        | 85.11±0.27    | 2000      | 88.13±0.27 |
> | Citeseer  | 50          | 77.36±0.38    | 500       | 75.24±0.38    |
> |               | 100     | 77.36±0.38 | 1000    | 78.07±0.35 |
> |               | 500         | 75.47±0.33    | 1500          | 76.42±0.36    |
> |               | 1000        | 75.71±0.39    | 2000      | 74.76±0.39    |
> | Chameleon| 5           | 35.39±0.72    | 100           | 41.57±0.73    |
> |               | 10          | 38.20±0.73    | 500           | 37.08±0.69    |
> |               | 50          | 41.01±0.64    | 1000          | 44.38±0.69    |
> |               | 100     | 41.57±0.83 | 1500      | 45.51±0.76 |
> |               | 500         | 39.33±0.60    | 2000          | 42.13±0.74    |
> | Squirrel  | 5       | 36.85±0.38 | 100           | 35.51±0.41    |
> |               | 10      | 30.34±0.44 | 500           | 33.71±0.40    |
> |               | 50          | 34.16±0.41    | 1000      | 37.08±0.41 |
> |               | 100         | 32.81±0.37    |1500      | 39.10±0.43 |
> |               | 500         | 34.61±0.42    | 2000          | 36.85±0.39    |
>
>
> **Q2. Sensitivity to community strength:**
>
> To study the sensitivity of our algorithms to the community strength (which we measure by the modularity score), we have added a theoretical investigation, where we can control the ground truth community strength, and report the modularity of our real world datasets to study performance trends.
>
>  **SBM:** We have furthermore added Section D.6 (Figures 9-11) to the appendix, which analyzes the impact of our proposed rewiring algorithms in our theoretical setting. We discuss scenarios below and above the community detection threshold. As our theory suggests, we find that ComMa, which relies solely on community structure, fares well in situations of pronounced community structure and good alignment between community and node labels. However, its performance deteriorates when the community structure is weak or not well detectable. In these scenarios, our proposals that take feature similarity into account induce higher performance gains, as feature similarity also carries signals about node labels and thus which nodes should be connected. This nicely motivates our methods, as it explains why it can be beneficial to combine topological information and features in rewiring decisions. If either information is less reliable or noisy, the other data source still carries relevant information that can guide the rewiring.

---

> ### Author Response · Authors · 2024-11-21
> **Response to Reviewer NZqY (3)**
>
> **Q2. (cont.)**
>
>  **Real world graphs:** Below and in Tables 13 and 14, we present a comparison of various metrics such as NMI, AMI and modularity score, Adjusted Homophily and Edge Label Informativeness proposed in [1] before and after rewiring along with the test accuracy of GraphSAGE on Cora, Citeseer, Chameleon, and Squirrel datasets. Accordingly,  feature similarity based rewiring decisions are not impacting the community structure significantly. On purpose, we do not rewire sufficiently many edges to change the graph topology so that the community structure is barely altered. Yet, the few edge alterations seem to improve the overall feature aggregation. This could also explain why our rewiring methods that take feature similarity into account achieve higher performance gains overall.
>
> References:
>
> [1] Emmanuel Abbe. Journal of Machine Learning Research 2018. Community detection and stochastic block models: Recent developments
>
> [2] Platonov et. al NeurIPS 2023. Characterizing Graph Datasets for Node Classification: Homophily–Heterophily Dichotomy and Beyond
>
> Table 13. Different community detection metrics before and after rewiring using FeaSt.
>
> | Dataset   | NMIBefore | NMIAfter | AMIBefore | AMIAfter | ModularityBefore | ModularityAfter | ELIBefore | ELIAfter | HomBefore | HomAfter | Test Acc         |
> |-----------|-----------|----------|-----------|----------|------------------|-----------------|-----------|----------|-----------|----------|------------------|
> | **FeaSt-Add** |           |          |           |          |                  |                 |           |          |           |          |                  |
> | Cora      | 0.4556    | 0.4726   | 0.4489    | 0.4656   | 0.8023           | 0.8021          | 0.5802    | 0.5822   | 0.7637    | 0.7659   | 89.74±0.26       |
> | Citeseer  | 0.3270    | 0.3384   | 0.3151    | 0.3249   | 0.8519           | 0.8401          | 0.4437    | 0.4633   | 0.6620    | 0.67998  | 79.48±0.40       |
> | Chameleon | 0.1035    | 0.1035   | 0.0823    | 0.0823   | 0.6680           | 0.6680          | 0.0138    | 0.0138   | 0.0295    | 0.0295   | 44.94±0.78       |
> | Squirrel  | 0.0176    | 0.0167   | 0.0153    | 0.0143   | 0.4451           | 0.4451          | 0.001325  | 0.00133  | 0.00861   | 0.00869  | 35.73±0.43       |
> | **FeaSt-Del** |           |          |           |          |                  |                 |           |          |           |          |                  |
> | Cora      | 0.4556    | 0.4497   | 0.4489    | 0.4379   | 0.8023           | 0.8039          | 0.5802    | 0.5816   | 0.7637    | 0.7645   | 87.32±0.30       |
> | Citeseer  | 0.3270    | 0.3400   | 0.3151    | 0.3212   | 0.8519           | 0.8558          | 0.4437    | 0.4523   | 0.6620    | 0.6694   | 78.38±1.46       |
> | Chameleon | 0.1035    | 0.1047   | 0.0823    | 0.0809   | 0.6680           | 0.6655          | 0.0138    | 0.0137   | 0.0295    | 0.0297   | 47.19±0.62       |
> | Squirrel  | 0.0176    | 0.0184   | 0.0153    | 0.0151   | 0.4451           | 0.4453          | 0.001325  | 0.00133  | 0.00861   | 0.00865  | 37.75±0.39       |

---

> ### Author Response · Authors · 2024-11-21
> **Response to Reviewer NZqY (4)**
>
> **Q2. (cont.)**
>
> Table 14. Different community detection metrics before and after rewiring using ComFy.
>
> | Dataset   | NMIBefore | NMIAfter | AMIBefore | AMIAfter | ModularityBefore | ModularityAfter | ELIBefore | ELIAfter | HomBefore | HomAfter | Test Acc         |
> |-----------|-----------|----------|-----------|----------|------------------|-----------------|-----------|----------|-----------|----------|------------------|
> | **ComFy-Add** |           |          |           |          |                  |                 |           |          |           |          |                  |
> | Cora      | 0.4556    | 0.4556   | 0.4489    | 0.4489   | 0.8023           | 0.8032          | 0.5802    | 0.5776   | 0.7637    | 0.7620   | 89.13±0.26       |
> | Citeseer  | 0.3270    | 0.3297   | 0.3151    | 0.3175   | 0.8519           | 0.8501          | 0.4437    | 0.4403   | 0.6620    | 0.6602   | 80.42±0.39       |
> | Chameleon | 0.1035    | 0.0842   | 0.0823    | 0.0706   | 0.6680           | 0.6687          | 0.0138    | 0.0140   | 0.0295    | 0.0307   | 47.19±0.62       |
> | Squirrel  | 0.0176    | 0.0176   | 0.0153    | 0.0153   | 0.4451           | 0.4451          | 0.0013    | 0.0013   | 0.0086    | 0.0086   | 37.75±0.39       |
> | **ComFy-Del** |           |          |           |          |                  |                 |           |          |           |          |                  |
> | Cora      | 0.4556    | 0.4499   | 0.4489    | 0.4382   | 0.8023           | 0.8021          | 0.5802    | 0.5806   | 0.7637    | 0.7634   | 88.33±0.31       |
> | Citeseer  | 0.3270    | 0.3263   | 0.3151    | 0.3133   | 0.8519           | 0.8678          | 0.4437    | 0.4461   | 0.6620    | 0.6650   | 81.37±0.36       |
> | Chameleon | 0.1035    | 0.1044   | 0.0823    | 0.0705   | 0.6680           | 0.6649          | 0.0138    | 0.0139   | 0.0295    | 0.0298   | 45.51±0.64       |
> | Squirrel  | 0.0176    | 0.0176   | 0.0153    | 0.0153   | 0.4451           | 0.4451          | 0.0013    | 0.0013   | 0.0086    | 0.0086   | 37.75±0.42       |

---

> > ### Comment · Reviewer_NZqY · 2024-11-25
> >
> > Thank you for all the detailed replies, and some of my concerns are addressed. I will keep my rating as 6.

---

### Official Review · Reviewer_gkN7 · 2024-11-04

**Soundness:** 3
**Presentation:** 3
**Contribution:** 3
**Rating:** 6
**Confidence:** 4

**Summary:**

The paper presents a hybrid graph rewiring method to increase classification performance on node-level classification tasks. Motivated by theoretical finidings on the planted partition model with features, the authors show that the desired edge addition and deletion strategies change deopending on whether the graph structure is homophilic or heterophilic and how strongly the targets are aligned with the structure of the graph. Therefore, the authors present ComFy, which performs the rewiring independently for each pair of communities thereby respecting the graph structure and more evenly spreading the edits over the graph.

**Strengths:**

- Good theoretical motivation through planted partition model.
- Extensive experiments on many benchmark datasets show the methods efficacy against the competition.

**Weaknesses:**

- Comfy is actually missing from the main text - not only the algorithm but also a textual explanation is very lacking.
- Planted partition model with only 2 communities is very simple and does not reflect real-world datasets.
- The elected competitors are highly related to the proposed method. A More diverse competition would be beneficial.

**Questions:**

- How do the proposed methods scale with increasing graph size? (Computing pairwise feature similarity may already be a problem for large graphs.) Are there optimizations or approximations that can make them more efficient for large-scale graphs?
- How robust are these methods when dealing with noisy or incomplete data? Do they still perform well when node features or labels contain errors?
- How sensitive to hyperparameters are the models? Both concerning the buget for rewiring and the hyperparameters of the GCN?

---

> ### Author Response · Authors · 2024-11-21
> **Response to Reviewer gkN7 (1)**
>
> We thank the reviewer for the time and effort that they spent on their review. We are very grateful for their valuable feedback and revised our manuscript accordingly. Below, we provide a point by point response.
>
> **W1. Explanation of ComFy:**
>
> In the textual explanation for ComFy in Section 3, we were missing a link to its pseudocode “(Alg. 6)” presented in Appendix B. We thank the reviewer for pointing it out, we have now added the reference.
>
> Moreover, we have decided to include a new Figure 1 that explains all the methods pictorially, which complements the textual descriptions in Sections 1 and 3. We agree with Reviewer tdhE that the old Figure 1 was less informative. We believe that this improves clarity of our proposed methods and our contributions.
>
> **W2. Simplicity of planted partition model:**
>
> For clarity, we have focused our theory on the simplest case of two communities, as it is sufficient to provide us with all relevant insights and intuition. Nevertheless, the theory is generalizable to more communities and we have added a corresponding extension of Theorem 1 to Appendix A.1.1, after its original proof.
>
> Moreover, we have shown that the same principles and intuition transfer to real world datasets in our experiments, as our proposed methods lead to significant performance gains.
>
> **W3. Diversity of competition:**
>
> Our work focuses on graph rewiring as a pre-processing step to boost GNN performance. This has the advantage that it is computationally relatively efficient. Accordingly, we have compared our proposals with state-of-the-art methods that were designed for the same purpose. In total, these define 7 baselines. Other methods to boost GNN performance are not necessarily competing but could also be seen as complementary, as they can be combined with our rewiring approach.
>
> Yet, to compare the effect of different algorithmic strategies, we have added the following comparisons to the manuscript: Table 18 in Section D.1 and Table 21 in Section D.3, where we also added additional GNN variants such as GIN and GraphSAGE and comparison with [1,2,3], which are not strictly rewiring for pre-processing but also alter the graph structure.
>
> [1] suggests multi-order moments to model a neighbor’s feature distribution and propose MM-GNN to use a multi-order moment embedding and an attention mechanism to weight importance of certain nodes going beyond single statistic aggregation mechanisms such as mean, max and sum. Note that the underlying GNN architecture is generally more powerful than a GCN.
>
> [2] proposes DiffWire, an inductive way to rewire the graph based on the Lovasz bound, which they use as inspiration to define novel GNN layers.
>
> [3] introduce a way to de-noise the graph based on a graph propensity score (GPS) and GPS-PE (with positional encoding). Although the authors call their method ``graph rewiring", it actually involves separating edges in the graph as training edges and message-passing edges and propose a self-supervised link prediction task to impute edges between nodes. We report the results of the respective papers, and hence have NA for some datasets. Code has not been shared for most methods.
>
> We find our proposals to be competitive on 6 out of 9 tested datasets.

---

> > ### Author Response · Authors · 2024-11-21
> > **Response to Reviewer gkN7 (5)**
> >
> > **Q3. (cont.):**
> >
> > GCN Hyperparams used for experiments (Table 15)
> >
> > | Dataset         | Learning Rate | Dropout | Hidden Dimension |
> > |-----------------|---------------|---------|------------------|
> > | Cora            | 0.01          | 0.41    | 128             |
> > | Citeseer        | 0.01          | 0.31    | 32              |
> > | Pubmed          | 0.01          | 0.41    | 32              |
> > | Cornell         | 0.001         | 0.51    | 128             |
> > | Texas           | 0.001         | 0.51    | 128             |
> > | Wisconsin       | 0.001         | 0.51    | 128             |
> > | Chameleon       | 0.001         | 0.21    | 128             |
> > | Squirrel        | 0.001         | 0.51    | 128             |
> > | Actor           | 0.001         | 0.51    | 128             |
> > | CS              | 0.001         | 0.51    | 512             |
> > | Photo           | 0.01          | 0.51    | 512             |
> > | Physics         | 0.01          | 0.51    | 512             |
> > | Roman-empire    | 0.003         | 0.31    | 512             |
> > | Amazon-ratings  | 0.003         | 0.31    | 512             |
> > | Minesweeper     | 0.003         | 0.31    | 512             |
> >
> >
> > GCN Performance wrt edge budgets for FeaSt (Table 16)
> >
> > | Dataset     | Edges Added | Accuracy      | Edges Deleted | Accuracy      |
> > |-------------|-------------|---------------|---------------|---------------|
> > | GCN+FeaSt |             |               |               |               |
> > | Cora       | 10          | 85.11±0.37    | 10            | 82.49±0.39    |
> > |                | 50          | 79.88±0.41    | 50            | 83.70±0.34    |
> > |                | 100         | 86.72±0.36    | 100           | 86.92±0.34    |
> > |                | 500         | 83.90±0.35    | 500       | 90.74±0.39|
> > |                | 1000    | 87.73±0.39 | 1000          | 85.51±0.31    |
> > | Citeseer   | 10          | 77.36±0.35    | 10        | 81.60±0.39 |
> > |                | 50          | 77.59±0.37    | 50            | 74.06±0.36    |
> > |                | 100         | 75.24±0.41    | 100           | 78.30±0.33    |
> > |                | 500         | 75.94±0.35    | 500           | 75.71±0.39    |
> > |                | 1000    | 78.54±0.34 | 1000      | 75.00±0.33    |
> > | Chameleon  | 20          | 43.26±0.62    | 20            | 42.70±0.69    |
> > |                | 50          | 38.20±0.71    | 50            | 41.01±0.68    |
> > |                | 100         | 41.01±0.64    | 100           | 35.96±0.68    |
> > |                | 500         | 37.08±0.64    | 500           | 40.45±0.63    |
> > |                | 1000        | 40.45±0.62    | 1000          | 39.33±0.73    |
> > | Squirrel   | 20          | 33.26±0.38    | 20            | 34.38±0.40    |
> > |                | 50      | 35.51±0.44 | 50            | 35.28±0.38    |
> > |                | 100         | 33.48±0.44    | 100       | 36.40±0.36 |
> > |                | 500         | 33.26±0.37    | 500           | 33.71±0.39    |
> > |                | 1000        | 33.26±0.38    | 1000          | 32.36±0.38    |
> >
> >
> >
> > GCN Performance wrt edge budgets for ComFy (Table 17)
> >
> > | Dataset    | Edges Added | Accuracy      | Edges Deleted | Accuracy      |
> > |------------|-------------|---------------|---------------|---------------|
> > | GCN+ComFy |             |               |               |               |
> > | Cora      | 50          | 86.72±0.27    | 500           | 86.52±0.27    |
> > |               | 100    | 87.73±0.26 | 1000      | 84.31±0.27    |
> > |               | 500         | 86.12±0.32    | 1500      | 85.71±0.31    |
> > |               | 1000        | 85.11±0.27    | 2000      | 88.13±0.27 |
> > | Citeseer  | 50          | 77.36±0.38    | 500       | 75.24±0.38    |
> > |               | 100     | 77.36±0.38 | 1000    | 78.07±0.35 |
> > |               | 500         | 75.47±0.33    | 1500          | 76.42±0.36    |
> > |               | 1000        | 75.71±0.39    | 2000      | 74.76±0.39    |
> > | Chameleon| 5           | 35.39±0.72    | 100           | 41.57±0.73    |
> > |               | 10          | 38.20±0.73    | 500           | 37.08±0.69    |
> > |               | 50          | 41.01±0.64    | 1000          | 44.38±0.69    |
> > |               | 100     | 41.57±0.83 | 1500      | 45.51±0.76 |
> > |               | 500         | 39.33±0.60    | 2000          | 42.13±0.74    |
> > | Squirrel  | 5       | 36.85±0.38 | 100           | 35.51±0.41    |
> > |               | 10      | 30.34±0.44 | 500           | 33.71±0.40    |
> > |               | 50          | 34.16±0.41    | 1000      | 37.08±0.41 |
> > |               | 100         | 32.81±0.37    |1500      | 39.10±0.43 |
> > |               | 500         | 34.61±0.42    | 2000          | 36.85±0.39    |

---

> > > ### Comment · Reviewer_gkN7 · 2024-11-25
> > >
> > > Dear Authors,
> > >
> > > Thank you for your in depth response to my comments. My questions have all been adequately answered. Regarding my mentioned weaknesses, points 2 and 3 still stand:
> > >
> > >  - "Planted Partition Model does not reflect real-world": Even generalizing the 2 community PPM to an n-community PPM is still an assumption to the generating process that does not reflect e.g. real-world social networks that follow a power law distribution. Further, only Theorem 1 has been generalized, which Reviewer zHQo has pointed out is essentially folklore.
> > >
> > > - "Diversity of the Competition": Only one of the added baselines is a graph rewiring technique and thus contributes a fair comparison with your technique. A much better comparison would have been:
> > >
> > >     1. Brüel-Gabrielsson, Rickard, Mikhail Yurochkin, and Justin Solomon. "Rewiring with positional encodings for graph neural networks." arXiv preprint arXiv:2201.12674 (2022).
> > >     2. Shen, Xu, et al. "Graph rewiring and preprocessing for graph neural networks based on effective resistance." IEEE Transactions on Knowledge and Data Engineering (2024).
> > >     3. Bober, Jakub, et al. "Rewiring networks for graph neural network training using discrete geometry." International Conference on Complex Networks and Their Applications. Cham: Springer Nature Switzerland, 2023.
> > >
> > >
> > > My concerns still stand and I maintain my score.

---

> > > > ### Author Response · Authors · 2024-11-25
> > > >
> > > > Thank you for your detailed and constructive response and engaging in a discussion with us.
> > > >
> > > > **Simplicity of planted partition model:**
> > > >
> > > > We agree that the planted partition model is not a realistic model of most real world graphs but its simplicity is also its strength. It allows us to study the interplay between community structure and feature similarity in an isolated and well controlled setting, where more complex, potentially confounding effects are removed. Its analytical tractability further allows us to gain concrete insights into the nature of the interplay. For these reasons, the planted partition model is a common and widely accepted theoretical setting in the GNN community. It also serves as a suitable theoretical setting to identify the basic mechanisms that make our proposed rewiring algorithms effective.
> > > >
> > > > We would like to emphasize that the main purpose of our work is not to provide a theoretically sophisticated new technique to gain insights into GNNs. Our work is centered around the topic of integrating features into graph rewiring. Our theorems serve the purpose to motivate our novel algorithmic proposals, which are highly effective, as we demonstrate in extensive experiments on real world graphs.
> > > >
> > > > **Additional comparisons:**
> > > >
> > > > Thank you for the pointer to other rewiring methods. We are happy to provide additional comparisons.
> > > >
> > > > We are afraid that 1. Brüel-Gabrielsson et al. "Rewiring with positional encodings for graph neural networks." arXiv preprint arXiv:2201.12674 (2022) has neither shared their code nor evaluated their technique on standard GNN benchmark datasets. Furthermore, they do not seem to propose a graph rewiring technique but define a type of transformer layer. They propose to take k-hop neighborhood information into account and augment node features with positional encodings. Adding edges to a graph to connect k-hop neighborhoods alone would perturb the original graph and increase its density significantly. Just from a computational point of view, this approach would be highly demanding and not competitive in this sense.
> > > >
> > > > 2. Shen, Xu, et al. "Graph rewiring and preprocessing for graph neural networks based on effective resistance." IEEE Transactions on Knowledge and Data Engineering (2024) has only been published recently (we believe after the submission deadline) but we are nevertheless happy to provide a comparison below.
> > > >
> > > > 3. Bober, Jakub, et al. "Rewiring networks for graph neural network training using discrete geometry." International Conference on Complex Networks and Their Applications, 2023 is based on different versions of discrete Ricci curvature. We compare the best performance on each of these datasets as reported by the authors below.
> > > >
> > > >
> > > > | Method                | Cora         | Citeseer     | Pubmed       | Cornell       | Texas         | Wisconsin     | Chameleon     | Squirrel      | Actor         | CS           | Photo        |
> > > > |-----------------------|--------------|--------------|--------------|---------------|---------------|---------------|---------------|---------------|---------------|--------------|--------------|
> > > > | GCN+CurvatureBased    | 81.56±0.24   | 72.19±0.30   | 77.75±0.38   | 58.39±0.64    | **65.48±1.23**| 55.81±0.77    | 46.92±0.73    | 37.82±0.36    | 29.81±0.30    | 90.89±0.11   | 56.74±3.12   |
> > > > | GCN+GPER | 88.90±0.30 | 76.10±0.7    | **87.50±0.20**| 53.50±2.5    | 56.20±2.50    | 56.30±1.40    | **67.20±1.70**| NA            | NA            | NA           | NA           |
> > > > | GCN+FeaSt-Add         | 87.73±0.39   | 78.54±0.34   | 86.43±0.09   | 59.46±1.49    | 54.05±1.51    | 60.00±1.09    | 43.26±0.6     | **39.33±0.73**| 31.25±0.22    | 92.10±0.08   | 79.10±0.47   |
> > > > | GCN+FeaSt-Del         | **90.74±0.39**| **81.60±0.39**| 86.76±0.10 | 51.35±1.63    | 64.86±1.43    | 60.00±1.27    | 42.70±0.69    | 36.40±0.36    | 31.97±0.21    | **92.71±0.06**| 81.10±0.51   |
> > > > | GCN+ComFyAdd          | 87.73±0.26   | 77.36±0.38   | 86.74±0.10   | 67.57±1.68    | 62.16±1.52    | 62.00±1.12    | 41.57±0.83    | 36.85±0.38    | **32.30±0.25**| 91.98±0.06   | 77.30±0.60   |
> > > > | GCN+ComFyDel          | 88.13±0.27   | 78.07±0.35   | 86.23±0.11   | **70.27±1.50**| 64.86±1.51    | **66.00±1.34**| 45.51±0.76    | 39.10±0.43    | 31.12±0.19    | 92.30±0.08   | **81.60±0.49**|

---

> > > > > ### Comment · Reviewer_gkN7 · 2024-11-26
> > > > >
> > > > > Dear Authors,
> > > > >
> > > > > Thank you for providing additional experiments. I would like to increase my score to 7. However, this is not possible. Thus, I maintain my score.

---

> ### Author Response · Authors · 2024-11-21
> **Response to Reviewer gkN7 (2)**
>
> **W3. (cont.):**
>
> Diverse methods (Table 18)
>
> | Method           | Cora        | Citeseer    | Pubmed      | Cornell      | Texas        | Wisconsin           | Chameleon             | Squirrel             | Actor       |
> |------------------|-------------|-------------|-------------|--------------|--------------|----------------------|------------------------|-----------------------|-------------|
> | MM-GNN           | 84.21±0.56  | 73.03±0.58  | 80.26±0.69  | NA           | NA           | NA                   | **63.32 ± 1.31**       | **51.38 ± 1.73**      | NA          |
> | GCN+DiffWire     | 83.66±0.60  | 72.26±0.50  | 86.07±0.10  | 69.04±2.2    | NA           | 79.05±2.1           | NA                    | NA                    | 31.98±0.30  |
> | GPS              | 79.5±0.80   | 71.5±0.60   | 77.7±0.30   | 74.6±3.00    | 80.0±1.80    | 77.3±4.40           | 41.5±3.60             | 43.0±0.90            | **38.3±0.70** |
> | GPS-PE           | 80.5±0.80   | 71.5±0.40   | 77.7±0.50   | 68.6±4.70    | 75.1±4.30    | 78.8±1.50           | 37.6±1.60             | 34.9±1.30            | 36.3±0.80   |
> | GCN+FeaStAdd     | 87.73±0.39  | 78.54±0.34  | 86.43±0.09  | 59.46±1.49   | 54.05±1.51   | 60.00±1.09          | 43.26±0.62            | 39.33±0.73           | 31.25±0.22  |
> | GCN+FeaStDel     | **90.74±0.39** | **81.60±0.39** | 86.76±0.10 | 51.35±1.63   | 64.86±1.43   | 60.00±1.27          | 42.70±0.69            | 36.40±0.36           | 31.97±0.21  |
> | GCN+ComFyAdd     | 87.73±0.26  | 77.36±0.38  | 86.74±0.10  | 67.57±1.68   | 62.16±1.52   | 62.00±1.12          | 41.57±0.83            | 36.85±0.38           | 32.30±0.25  |
> | GCN+ComFyDel     | 88.13±0.27  | 78.07±0.35  | 86.23±0.11  | 70.27±1.50   | 64.86±1.51   | 66.00±1.34          | 45.51±0.76            | 39.10±0.43           | 31.12±0.19  |
> | GIN+FeaStAdd     | 87.12±0.34  | 75.71±0.41  | 88.36±0.11  | 51.35±1.62   | 70.27±1.48   | 62.00±1.40          | 42.70±0.64            | 38.20±0.48           | 28.62±0.23  |
> | GIN+FeaStDel     | 85.31±0.34  | 73.35±0.48  | **89.83±0.12** | 59.46±1.73 | 72.97±1.34   | 70.00±1.31          | 45.51±0.60            | 40.67±0.43           | 29.21±0.23  |
> | GIN+ComFyAdd     | 84.10±0.28  | 75.00±0.46  | 89.75±0.14  | 62.16±1.99   | 67.57±1.48   | 68.00±1.32          | 46.07±0.72            | 38.43±0.47           | 29.74±0.21  |
> | GIN+ComFyDel     | 85.71±0.37  | 74.29±0.39  | 88.46±0.11  | 56.76±1.60   | 67.57±1.50   | 66.00±1.42          | 51.12±0.73            | 40.67±0.54           | 30.33±0.22  |
> | GraphSAGE+FeaStAdd | 89.74±0.26 | 79.48±0.40 | 86.84±0.11  | 81.08±1.46   | 75.68±1.52   | 80.00±1.04          | 44.94±0.78            | 35.73±0.43           | 37.37±0.22  |
> | GraphSAGE+FeaStDel | 87.32±0.30 | 80.42±0.39 | 87.62±0.10  | 78.38±1.46   | 81.08±1.43   | 86.00±1.07          | 47.19±0.62            | 37.75±0.39           | 37.76±0.21  |
> | GraphSAGE+ComFyAdd | 89.13±0.26 | 81.37±0.36 | 88.33±0.09  | **89.19±1.37** | 81.08±1.52 | **86.00±1.06**      | 43.82±0.72            | 37.30±0.41           | 35.86±0.22  |
> | GraphSAGE+ComFyDel | 88.33±0.31 | 81.60±0.37 | 88.03±0.11  | 78.38±1.41   | **83.78±1.47** | 78.00±1.13          | 45.51±0.64            | 37.75±0.42           | 36.45±0.22  |

---

> ### Author Response · Authors · 2024-11-21
> **Response to Reviewer gkN7 (3)**
>
> **W3. (cont.):**
>
> Table 21: Our methods are applied to a more diverse set of architectures (GIN, GraphSAGE) which also correspond to other kinds of aggregation (sum and max). This demonstrates that our rewiring schemes are architecture agnostic and can be applied to diverse setups.
>
> Additional baselines with GIN, GraphSAGE and MLP (Table 21).
>
> | Method             | Cora        | Citeseer    | Pubmed       | Cornell       | Texas         | Wisconsin    | Chameleon     | Squirrel      | Actor       |
> |--------------------|-------------|-------------|--------------|---------------|---------------|--------------|---------------|---------------|-------------|
> | MLP                | 73.02±0.39  | 70.84±0.51  | 87.68±0.10   | 73.54±1.45    | 76.22±1.45    | 81.68±1.06   | 35.70±0.69    | 31.84±0.40    | 36.05±0.23  |
> | **GIN**            | 85.51±0.29  | 74.53±0.41  | 88.33±0.12   | 37.84±1.62    | 54.05±1.61    | 56.00±1.21   | 41.57±0.64    | 37.08±0.39    | 24.21±0.22  |
> | GIN+FeaStAdd       | 87.12±0.34  | 75.71±0.41  | 88.36±0.11   | 51.35±1.62    | 70.27±1.48    | 62.00±1.40   | 42.70±0.64    | 38.20±0.48    | 28.62±0.23  |
> | GIN+FeaStDel       | 85.31±0.34  | 73.35±0.48  | **89.83±0.12** | 59.46±1.73 | 72.97±1.34    | 70.00±1.31   | 45.51±0.60    | 40.67±0.43    | 29.21±0.23  |
> | GIN+ComFyAdd       | 84.10±0.28  | 75.00±0.46  | 89.75±0.14   | 62.16±1.99    | 67.57±1.48    | 68.00±1.32   | 46.07±0.72    | 38.43±0.47    | 29.74±0.21  |
> | GIN+ComFyDel       | 85.71±0.37  | 74.29±0.39  | 88.46±0.11   | 56.76±1.60    | 67.57±1.50    | 66.00±1.42   | **51.12±0.73** | **40.67±0.54** | 30.33±0.22 |
> | **GraphSAGE**      | 87.73±0.26  | 77.12±0.31  | 86.56±0.10   | 67.57±1.36    | 78.38±1.37    | 76.00±1.18   | 38.76±0.61    | 35.96±0.38    | 35.99±0.21  |
> | GraphSAGE+FeaStAdd | **89.74±0.26** | 79.48±0.40 | 86.84±0.11 | 81.08±1.46    | 75.68±1.52    | 80.00±1.04   | 44.94±0.78    | 35.73±0.43    | 37.37±0.22  |
> | GraphSAGE+FeaStDel | 87.32±0.30  | 80.42±0.39  | 87.62±0.10   | 78.38±1.46    | **81.08±1.43** | **86.00±1.07** | 47.19±0.62  | 37.75±0.39    | **37.76±0.21** |
> | GraphSAGE+ComFyAdd | 89.13±0.26  | 81.37±0.36  | 88.33±0.09   | **89.19±1.37** | **81.08±1.52** | **86.00±1.06** | 43.82±0.72  | 37.30±0.41    | 35.86±0.22  |
> | GraphSAGE+ComFyDel | 88.33±0.31  | **81.60±0.37** | 88.03±0.11 | 78.38±1.41    | 83.78±1.47    | 78.00±1.13   | 45.51±0.64    | 37.75±0.42    | 36.45±0.22  |
>
> References :
>
> [1] Bi et al. ACM, WSDM 2023., MM-GNN: Mix-Moment Graph Neural Network towards Modeling Neighborhood Feature Distribution.
>
> [2] A. Arnaiz-Rodriguez et al.,Proceedings of the First Learning on Graphs Conference (LoG 2022), PMLR 198, Virtual Event, December 9–12, 2022 DiffWire: Inductive Graph Rewiring via the Lovász Bound.
>
> [3] Dong et al. ICML 2023., Towards Understanding and Reducing Graph Structural Noise for GNNs.
>
> **Q1. Scaling with graph size:**
>
> Our run time analysis states how our algorithms scale with the number of nodes and edges of a graph. We have included the statistics of their number of nodes and edges in a new Table 5 (Section C.1). All these state that running our algorithms is usually feasible on large scale graphs. But do our methods also lead to performance gains on large graphs? To answer these questions, we have added Section D.2, where we present results on 3 large homophilic datasets (Table 19) - CS, Physics and Photo [4]. We report results for FeaSt and ComFy along with the edges modified and the time taken to rewire the graph.
>
> As a general remark, if required, we could further speed up our algorithms by sampling a subgraph and focus our rewiring on this subgraph. The algorithms would still perform relatively well, as there still are edges to be modified that can improve performance. Below and in Table 20 we present the results for a specific large homophilic graph, Physics, which has 34,493 nodes and 495,924 edges. We choose the same best-performing budget hyperparameter from Table 13 and present results for  20% sampling ratio.
>
> References:
>
> [4] Shchur et al. Relational Representation Learning Workshop (R2L 2018), NeurIPS 2018 Pitfalls of Graph Neural Network Evaluation.

---

> ### Author Response · Authors · 2024-11-21
> **Response to Reviewer gkN7 (4)**
>
> **Q1. (cont.):**
>
> Results on Large Homophilic Graphs (Table 19)
>
> | Dataset   | Method      | Edges Modified | Rewire Time (s) | Accuracy         |
> |-----------|-------------|----------------|------------------|------------------|
> | **CS**    | GCNBaseline | NA             | NA              | 91.76±0.08       |
> |           | FeaStAdd    | 500            | 52.20           | 92.10±0.08       |
> |           | FeaStDel    | 10000          | 53.52           | **92.71±0.06**   |
> |           | ComFyAdd    | 100            | 318.58          | 91.98±0.06       |
> |           | ComFyDel    | 500            | 331.71          | 92.30±0.08       |
> | **Physics** | GCNBaseline | NA           | NA              | 94.55±0.04       |
> |           | FeaStAdd    | 100            | 190.24          | 94.85±0.05       |
> |           | FeaStDel    | 500            | 192.07          | 95.01±0.05       |
> |           | ComFyAdd    | 100            | 1282.06         | **95.04±0.05**   |
> |           | ComFyDel    | 500            | 1300.39         | 94.69±0.05       |
> | **Photo** | GCNBaseline | NA             | NA              | 78.70±0.41       |
> |           | FeaStAdd    | 100            | 42.63           | 79.10±0.47       |
> |           | FeaStDel    | 10000          | 40.34           | 81.10±0.51       |
> |           | ComFyAdd    | 100            | 82.49           | 77.30±0.60       |
> |           | ComFyDel    | 1000           | 80.94           | **81.60±0.49**   |
>
> Results on Node Sampling (Table 20)
>
> | Sampling Ratio | Method   | Rewire Time (s) | Accuracy         |
> |----------------|----------|-----------------|------------------|
> | **100**        | FeaStAdd | 190.24          | 94.85±0.05       |
> |                | FeaStDel | 192.07          | 95.01±0.05       |
> |                | ComFyAdd | 1282.06         | **95.04±0.05**   |
> |                | ComFyDel | 1300.39         | 94.69±0.05       |
> | **20**         | FeaStAdd | 32.60           | 94.60±0.05       |
> |                | FeaStDel | 33.03           | 94.86±0.04       |
> |                | ComFyAdd | 718.40          | 94.62±0.05       |
> |                | ComFyDel | 713.65          | 94.53±0.05       |
>
> **Q2. Robustness of methods:**
>
> We have added ablations to Sections D.4 and D.5 that study the robustness of our method ComFy and FeaSt with respect to different degrees of feature and label noise. We plot the GCN test accuracy vs feature and label noise and compare with GCN+FeaSt and GCN+ComFy both additions and deletions on chameleon and squirrel datasets. Gaussian noise with different noise level {0,0.01,0.03,0.05,0.08,0.1,0.2} is added to the node features and for label noise we randomly flip {0,3,5,10,20,50} percent of training labels. We generally find that our method responds to noise similarly as the baseline. Yet, as it starts from a higher accuracy (at 0 noise) it also maintains a higher level of performance when we increase the noise.
>
> **Q3. Sensitivity to hyperparameters:**
>
> The GCN hyperparameters are tuned based on the validation set through a grid search. We have added corresponding tables in Section C.3 (Tables 15,16,17). As is common, we found that the performance of not only our method but GNNs in general is sensitive to the learning rate but otherwise robust across datasets and architectures. Our rewiring techniques do not require any change of the basic GNN hyperparameters. In fact, we use the same ones as the baselines. Our rewiring methods come with an additional hyperparameter, i.e., the rewiring budget (and thus how many edges are added or deleted). Tables 14 and 15 in the manuscript (and the table below) present results for 4 datasets (Cora, Citeseer, Chameleon and Squirrel) and how their performance varies with respect to edge modification budgets. While the performance is clearly sensitive to this choice, the rewiring budget seems to be task specific but not very architecture specific, as we could use the same budgets for different GNN variants, such as GIN and GraphSAGE.

---

### Official Review · Reviewer_zHQo · 2024-11-04

**Soundness:** 1
**Presentation:** 1
**Contribution:** 1
**Rating:** 3
**Confidence:** 3

**Summary:**

While the existing rewiring technique aims to maximize the spectral gap, this paper aims to minimize the spectral gap.
This paper explains the benefit of minimization through the example of a stochastic block model.
Based on the analysis, this paper proposed three rewiring strategies: comma, feast, and comfy.
This paper demonstrates the effectiveness of the proposed methods.

**Strengths:**

The extensive experimental results.

**Weaknesses:**

Thm. 1 seems to be very incremental; in which part do the authors claim the result is novel? The second eigenvalue of the Laplacian of the expected SMB is almost folklore.

Right below the Thm.1, the authors claim as follows; " increasing q and decreasing p increases the spectral gap but makes the community structure less pronounced, and vice versa"

However, if we believe that the spectral gap is proportional to q-p/q+p, the gap is larger when p=0.3 and q=0.0001 than when p=0.9 and q=0.1. However, it is intuitive that the former provides the more pronounced community structure, albeit the larger spectral gap. If the rest of the authors' observations are based on this result, how can I believe that the larger spectral gap is less pronounced community structure?

Thm. 2 seems that the features do not have assumption. However, its proof immediately assumes that the the feature follows normal distribution. Therefore, the proof of the Thm.2 seems  incomplete.

In Sec. 3, the authors' proposed three algorithms based on the theoretical observations; however, I wasn't able to pull out which part of the algorithm reflects the theoretical observations. Part of which may be due to the algorithms are in Appendix, which also may be the major formatting issue of this paper.


Since the theoretical results are not reliable, and the algorithmic improvement is unclear, I vote for rejection.

**Questions:**

See weakness section.

---

> ### Author Response · Authors · 2024-11-21
> **Response to Reviewer zHQo (1)**
>
> We thank the reviewer for the time and effort that they spent on their review. We are very grateful for their valuable feedback and revised our manuscript accordingly. Below, we provide a point by point response.
>
> **W1. Novelty and role of Theorem 1:**
>
> Theorem 1 is not our main contribution but necessary to derive the rest of the paper’s arguments. It states how the spectral gap is linked to the graph’s community strength. It implies, for instance, that instead of rewiring to optimize the spectral gap, it would be a valid and computationally more efficient approach to rewire according to the community structure. Furthermore, if spectral gap maximization is applied iteratively without taking the initial community structure into account (as proposed in the literature), the community structure might get destroyed in the process. And this, as shown in the following theorems, may have adverse consequences for model performance.
>
> **W2. How the spectral gap depends on $p$ and $q$:**
>
> Our statements are correct, but we acknowledge that the factor (q-p)/(q+p) is difficult to interpret intuitively, because it is negative. To highlight this fact, we have instead written it as -(p-q)/(q+p) in our revised draft. It does imply that a larger spectral gap corresponds to less pronounced community structure. Figure 3a also confirms this statement empirically. Note the perfect anti-correlation between community structure and the spectral gap.
>
> But let us verify this also in your given example.
>
> If one calculates the spectral gap of the normalized Laplacian (using nx.normalized_laplacian_matrix, np.linalg.eigh) and the community strength (using nx.algorithms.community.modularity_max.greedy_modularity_communities and nx.algorithms.community.modularity), on each graph setting (generated by nx.stochastic_block_model with 1000 nodes) we obtain (averaged over 10 seeds):
>
> * p=0.3, q=0.0001, spectral gap: 0.001, -(p-q)/(p+q): -0.999, modularity (community strength): 0.500
>
> * p=0.9, q=0.1, spectral gap: 0.199, -(p-q)/(p+q): -0.800, modularity (community strength):: 0.400
>
> The spectral gap and -(p-q)/(p+q) are larger for p=0.9, q=0.1, but the community strength is lower. Therefore, our theorem is in line with this case.
>
> **W3. Assumptions of Thm. 2:**
>
> We thank the reviewer for pointing this out. We have added an extra sentence to Theorem 2. Theorem 2 is supposed to have the same assumptions as Theorem 3 (except for how the labels are distributed with respect to the blocks).
>
> **W4. Relation between theoretical insights and algorithms:**
>
> Our theory and proposed algorithms study the interplay of community structure and node features in guiding graph rewiring as a pre-processing technique to improve GNN performance.
>
> We propose ComMa as a fast and parallel alternative to spectral gap rewiring methods, as our theory implies a connection between increasing (/decreasing) the spectral gap and reducing (/strengthening) the community structure. ComMa rewired based on the community structure of the original graph (and does not require repeated computations of the spectral gap or the community structure) and is therefore computationally more efficient than Greedy spectral maximization (/minimization).
>
> However, our theory also points out limitations of graph rewiring techniques that solely rely on the graph topology (like the community structure) and do not take features or node labels into account. They can only succeed if the community structure and node labels are statistically dependent and thus the alignment between community and node labels is high. The additional Figure 9 in Appendix D.6 corroborates this point, as it analyzes how ComMa struggles if the alignment is insufficient or the community structure is statistically difficult to detect.
>
> Node features provide additional information that is available even before training a model. In our theoretical SBM setting (and usually also in practice), their similarity also carries information about node labels. Features can provide valuable signals, especially, when the community structure is less reliable. Our other two rewiring methods (FeaSt and ComFy) therefore take those features into account. This is why they fare better in situations when community structure is less reliable (see Figure 10).
>
> While FeaSt relies solely on features, our theory (and the rewiring literature) suggest that there can be value in accounting for community structure. For that reason, ComFy combines community structure and feature similarity information in a joint rewiring criterion. Respecting the latent community structure, which we discover with Louvain (but any other community detection algorithm could be used instead), we maximize feature similarity locally to still perform task-specific rewiring. Thus, ComFy brings all our arguments together. ComMa and FeaSt can also be interpreted as ablations that study the impact of community structure and feature similarity as isolated rewiring criteria.

---

> ### Author Response · Authors · 2024-11-21
> **Response to Reviewer zHQo (2)**
>
> **W4. Details on algorithms:**
>
> More details on the algorithms are presented on Page 8. To clarify the exposition of our algorithms in the paper, we have included a new Fig. 1 (in exchange of the previous Fig. 1, which we and Reviewer tdhE deemed not very illustrative). The new Fig. 1 explains our proposed methods pictorially in a table. We believe that this improves the clarity of our algorithmic contributions.
> In case that issues with clarity remain, we would be happy to address them. We assure the reviewer that our theoretical results are reliable. All proofs are provided in the appendix.

---

> ### Comment · Area_Chair_6Hrh · 2024-11-25
>
> Could please acknowledge and respond to the rebuttal.

---

> > ### Comment · Reviewer_zHQo · 2024-11-27
> >
> > From the response, I really lost track of the point of this paper.  In the abstract, the authors stated that "However, as we show, minimizing the spectral gap can also improve generalization." Also, as far as I understand from this paper, in the rewiring community, the spectral gap is the key point. However, in the first response, the authors stated that the authors' result is in line with modularity. How do I read the rest of this paper mentioning the spectral gap? What is the relation of your result to the modularity? Thus, I still maintain my score.

---

> > > ### Author Response · Authors · 2024-11-29
> > >
> > > We thank you for engaging in a thoughtful discussion of our paper. As you correctly point out, spectral gap maximization has primarily been employed in the graph rewiring community to improve GNN performance. However, our work highlights scenarios where spectral gap minimization can be more effective. These scenarios arise from having different levels of alignment between latent communities and task labels.
> > >
> > > Our theoretical analysis of SBMs demonstrates that the spectral gap reflects the strength of the graph’s community structure. Specifically, maximizing the spectral gap weakens the community structure, and in cases where node labels align perfectly with these communities, this weakening adversely impacts performance. The modularity score quantifies the strength of a graph's community structure. In Figure 3b, we show that modularity (i.e., latent community strength) is perfectly anti-correlated with the spectral gap, providing experimental validation for the theorem.
> > >
> > > We conclude that, since spectral gap-based rewiring methods rely solely on topological properties, they are limited if the alignment between community structure and node labels is not ideal. To address this limitation, we propose and compare methods that leverage topology alone (ComMa), node features alone (FeaSt), and a combination of both (ComFy). We have described them more thoroughly in our answer to W4 above.
> > >
> > > We appreciate that your observations allow us to clarify our arguments. We hope this response addresses your concerns and we welcome any further questions you may have.

---

> > > > ### Author Response · Authors · 2024-12-03
> > > >
> > > > We thank Reviewer zHQo for the valuable feedback. Since the discussion period is approaching its end, we would highly appreciate feedback whether any points of confusion remain. We would be happy to provide further explanations in this case.

---

### Author Response · Authors · 2024-11-22
**Summary of Rebuttal**

We sincerely thank all reviewers for their feedback. We appreciate the reception of our work and are glad that its novelty and thoroughness have been recognized. Below, we summarize our key contributions in the paper:

1. Traditional graph rewiring techniques focus on topological information like the spectral gap and thus community structure to overcome challenges such as oversquashing. We provide an intuition under which conditions this approach is promising and can boost GNN performance.
2. Our theoretical analysis on SBMs and experiments attributes the success of spectral rewiring to the alignment of node and community labels. This implies that spectral rewiring is less effective for specific label distributions or unclear community structure. As we argue, taking features into account can therefore improve graph rewiring considerably.
3. To that end, we propose three scalable methods, which allow us to analyze the impact of feature similarity and community structure on rewiring: ComMa, FeaSt, and ComFy.
4. Extensive real-world experiments demonstrate the merit of the proposed methods to boost GNN performance and confirm the substantial potential of features to guide graph rewiring. In summary, we find that homophilic graphs tend to benefit most from FeaSt (feature similarity maximization), while heterophilic graphs excel with ComFy, which balances feature similarity and community structure.

To address specific requests and provide further clarity, we have included the following additions in the manuscript:
* New Figure 1, which explains our proposed methods pictorially in a table.
* Section A.1.1:. Extension of Theorem 1’s proof for more than 2 communities and different-sized blocks.
* Section C.1: Table 5 contains a summary of the used benchmark datasets and their properties regarding community structure and graph-task alignment.
* Section C.2: Tables 13 and 14 report how the NMI, AMI and Modularity scores change after applying FeaSt and ComFy on different datasets .
* Section C.3: How different hyperparameters affect performance.
* Section D.1: Comparison with more diverse methods.
* Section D.2: Results on scalability: on large homophilic datasets, and a version of our algorithms which operate on a subset of randomly sampled nodes.
* Section D.3: Results for MLP, GIN and GraphSAGE.
* Section D.4: Results in presence of feature noise.
* Section D.5: Results in presence of label noise.
* Section D.6: Results on lower community structure SBMs (including scenarios below the community detection threshold).
* Section D.7: 3D plots to illustrate the formula for Theorem 3, which shows the expected proportion of misclassified nodes P(M) in an SBM for different values of p, q, and alignment.

Our work addresses a critical yet underexplored dimension of graph rewiring and presents novel methods to significantly advance graph neural network performance. This work not only deepens the understanding of how graph structure impacts learning tasks, but also provides practical tools for leveraging these insights in real-world applications.

---

### Meta-Review · Area_Chair_6Hrh · 2024-12-19

**Metareview:**

This paper explores how optimizing the spectral gap and rewiring graph structures can enhance the performance of graph neural networks by leveraging community and label alignment. It introduces three new rewiring strategies—ComMa, FeaSt, and ComFy—that improve computational efficiency, global homophily, and local feature similarity while maintaining community structure, leading to better generalization and performance.

All reviewer's concerns have been addressed except of one concern by reviewer zHQo, who raised a concern during the rebuttal period about the relation of modularity and the spectral gap. It's unclear if the reply by the authors addressed the reviewer's concern, since the reviewer didn't reply. Given that 3 out of 4 reviewer gave score 6 and it's unclear if reviewer's zHQo has been addressed, I recommend acceptance of the paper.

**Additional Comments On Reviewer Discussion:**

The authors put significant effort during the rebuttal and they addressed the concerns of the reviewers.

---

### Decision · Program_Chairs · 2025-01-22

Accept (Poster)